# Von Mises-Fisher Mixture Model with Dynamic Shrinkage for Realistic Test-Time Transduction

Jiazhen Huang [1]   Zhiming Liu [1]   Changhu Wang [2]   Wei Ju [3]   Ziyue Qiao [4]   Xiao Luo [5]

## Abstract

A range of methods aim to enhance the performance of vision-language models (VLMs) at test time. Among them, transduction has emerged as a promising paradigm due to its strong compatibility and efficiency. However, realistic evaluations often involve highly imbalanced class distributions, which cause performance degradation or even collapse. In this work, we systematically revisit transduction from the perspective of penalized likelihood estimation (PLE), showing that PLE with a KL-divergence anchor term naturally yields an adaptive shrinkage behavior between prior anchors and empirical estimates. From this viewpoint, the brittleness of transductive methods can be attributed to the absence of anchoring mechanism and static modeling of the shrinkage strength. Therefore, we propose Mixture of Von Mises-Fisher Models with Dynamic Shrinkage (MOON). MOON is built upon a mixture of von Mises-Fisher distributions to model feature representations on the unit hypersphere. To handle imbalance, MOON dynamically adjusts the shrinkage strength using zero-shot priors at both instance and class levels. Thus, it suppresses unreliable assignments and prevents harmful updates from outlier classes, thereby mitigating negative transfer. MOON is model-agnostic, training-free, and requires no task-specific hyperparameter tuning. Extensive experiments further validate the advantage of MOON in both performance and efficiency.

## 1. Introduction

Vision-language models (VLMs) have achieved remarkable success in the computer vision community. By pre-training on massive image-text datasets, these models have learned strong multimodal representation capabilities that align visual and textual concepts in a shared latent space. Therefore, VLMs generalize effectively to a wide range of downstream tasks and domains, from low-level image classification (Lu et al., 2022) to high-level visual question answering (Liu et al., 2023). This process requires few or even no training data. For example, CLIP (Radford et al., 2021) trains dual encoders through large-scale image-text contrastive learning. During inference, zero-shot predictions can be simply achieved by computing image-text embedding similarities.

Motivated by this, many methods aim to enhance the zero-shot predictive performance of VLMs at test time, where only unlabeled test samples are given. Among them, the most popular *online test-time adaptation* (TTA) (Döbler et al., 2024) methods adjust model behavior on-the-fly using current incoming data streams. In parallel, a closely related line of research, often referred to as *transduction* or *transductive learning* (Liu et al., 2020), focuses on exploiting the structure of available unlabeled data to perform joint inference over all test samples within a task[1]. Specifically, transductive methods are typically achieved by performing soft probabilistic clustering at the embedding or logit level. Due to its "black-box" nature, which requires neither access to model internals nor expensive gradient backpropagation, transduction exhibits strong model-agnostic compatibility and high efficiency. Therefore, it has emerged as a particularly promising paradigm for realistic deployment.

However, in realistic test-time scenarios, only small batches of unlabeled samples are available, where the underlying class distributions are often highly imbalanced. For instance, data streams may exhibit strong temporal correlations, or only a few classes may appear within a mini-batch. Nevertheless, most existing methods and benchmarks assume that class marginals are fixed and uniform. As discussed in prior studies (Veilleux et al., 2021), this often leads to degenerate predictions and severely limits their applicability. Our evaluations in Fig. 1 further demonstrate this brittleness that under realistic settings, many transductive and online TTA methods suffer from performance degradation or even col-

---

[1]Tsinghua University, China [2]Fred Hutchinson Cancer Center, USA [3]Peking University, China [4]Great Bay University, China [5]University of Wisconsin-Madison, USA. Correspondence to: Wei Ju <juwei@pku.edu.cn>, Ziyue Qiao <ziyuejoe@gmail.com>.

*Proceedings of the 43$^{rd}$ International Conference on Machine Learning*, Seoul, South Korea. PMLR 306, 2026. Copyright 2026 by the author(s).

---

[1]This notion is commonly contrasted with *inductive learning*, where predictions are independently made for each sample.

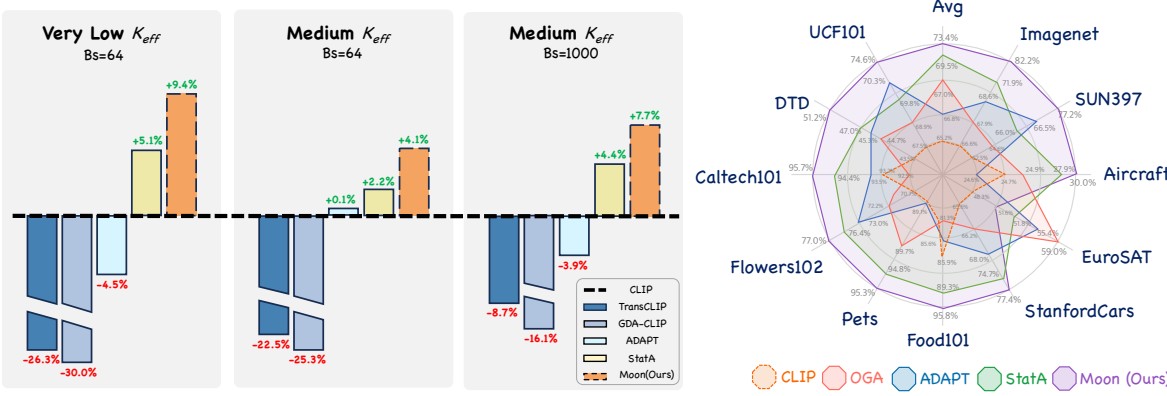

(a) Batch adaptation with a limited number of effective classes.   (b) Online adaptation under non-i.i.d. data streams.

*Figure 1.* **Performance comparison on two *realistic* settings.** Existing transductive or online TTA methods suffer from performance degradation or even collapse, while our proposed MOON consistently enhances VLM prediction and outperforms state-of-the-art baselines.

lapse. Statistically, the failures of transductive methods may arise as they implicitly treat prediction as a latent-variable estimation process. Under class imbalance, majority classes dominate the statistics, while estimates derived from limited, biased observations of minority classes become unreliable and increasingly deviate over iterations[2]. This ultimately leads to negative transfer (Wang et al., 2019).

To alleviate this issue, recent studies introduce explicit regularizations to penalize excessive deviation of empirical estimates, among which KL-divergence-based anchor terms have proven effective (Zanella et al., 2025). Based on these observations, transduction can be systematically revisited under a unified penalized likelihood estimation (PLE) formulation, where prior knowledge is incorporated as a statistical anchor. Specifically, we theoretically demonstrate that KL-anchored estimator naturally yields an adaptive shrinkage behavior, where class statistics are updated as a convex combination between prior anchors and empirical estimates. Hence, we attribute the brittleness of existing transductive methods to the following limitations: First, methods without anchoring mechanisms tend to overfit local statistics, which rapidly amplifies noisy assignments and leads to catastrophic collapse; Second, even anchor-based methods typically rely on a static modeling of shrinkage strength that implicitly assumes the reliability of statistics remains constant across samples and classes. As illustrated in Fig. 2(a), this design remains far from optimal in both accuracy and robustness, and inevitably requires task-specific hyperparameter tuning, which is impractical in practice.

Therefore, we propose MOON, a simple yet effective method for realistic test-time transduction. MOON follows the KL-anchored PLE objective and models feature representations using a mixture of von Mises-Fisher (vMF) distributions on

---

[2]This also applies to online TTA methods: although predictions are made sequentially, memory banks or distributional anchors are still influenced by previous statistics of the data stream.

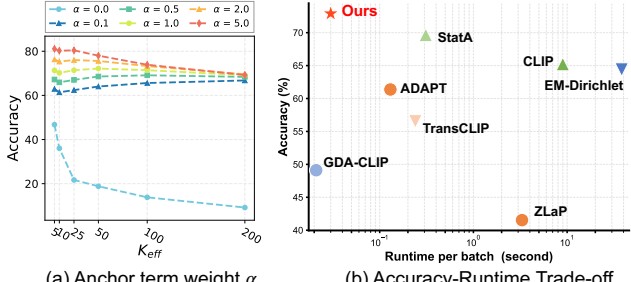

(a) Anchor term weight $\alpha$   (b) Accuracy-Runtime Trade-off

*Figure 2.* **(a) Controlling shrinkage strength with anchor weight $\alpha$ of state-of-the-art method StatA.** Such static modeling is suboptimal in accuracy and robustness. **(b) Accuracy-Runtime Tradeoff.** MOON enables effective and efficient adaptation.

the unit hypersphere. To robustly handle class imbalance, MOON dynamically adjusts the shrinkage strength using zero-shot priors at both instance and class levels. At the instance level, unreliable assignments are suppressed based on entropy, promoting a more robust label coverage. At the class level, harmful updates from outlier classes are identified and prevented, thus mitigating negative transfer. As a result, MOON enables a fine-grained, fully data-driven shrinkage, and can be seamlessly plugged into existing VLMs for enhancement. Our contribution can be concluded as follows:

❶ *New Perspective with Theoretical Support.* We systematically analyze the brittleness of existing methods under realistic class imbalance. By revisiting transduction from the perspective of penalized likelihood estimation (PLE), we theoretically prove that such estimators inherently exhibit adaptive shrinkage, which allows us to identify two limitations: absence of anchoring mechanism and static shrinkage strength modeling.

❷ *Novel Methodology.* We propose MOON, which is based on a mixture of von Mises-Fisher (vMF) distributions. It dynamically adjusts the shrinkage strength using zero-shot priors, which effectively suppresses unreliable assignments and prevents harmful updates from outlier classes, mitigating negative transfer.

❸ *Empirical Validation.* We conduct extensive experiments across 11 datasets under two realistic settings, demonstrating that MOON is: **(i) Effective**: improving zero-shot CLIP by 13.2% across 10 scenarios on ImageNet, outperforming the strongest baseline by 8.8%; **(ii) Efficient:** Being training-free, processing thousands of samples within tens of milliseconds, which is merely 3.3% of CLIP's inference latency; **(iii) Practical**: Operating under a model-agnostic black-box assumption and requiring no task-specific hyperparameter tuning, ensuring seamless deployment.

## 2. Related Work

**Enhancing VLMs at Test-Time.** A growing body of work focuses on adapting VLMs at test time to enhance their performance on downstream tasks, where only unlabeled target samples are available. Among them, the most popular test-time adaptation (TTA) methods can be broadly divided into two categories. The first focuses on updating model parameters online via lightweight fine-tuning, such as prompt tuning (Shu et al., 2022; Feng et al., 2023) or adapters (Abdul Samadh et al., 2023). This often requires expensive data augmentation and gradient backpropagation. The second avoids training and instead directly adjusts model outputs by maintaining caches (Zhang et al., 2024a), memories (Zhang et al., 2024b), or distribution modeling (Han et al., 2024) over historical data streams. Our work primarily relates to this category and refers to these methods as *online TTA*. In parallel, the concept of *transduction* was originally explored in few-shot learning (Martin et al., 2022), where it aims to exploit the structure of available data to perform joint inference over test samples. When extended to zero-shot learning, this paradigm can be viewed as a subclass within a broader TTA framework. Recent works like TransCLIP (Zanella et al., 2024) and ZLaP (Kalantidis et al., 2024) have investigated transduction for VLMs, while StatA (Zanella et al., 2025) further discusses it under the test-time class imbalance. Despite making progress, several limitations still bottleneck their performance. This motivates our MOON.

**Imbalanced Learning in Realistic Scenarios.** Most existing TTA methods and benchmarks assume perfectly class-balanced tasks at inference, i.e., the marginal class probabilities are treated as uniform. In contrast, realistic deployment scenarios often exhibit highly imbalanced class distributions, such as sparse, long-tailed, or non-i.i.d. batches (Ochal et al., 2023). In such contexts, the inductive biases of standard TTA methods can be harmful. First, limited and biased statistics might lead models to overfit to locally dominant distributions, resulting in negative transfer. Second, commonly adopted tricks like marginal entropy minimization (Wang et al., 2020) also become counter-productive as they force the model to align with a mismatched uniform prior. Consequently, both transductive and online TTA methods

may suffer from performance degradation or even collapse, as also empirically demonstrated in prior works (Zhao et al., 2023; Veilleux et al., 2021). Recent solutions either mitigate sampling bias with memories (Gong et al., 2022) or introduce statistical regularization to stabilize estimates (Zanella et al., 2025). Our MOON aligns with the latter by adopting a KL-anchored PLE framework with dynamic shrinkage.

## 3. Revisiting Test-Time Transduction

**Problem definition.** Consider a batch of $N$ test samples $\{\mathbf{x}_i\}_{i=1}^N$, with the label space consisting of $K$ candidate classes. Let $\theta_v(\cdot)$ and $\theta_t(\cdot)$ denote the visual and textual encoders of a pre-trained VLM, respectively. For each class $k$, we obtain its textual embedding $\mathbf{t}_k = \theta_t(\mathbf{c}_k) \in \mathbb{R}^d$ with a prompt $\mathbf{c}_k$ (e.g., "a photo of a [classname]"). Similarly, the visual feature embedding is extracted as $\mathbf{f}_i = \theta_v(\mathbf{x}_i) \in \mathbb{R}^d$. After $\ell_2$-normalized onto the unit hypersphere $\mathbb{S}^{d-1}$, zero-shot predictions are computed via cosine similarity:

$$\hat{\mathbf{y}}_i = \{\hat{\mathbf{y}}_{i,k}\}_{k=1}^K \in \Delta_K, \quad \hat{\mathbf{y}}_{i,k} = \frac{\exp(\mathbf{f}_i^\top \mathbf{t}_k / \tau)}{\sum_j \exp(\mathbf{f}_i^\top \mathbf{t}_j / \tau)}, \quad (1)$$

where $\Delta$ denotes the probability simplex, and $\tau$ is a fixed temperature coefficient from pre-training.

**Realistic imbalanced settings.** Following StatA (Zanella et al., 2025), we consider two realistic settings: **(i) Batch adaptation:** each batch contains a limited number of effective classes $K_{\text{eff}}$ ($1 \leq K_{\text{eff}} \leq \min\{N, K\}$), where batches are processed independently. **(ii) Online adaptation:** test samples arrive as a non-i.i.d. data stream, whose temporal correlation is controlled by a Dirichlet parameter $\xi$. Here, historical batch information is accessible. For details on the sampling for the imbalanced data, please refer to App. C.4.

**Penalized likelihood estimation.** Recent transductive methods can be broadly conceptualized as a family of soft probabilistic clustering algorithms, which aim to infer latent class assignments and class-conditional distributions over the unlabeled test set. We revisit this process within a penalized likelihood estimation (PLE) formulation that jointly estimates the following variables: **(i) Assignment vectors** $\mathbf{z}_i = \{z_{i,k}\}_{k=1}^K \in \Delta_K$, representing the latent posterior class probability within the probability simplex (initialized from $\hat{\mathbf{y}}_i$). **(ii) Mixture models** $\mathbf{M} = \{\mathbf{M}_k\}_{k=1}^K$, where each component $\mathbf{M}_k$ models the feature distribution of class $k$ with a set of statistical parameters (e.g., mean and covariance). The general optimization objective is given by:

$$\arg\min_{\mathbf{z},\mathbf{M}} \mathcal{L}_{\text{PLE}} = \arg\min_{\mathbf{z},\mathbf{M}} \Big( -\sum_{i=1}^N \mathbf{z}_i^\top \log \mathbf{p}_i + \mathcal{R}(\mathbf{z}) \Big) + \alpha \mathcal{R}(\mathbf{M}).$$
$$(2)$$

Here, the first term represents the standard negative log-likelihood (NLL), with $\mathbf{p}_i$ denoting class-conditional likeli-

hoods under $\mathbf{M}$. The terms $\mathcal{R}(\mathbf{z})$ and $\mathcal{R}(\mathbf{M})$ serve as penalization regularizers for assignments and distribution parameters, respectively. Specifically, $\mathcal{R}(\mathbf{z})$ is typically introduced to mitigate the inherent biases of unsupervised clustering, e.g., by encouraging smoothness or consistency with priors. $\mathcal{R}(\mathbf{M})$ is the key to prevent the model from overfitting to local statistics in realistic class-imbalanced scenarios, where $\alpha$ is a hyperparameter. This is achieved by penalizing the deviation between empirical estimate and the zero-shot prior anchor $\mathbf{M}'$ via a KL divergence: $\mathcal{R}(\mathbf{M}) = \mathrm{KL}(\mathbf{M}'\|\mathbf{M})$.

**Adaptive anchor shrinkage.** The effectiveness of the PLE formulation in realistic class imbalance largely hinges on the KL-based distribution anchor term $\mathcal{R}(\mathbf{M})$. We theoretically demonstrate that KL-anchored PLE naturally yields an *adaptive shrinkage* behavior, where statistics are encouraged to update between empirical estimates and prior anchors. While recent works are implemented under a standard Gaussian assumption by default, this behavior naturally generalizes to the entire exponential family of distributions.

Formally, consider a $K$-class latent-variable mixture model with soft assignments $\{z_{i,k}\}_{i=1}^{N}$ and class-conditional densities from a regular minimal exponential family, i.e.,

$$p(x \mid \eta) = h(x) \exp\big(\eta^{\top} T(x) - A(\eta)\big), \qquad (3)$$

where $\eta$ is the natural parameter, $T(x)$ represents the sufficient statistic, $A(\eta)$ is the log-partition function[3], and $h(x)$ denotes the base measure. Let $n_k = \sum_i z_{i,k}$ and $S_k = \sum_i z_{i,k} T(x_i)$ denote the class-wise soft count and sufficient-statistic sum. Given a fixed anchor distribution $q_k(x) = p(x \mid \eta_k')$ with mean parameter $\mu_k' = \nabla A(\eta_k')$, the KL-anchored PLE optimizes during parameter update:

$$\min_{\{\eta_k\}} -\sum_{i=1}^{N}\sum_{k=1}^{K} z_{i,k} \log p(x_i \mid \eta_k) + \alpha \sum_{k=1}^{K} \mathrm{KL}\big(q_k \| p(\cdot \mid \eta_k)\big), \alpha > 0.$$
$$(4)$$

Then the unique minimizer $\eta_k^{\star}$ satisfies the closed-form shrinkage in the *mean-parameter space*:

$$\nabla A(\eta_k^{\star}) = \frac{S_k + \alpha \mu_k'}{n_k + \alpha} = \beta_k \hat{\mu}_k + (1 - \beta_k)\mu_k', \qquad (5)$$

where $\hat{\mu}_k = \frac{S_k}{n_k}$ is the empirical estimate when $n_k > 0$, $\beta_k = \frac{n_k}{n_k + \alpha} \in [0, 1]$ denotes the shrinkage strength. Eq. (5) reveals that the update is a *data-driven convex combination* between the empirical estimate and the prior anchor, with $\beta_k$, controlled by soft count $n_k$ and anchor weight $\alpha$, adapting automatically to the amount of evidence available for each class: classes with abundant support approach standard maximum likelihood estimation (MLE), while rare classes remain strongly regularized by the anchor.

***Limitation 1: no anchor.*** Notably, in the absence of the anchor term ($\alpha = 0$), we have $\beta_k \equiv 1$, and Eq. (5) degenerates to standard MLE. Under realistic class imbalance,

this causes estimation overfitting to locally biased statistics dominated by majority classes, which can rapidly amplify errors over iterations and lead to catastrophic collapse.

***Limitation 2: bounded but static shrinkage.*** When $\alpha > 0$, Eq. (5) further implies $\|\nabla A(\eta_k^{\star}) - \mu_k'\| = \beta_k \|\hat{\mu}_k - \mu_k'\|$, linearly bounding the deviation by $\beta_k$. In particular, if $n_k = 0$ (outlier classes), the update stays *exactly* at its anchor, i.e., $\nabla A(\eta_k^{\star}) = \mu_k'$, preventing any harmful deviation. However, we could find that these types of anchor-based methods still employ a *static* anchor strength $\alpha$, implicitly assuming equal reliability of statistics across instances and classes. Therefore, their performance and robustness may remain suboptimal (as shown in Fig. 2(a)), which inspires our MOON. Detailed proofs are provided in App. D.

## 4. Our Proposed MOON

Since Eq. (5) applies to the entire exponential family, we propose to adopt a mixture of von Mises-Fisher (vMF) (Gopal & Yang, 2014; Hasnat et al., 2017; Govindarajan et al., 2024) distributions for modeling as normalized VLM embeddings are intrinsically constrained to the unit hypersphere. This distribution is commonly regarded as the natural generalization of Gaussian distribution onto the sphere (Martin et al., 2024). Formally, for a $d$-dimensional unit vector $\mathbf{f}_i \in \mathbb{S}^{d-1}$, the probability density function of a vMF component $\mathcal{V}_k(\boldsymbol{\mu}_k, \kappa_k)$ is defined as:

$$p_{i,k}^{\mathrm{vMF}} = p(\mathbf{f}_i; \boldsymbol{\mu}_k, \kappa_k) \propto \mathcal{C}_d(\kappa_k) \exp(\kappa_k \boldsymbol{\mu}_k^{\top} \mathbf{f}_i), \qquad (6)$$

where $\boldsymbol{\mu}_k \in \mathbb{S}^{d-1}$ denotes the mean direction vector, and $\kappa_k \geq 0$ is a scalar concentration parameter measuring the isotropic precision. $\mathcal{C}_d(\kappa_k) = \frac{\kappa_k^{d/2-1}}{(2\pi)^{d/2} I_{d/2-1}(\kappa_k)}$ is the normalization constant, derived from the order-$\nu$ modified Bessel function of the first kind $I_\nu(\cdot)$. On the basis of this, the KL divergence between $\mathcal{V}_k$ and anchor $\mathcal{V}_k'$ takes: $\mathrm{KL}\left(\mathcal{V}_k' \| \mathcal{V}_k\right) = \log \frac{\mathcal{C}_d(\kappa_k')}{\mathcal{C}_d(\kappa_k)} + \kappa_k' \mathcal{A}_d(\kappa_k') - \kappa_k \mathcal{A}_d(\kappa_k') \boldsymbol{\mu}_k^{\top} \boldsymbol{\mu}_k'$ (details in App. E.2), where $A_d(\kappa) = \frac{I_{d/2}(\kappa)}{I_{d/2-1}(\kappa)}$ represents the Bessel function ratio. We initialize the anchor distribution $\mathcal{V}_k'$ with zero-shot priors leveraged from text:

$$\boldsymbol{\mu}_k' = \mathbf{t}_k, \quad \mathcal{A}_d(\boldsymbol{\kappa}_k') = 1 - \frac{\sum_i \mathbf{z}_{i,k} \|\mathbf{f}_i - \boldsymbol{\mu}_k'\|^2}{2\sum_i \mathbf{z}_{i,k}}. \qquad (7)$$

The derivation of $\mathcal{A}_d(\boldsymbol{\kappa}_k')$ is provided in App. F. Building upon the above formulation, we arrive at the final objective:

$$\mathcal{L}_{\mathrm{PLE}}(\mathbf{z}; \boldsymbol{\mu}, \boldsymbol{\kappa}) = \boldsymbol{\gamma} \left( -\sum_{i=1}^{N} \mathbf{z}_i^{\top} \log(\mathbf{p}_i^{\mathrm{vMF}}) + \mathcal{R}(\mathbf{z}) \right)$$
$$+ \boldsymbol{\alpha} \sum_{k=1}^{K} \mathrm{KL}\left(\mathcal{V}_k' \| \mathcal{V}_k\right), \qquad (8)$$

$$\text{where} \quad \mathcal{R}(\mathbf{z}) = -\sum_{i,j} \omega_{ij} \mathbf{z}_i^{\top} \mathbf{z}_j + \sum_{i=1}^{N} \mathrm{KL}(\mathbf{z}_i \| \hat{\mathbf{y}}_i).$$

For $\mathcal{R}(\mathbf{z})$, we choose a widely-adopted combination of a Laplacian regularizer and a text-supervision term (Zanella

---

[3]This is different from the Bessel function ratio $\mathcal{A}_d(\cdot)$ below.

et al., 2024; 2025), where $\omega_{ij} = \mathbf{f}_i^\top \mathbf{f}_j$ denotes feature affinity. The former performs label propagation among nearby samples to encourage smooth assignments, while the latter penalizes deviations from zero-shot predictions. Moreover, we introduce two weights at instance and class level, $\boldsymbol{\gamma}$ and $\boldsymbol{\alpha}$, enabling dynamic adjustment of shrinkage strength. Different from Eq. (2), both weights are driven by priors.

## 4.1. Dynamic Shrinkage for Realistic Class Imbalance

**Instance-level adjustment.** The first two terms in Eq. (8) actually form a standard MLE objective, which typically treats all test samples equally. However, in realistic scenarios, certain zero-shot predictions may be inherently noisy; estimation biases may also accumulate and propagate over iterations. A natural idea is to employ predictive entropy as a metric for reliability, as it is widely adopted in TTA for model optimization or memory updates (Wang et al., 2020; Karmanov et al., 2024). Therefore, we introduce an entropy-based weight $\gamma_i \in [0, 1]$ to dynamically re-weight the contribution of each sample to the MLE objective:

$$\gamma_i = 1 - \frac{H(\hat{\mathbf{y}}_i)}{\log K}, \qquad (9)$$

where $H(\hat{\mathbf{y}}_i) = -\sum_{k=1}^K \hat{y}_{i,k} \log \hat{y}_{i,k}$ denotes entropy, and $\log K$ serves as the normalization factor. Through this mechanism, certain predictions are encouraged, while uncertain or ambiguous ones are suppressed. As explicitly shown in Eq. (14), it serves as a coefficient for assignments $\mathbf{z}_i$ and filters out unreliable samples during the parameter estimation of $\boldsymbol{\mu}_k$ and $\kappa_k$. In implementation, we update $\gamma_i$ with current assignments $\mathbf{z}_i$ for stability, i.e., $\gamma_i = 1 - \frac{H(\mathbf{z}_i)}{\log K}$.

**Class-level adjustment.** Class imbalance inherently induces class sparsity, such as $K_{\text{eff}} \ll K$ or $\xi \to 0$. More importantly, it's impossible to identify which classes are *effective* (i.e., present) within batch, as labels are unavailable at test time. This makes transduction particularly vulnerable to negative transfer from *outlier* (i.e., absent) classes. Although the distribution anchor in Eq. (8) alleviates this issue by inducing an adaptive shrinkage behavior, it treats all classes equally and lacks dynamic, fine-grained shrinkage strength modeling. Moreover, it operates with a single scalar hyperparameter $\alpha$, which requires task-specific tuning.

To address this limitation, Partial Domain Adaptation (PDA) (Cao et al., 2018) has provided successful experiences that we can learn from. PDA studies settings in which the target label space forms a subset of the source label space. This also works for our test-time settings, as the effective class set of a given test batch is also a subset of VLM's predefined candidate class set. Consequently, the absence of classes can be interpreted as a form of source-target label space mismatch, which should be suppressed during adaptation. In PDA, such mismatch is typically quantified with zero-shot

prediction confidence, since classes with higher confidence are more likely and frequent to be present in the target domain. Leveraging this insight, we replace the fixed scalar $\alpha$ with class-level dynamic weights $\boldsymbol{\alpha} = \{\alpha_k\}_{k=1}^K$.

Intuitively, highly confident classes should encourage the parameters $\boldsymbol{\mu}_k$ and $\kappa_k$ to align more closely to empirical estimates, pushing the strength $\beta_k = \frac{n_k}{n_k + \alpha_k}$ towards 1. Therefore, $\alpha_k$ should be negatively correlated with class confidence. We define $\alpha_k$ to be inversely related to confidence, as this form is widely used in statistical learning to impose regularization on less reliable signals (Zou, 2006):

$$\alpha_k = \frac{1}{\lambda_k}, \qquad (10)$$

where $\lambda_k$ denotes the $k$-th class confidence derived from zero-shot priors. PDA methods often directly estimate $\lambda_k$ from average confidence. In test-time settings, however, such a design is insufficient, as the effective label set varies across batches. On the one hand, those rare but effective classes, occurring infrequently yet consistently in the data streams, might be confused with truly outlier classes and instead suppress positive transfer. On the other hand, we don't want to lose the generality under a distribution closer to uniform. For balance, $\lambda_k$ is defined as the geometric mean of the average and the maximum confidence:

$$\lambda_k = \sqrt{\frac{1}{N} \sum_{i=1}^N \hat{\mathbf{y}}_{i,k} \odot \max_i \hat{\mathbf{y}}_{i,k}}. \qquad (11)$$

This mildly sacrifices accuracy under severe class imbalance, but yields a more general solution across broader scenarios.

## 4.2. Optimization Algorithm

Since the proposed PLE objective jointly involves the assignments $\mathbf{z}$ and mixture parameters $\{\mathcal{V}_k(\boldsymbol{\mu}_k, \kappa_k)\}_{k=1}^K$, we adopt an efficient optimization algorithm following recent works (Zanella et al., 2024). The algorithm follows the Block Successive Minimization (BSUM) framework (Razaviyayn et al., 2013), which alternately updates two blocks of variables via iterative block-coordinate descent on surrogate objectives. This algorithm is also theoretically guaranteed to converge, as shown in App. A. Given that both $\boldsymbol{\gamma}$ and $\boldsymbol{\alpha}$ are non-negative, all terms in Eq. (8) except the Laplacian regularizer are convex with respect to each block.

**Linear approximation w.r.t assignments z.** Due to the presence of concave Laplacian regularizer $\sum_{i,j} \omega_{ij} \mathbf{z}_i^\top \mathbf{z}_j$, a closed-form update for $\mathbf{z}$ cannot be obtained directly. Therefore, we construct a linear upper bound by replacing this term with its first-order Taylor expansion at current iteration, i.e., $-\sum_i \mathbf{z}_i^\top \left(\sum_j \omega_{ij} \mathbf{z}_j^{(t)}\right)$, where $\mathbf{z}_j^{(t)}$ denotes the assignment obtained at iteration $t$. By minimizing the constructed approximate surrogate objective, we have:

$$\mathbf{z}_i^{(t+1)} = \frac{\hat{\mathbf{y}}_i \odot \exp(\log \mathbf{p}_i^{\text{vMF}} + \sum_j \omega_{ij} \mathbf{z}_j^{(t)})}{(\hat{\mathbf{y}}_i \odot \exp(\log \mathbf{p}_i^{\text{vMF}} + \sum_j \omega_{ij} \mathbf{z}_j^{(t)}))^\top \mathbb{1}_K}. \qquad (12)$$

Detailed derivations are provided in App. G.1. In implementation, we omit the inner-loop optimization required in previous works and perform a single pass per iteration for efficiency. Note that $\gamma$ does not appear in Eq. (12) as it could be canceled out during the derivation of $\mathbf{z}_i$.

**Closed-form update w.r.t parameters $\mu$ and $\kappa$.** When fixing $\mathbf{z}$, Eq. (8) becomes strictly convex with respect to the mixture parameters $\mu$ and $\kappa$. Therefore, we can derive closed-form updates by setting partial derivatives to zero:

$$\mu_k = \frac{\sum_i \gamma_i \mathbf{z}_{i,k} \mathbf{f}_i + \alpha_k \mathcal{A}_d(\kappa'_k)\mu'_k}{\|\sum_i \gamma_i \mathbf{z}_{i,k} \mathbf{f}_i + \alpha_k \mathcal{A}_d(\kappa'_k)\mu'_k\|},$$
$$\mathcal{A}_d(\kappa_k) = \frac{\|\sum_i \gamma_i \mathbf{z}_{i,k} \mathbf{f}_i + \alpha_k \mathcal{A}_d(\kappa'_k)\mu'_k\|}{\sum_i \gamma_i \mathbf{z}_{i,k} + \alpha_k}. \quad (13)$$

As shown in Sec. 3, we can rewrite the above updates in a more intuitive form. Under the mild assumption $\mathcal{A}_d(\kappa'_k) \approx 1$, Eq. (13) is equivalent to (proof in App. H):

$$\mu_k = \frac{\beta_k \mathbf{v}_k + (1-\beta_k)\mu'_k}{\|\beta_k \mathbf{v}_k + (1-\beta_k)\mu'_k\|}, \quad \mathcal{A}_d(\kappa_k) = \|\beta_k \mathbf{v}_k + (1-\beta_k)\mu'_k\|, \quad (14)$$

where $\mathbf{v}_k = \frac{\sum_{i=1}^N \gamma_{i,k}\mathbf{z}_{i,k}\mathbf{f}_i}{\sum_{i=1}^N \gamma_{i,k}\mathbf{z}_{i,k}}$ and $\beta_k = \frac{\sum_{i=1}^N \gamma_{i,k}\mathbf{z}_{i,k}}{\sum_{i=1}^N \gamma_{i,k}\mathbf{z}_{i,k}+\alpha_k}$. This offers an intuitive interpretation of the anchor shrinkage as in Eq. (5). Here, $\mathbf{v}_k$ represents empirical estimates from standard MLE, and $\mu'_k$ serves as prior anchor. Our proposed adjustments are seamlessly integrated here: $n_k = \sum_i \gamma_i \mathbf{z}_{i,k}$ replaces soft count $\sum_i \mathbf{z}_{i,k}$, ensuring that noisy predictions are suppressed. $\alpha_k$ further penalizes harmful deviations: for effective classes, $\alpha_k \to 1$ while $n_k$ increases, allowing the model to learn more from data; for outlier classes, $\alpha_k$ dominates $\beta_k$, forcing the updates to shrink towards the anchor, thereby mitigating negative transfer.

**Overall procedure.** The overall procedure of MOON is summarized in App. A. The initializations and updates mentioned above directly yield the Bessel function ratio $\mathcal{A}_d(\kappa_k)$, which corresponds to the mean resultant length of the vMF distribution $\bar{r}_k$. We then employ the well-known approximation (Banerjee et al., 2005) to estimate $\kappa_k$:

$$\kappa_k \approx \frac{d\bar{r}_k - \bar{r}_k^3}{1 - \bar{r}_k^2}, \qquad \bar{r}_k \triangleq \mathcal{A}_d(\kappa_k). \quad (15)$$

Note that the parameter estimation of vMF mixtures is simpler and more computationally efficient than GMMs, as it uses fewer parameters and avoids the expensive quadratic forms and inversions of $\mathcal{R}^{d\times d}$ covariance matrices.

# 5. Experiments

We evaluate our method in several scenarios under two realistic settings, as defined in Sec. 3. We report the Top-1 accuracy across 11 public fine-grained classification datasets, and adopt CLIP ViT-B/16 as our default VLM backbone.

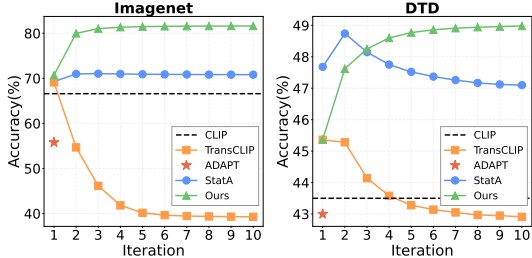

*Figure 3.* **Convergence analysis on ImageNet and DTD.** We demonstrate performance curves over iterations for each method.

Please see App. C for details on datasets, baselines, prompt templates, and other experimental specifics.

## 5.1. Main Results

**Batch adaptation.** We first report the results under batch adaptation in Tab. 1(a) and (b), with batch sizes of 64 and 1,000, respectively. The results show that existing transductive methods generally suffer from severe performance degradation under realistic class-imbalance, and most of them even collapse and underperform zero-shot CLIP. While StatA mitigates this issue by introducing anchor term $\mathcal{R}(\mathbf{M})$, its performance remains suboptimal. In contrast, our MOON consistently achieves the best average performance across all scenarios, effectively enhancing VLM predictions. Notably, the performance gains of MOON become more pronounced as class imbalance becomes more severe ($\frac{K_{\text{eff}}}{\min(N,K)}$ decreases). Moreover, MOON delivers the most significant gains on challenging large-scale datasets such as ImageNet, highlighting its superiority in practical applications.

**Online adaptation.** We further evaluate methods under online adaptation. As shown in Tab. 2, most online TTA methods remain relatively stable across different correlation strengths, without exhibiting performance degradation. Nevertheless, MOON still achieves state-of-the-art performance generally. We observe a slight drop only in the *Low* scenario, where the class distribution becomes closer to uniform. This can be attributed to the inherent bias of our $\alpha$, as it is designed to favor sparse effective class sets. Similarly, MOON brings better improvements in scenarios with stronger correlations. For example, MOON outperforms StatA by 10.7% on ImageNet in the *Separate* scenario.

## 5.2. Efficiency Analysis

**Runtime.** Tab. 3 reports the runtime per batch on the ImageNet dataset. We observe that the CLIP inference, including both visual and textual encoding, dominates the total computational cost. Considering the net runtime of methods, our MOON highlights its exceptional efficiency. Despite requiring iterative optimization, MOON is still twice as fast as the single-pass ADAPT. Moreover, this efficiency advantage becomes increasingly pronounced as the batch size

*Table 1*. **Main results for batch adaptation**, averaged over 1,000 runs. The best and second-best results are marked in **bold** and underlined, respectively. We report three scenarios for each batch size 64 and 1,000, with varying range of effective classes $K_{\text{eff}}$. Subscript green indicates improvement, red indicates decline, and gray indicates no change compared with zero-shot performance.

(a) Setting where the batch size is 64: Very Low (1–4 $K_{\text{eff}}$), Low (2–10), and Medium (5–25).

| $K_{\text{eff}}$ | Method | ImageNet | SUN397 | Aircraft | EuroSAT | StanfordCars | Food101 | Pets | Flowers102 | Caltech101 | DTD | UCF101 | Avg. |
|---|---|---|---|---|---|---|---|---|---|---|---|---|---|
| | CLIP | 66.6 | 62.5 | 24.7 | 48.3 | 65.6 | 85.9 | 89.1 | 70.7 | 93.2 | 43.5 | 67.5 | 65.2 |
| | MTA | $69.3_{+2.7}$ | $64.8_{+2.3}$ | $27.4_{+2.7}$ | $46.9_{-1.4}$ | $68.0_{+2.4}$ | $87.2_{+1.3}$ | $89.4_{+0.3}$ | $71.7_{+1.0}$ | $94.0_{+0.8}$ | $44.4_{+0.9}$ | $69.0_{+1.5}$ | $66.6_{+1.3}$ |
| Very Low (1–4) | Dirichlet | $79.2_{+12.6}$ | $75.7_{+13.2}$ | $28.2_{+3.5}$ | $47.2_{-1.1}$ | $68.2_{+2.6}$ | $88.1_{+2.2}$ | $87.5_{-1.6}$ | $71.2_{+0.5}$ | $88.8_{-4.4}$ | $50.3_{+6.8}$ | $69.0_{+1.5}$ | $68.5_{+3.3}$ |
| | ZLaP | $14.5_{-52.1}$ | $13.0_{-49.5}$ | $8.4_{-16.3}$ | $36.6_{-11.7}$ | $23.7_{-41.9}$ | $31.9_{-54.0}$ | $57.0_{-32.1}$ | $22.4_{-48.3}$ | $52.4_{-40.8}$ | $13.0_{-30.5}$ | $29.2_{-38.3}$ | $27.5_{-37.8}$ |
| | GDA-CLIP | $20.5_{-46.1}$ | $19.8_{-42.7}$ | $10.3_{-14.4}$ | $39.9_{-8.4}$ | $28.2_{-37.4}$ | $51.4_{-34.5}$ | $60.4_{-28.7}$ | $29.6_{-41.1}$ | $52.9_{-40.3}$ | $39.9_{-3.6}$ | $35.2_{-32.3}$ | $35.3_{-30.0}$ |
| | TransCLIP | $21.6_{-45.0}$ | $21.1_{-41.4}$ | $11.6_{-13.1}$ | $45.1_{-3.2}$ | $34.7_{-30.9}$ | $59.2_{-26.7}$ | $72.4_{-16.7}$ | $36.4_{-34.3}$ | $62.3_{-30.9}$ | $26.1_{-17.4}$ | $37.7_{-29.8}$ | $38.9_{-26.3}$ |
| | ADAPT | $60.8_{-5.8}$ | $56.0_{-6.5}$ | $21.4_{-3.3}$ | $45.9_{-2.4}$ | $59.7_{-5.9}$ | $81.4_{-4.5}$ | $84.7_{-4.4}$ | $66.8_{-3.9}$ | $90.4_{-2.8}$ | $39.2_{-4.3}$ | $61.4_{-6.1}$ | $60.7_{-4.5}$ |
| | StatA | $72.9_{+6.3}$ | $66.0_{+3.5}$ | $29.3_{+4.6}$ | $56.8_{+8.5}$ | $76.2_{+10.6}$ | $90.3_{+4.4}$ | $95.5_{+6.4}$ | $77.6_{+6.9}$ | $93.0_{-0.2}$ | $46.1_{+2.6}$ | $70.2_{+2.7}$ | $70.4_{+5.1}$ |
| | MOON | $82.8_{+16.2}$ | $77.2_{+14.7}$ | $32.0_{+7.3}$ | $53.7_{+5.4}$ | $78.6_{+13.0}$ | $96.4_{+10.5}$ | $96.0_{+6.9}$ | $77.7_{+7.0}$ | $94.9_{+1.7}$ | $55.6_{+12.1}$ | $75.7_{+8.2}$ | **$74.6_{+9.4}$** |
| Low (2–10) | Dirichlet | $80.1_{+13.5}$ | $78.0_{+15.5}$ | $28.1_{+3.4}$ | $43.5_{-4.8}$ | $71.5_{+5.9}$ | $92.3_{+6.4}$ | $92.7_{+3.6}$ | $74.7_{+4.0}$ | $93.0_{-0.2}$ | $48.9_{+5.4}$ | $70.9_{+3.4}$ | $70.3_{+5.1}$ |
| | ZLaP | $19.1_{-47.5}$ | $19.0_{-43.5}$ | $12.0_{-12.7}$ | $46.4_{-1.9}$ | $27.9_{-37.7}$ | $43.5_{-42.4}$ | $66.6_{-22.5}$ | $31.3_{-39.4}$ | $60.8_{-32.4}$ | $22.4_{-21.1}$ | $38.7_{-28.8}$ | $35.2_{-30.0}$ |
| | GDA-CLIP | $18.6_{-48.0}$ | $19.4_{-43.1}$ | $12.6_{-12.1}$ | $50.3_{+2.0}$ | $25.2_{-40.4}$ | $49.7_{-36.2}$ | $60.4_{-28.7}$ | $33.1_{-37.6}$ | $55.2_{-38.0}$ | $28.4_{-15.1}$ | $37.2_{-30.3}$ | $35.5_{-29.8}$ |
| | TransCLIP | $20.3_{-46.3}$ | $22.4_{-40.1}$ | $14.3_{-10.4}$ | $53.9_{+5.6}$ | $30.8_{-34.8}$ | $55.6_{-30.3}$ | $69.4_{-19.7}$ | $40.9_{-29.8}$ | $64.6_{-28.6}$ | $31.6_{-11.9}$ | $40.9_{-26.6}$ | $40.4_{-24.8}$ |
| | ADAPT | $65.3_{-1.3}$ | $60.7_{-1.8}$ | $23.5_{-1.2}$ | $51.5_{+3.2}$ | $62.3_{-3.3}$ | $84.5_{-1.4}$ | $87.5_{-1.6}$ | $68.8_{-1.9}$ | $91.0_{-2.2}$ | $41.8_{-1.7}$ | $63.8_{-3.7}$ | $63.7_{-1.5}$ |
| | StatA | $72.8_{+6.2}$ | $66.9_{+4.4}$ | $27.7_{+3.0}$ | $51.3_{+3.0}$ | $73.5_{+7.9}$ | $89.5_{+3.6}$ | $93.7_{+4.6}$ | $76.6_{+5.9}$ | $93.6_{+0.4}$ | $46.9_{+3.4}$ | $69.6_{+2.1}$ | $69.3_{+4.1}$ |
| | MOON | $83.8_{+17.2}$ | $78.0_{+15.5}$ | $29.8_{+5.1}$ | $48.3_{\pm0.0}$ | $76.7_{+11.1}$ | $95.5_{+9.6}$ | $94.6_{+5.5}$ | $77.3_{+6.6}$ | $95.3_{+2.1}$ | $50.9_{+7.4}$ | $74.2_{+6.7}$ | **$73.1_{+7.9}$** |
| Medium (5–25) | Dirichlet | $77.7_{+11.1}$ | $72.9_{+10.4}$ | $26.1_{+1.4}$ | $38.6_{-9.7}$ | $71.6_{+6.0}$ | $90.8_{+4.9}$ | $88.4_{-0.7}$ | $71.5_{+0.8}$ | $93.7_{+0.5}$ | $42.9_{-0.6}$ | $67.8_{+0.3}$ | $67.5_{+2.2}$ |
| | ZLaP | $29.0_{-37.6}$ | $27.9_{-34.6}$ | $16.5_{-8.2}$ | $49.0_{+0.7}$ | $36.0_{-29.6}$ | $59.1_{-26.8}$ | $76.4_{-12.7}$ | $42.9_{-27.8}$ | $72.0_{-21.2}$ | $32.0_{-11.5}$ | $50.3_{-17.2}$ | $44.7_{-20.6}$ |
| | GDA-CLIP | $19.2_{-47.4}$ | $21.4_{-41.1}$ | $15.8_{-8.9}$ | $56.2_{+7.9}$ | $27.4_{-38.2}$ | $52.9_{-33.0}$ | $68.7_{-20.4}$ | $40.1_{-30.6}$ | $59.2_{-34.0}$ | $35.2_{-8.3}$ | $43.7_{-23.8}$ | $40.0_{-25.3}$ |
| | TransCLIP | $15.5_{-51.1}$ | $22.8_{-39.7}$ | $17.0_{-7.7}$ | $58.2_{+9.9}$ | $32.9_{-32.7}$ | $56.3_{-29.6}$ | $72.6_{-16.5}$ | $45.0_{-25.7}$ | $65.6_{-27.6}$ | $37.5_{-6.0}$ | $46.5_{-21.0}$ | $42.7_{-22.5}$ |
| | ADAPT | $66.8_{+0.2}$ | $61.7_{-0.8}$ | $25.0_{+0.3}$ | $52.8_{+4.5}$ | $65.4_{-0.2}$ | $85.9_{\pm0.0}$ | $88.7_{-0.4}$ | $69.7_{-1.0}$ | $92.2_{-1.0}$ | $43.5_{\pm0.0}$ | $66.7_{-0.8}$ | $65.3_{+0.1}$ |
| | StatA | $70.7_{+4.1}$ | $65.3_{+2.8}$ | $26.0_{+1.3}$ | $45.0_{-3.3}$ | $71.1_{+5.5}$ | $88.2_{+2.3}$ | $90.8_{+1.7}$ | $73.7_{+3.0}$ | $93.9_{+0.7}$ | $47.5_{+4.0}$ | $69.1_{+1.6}$ | $67.4_{+2.2}$ |
| | MOON | $79.5_{+12.9}$ | $72.6_{+10.1}$ | $26.0_{+1.3}$ | $42.9_{-5.4}$ | $73.6_{+8.0}$ | $92.5_{+6.6}$ | $90.7_{+1.6}$ | $74.4_{+3.7}$ | $94.6_{+1.4}$ | $44.7_{+1.2}$ | $71.5_{+4.0}$ | **$69.4_{+4.1}$** |

(b) Setting where the batch size is 1,000: Medium (5–25 $K_{\text{eff}}$), High (25–50), and Very High (50-100).

| $K_{\text{eff}}$ | Method | ImageNet | SUN397 | Aircraft | EuroSAT | StanfordCars | Food101 | Pets | Flowers102 | Caltech101 | DTD | UCF101 | Avg. |
|---|---|---|---|---|---|---|---|---|---|---|---|---|---|
| | CLIP | 66.6 | 62.5 | 24.7 | 48.3 | 65.6 | 85.9 | 89.1 | 70.7 | 93.2 | 43.5 | 67.5 | 65.2 |
| | MTA | $69.3_{+2.7}$ | $64.8_{+2.3}$ | $27.4_{+2.7}$ | $46.9_{-1.4}$ | $68.0_{+2.4}$ | $87.2_{+1.3}$ | $89.4_{+0.3}$ | $71.7_{+1.0}$ | $94.0_{+0.8}$ | $44.4_{+0.9}$ | $69.0_{+1.5}$ | $66.6_{+1.3}$ |
| Medium (5–25) | Dirichlet | $60.9_{-5.7}$ | $75.4_{+12.9}$ | $26.7_{+2.0}$ | $38.8_{-9.5}$ | $74.1_{+8.5}$ | $76.2_{-9.7}$ | $91.0_{+1.9}$ | $71.6_{+0.9}$ | $92.4_{-0.8}$ | $36.2_{-7.3}$ | $65.4_{-2.1}$ | $64.4_{-0.8}$ |
| | ZLaP | $16.6_{-50.0}$ | $20.1_{-42.4}$ | $16.4_{-8.3}$ | $49.0_{+0.7}$ | $32.2_{-33.4}$ | $55.5_{-30.4}$ | $76.4_{-12.7}$ | $40.6_{-30.1}$ | $67.7_{-25.5}$ | $34.2_{-9.3}$ | $48.1_{-19.4}$ | $41.5_{-23.7}$ |
| | GDA-CLIP | $32.3_{-34.3}$ | $32.5_{-30.0}$ | $19.0_{-5.7}$ | $61.0_{+12.7}$ | $39.5_{-26.1}$ | $68.1_{-17.8}$ | $79.6_{-9.5}$ | $50.3_{-20.4}$ | $69.1_{-24.1}$ | $39.2_{-4.3}$ | $49.5_{-18.0}$ | $49.1_{-16.1}$ |
| | TransCLIP | $39.9_{-26.7}$ | $42.7_{-19.8}$ | $22.0_{-2.7}$ | $63.1_{+14.8}$ | $49.9_{-15.7}$ | $80.6_{-5.3}$ | $87.9_{-1.2}$ | $58.7_{-12.0}$ | $79.1_{-14.1}$ | $42.9_{-0.6}$ | $55.0_{-12.5}$ | $56.5_{-8.7}$ |
| | ADAPT | $55.8_{-10.8}$ | $52.8_{-9.7}$ | $23.3_{-1.4}$ | $63.5_{+15.2}$ | $55.9_{-9.7}$ | $77.4_{-8.5}$ | $85.2_{-3.9}$ | $67.1_{-3.6}$ | $89.0_{-4.2}$ | $43.0_{-0.5}$ | $62.3_{-5.2}$ | $61.4_{-3.9}$ |
| | StatA | $70.8_{+4.2}$ | $64.5_{+2.0}$ | $28.4_{+3.7}$ | $60.4_{+12.1}$ | $74.0_{+8.4}$ | $87.5_{+1.6}$ | $93.1_{+4.0}$ | $77.5_{+6.8}$ | $92.8_{-0.4}$ | $47.1_{+3.6}$ | $70.2_{+2.7}$ | $69.7_{+4.4}$ |
| | MOON | $81.6_{+15.0}$ | $75.5_{+13.0}$ | $29.6_{+4.9}$ | $58.9_{+10.6}$ | $76.1_{+10.5}$ | $93.1_{+7.2}$ | $92.4_{+3.3}$ | $77.3_{+6.6}$ | $94.9_{+1.7}$ | $49.0_{+5.5}$ | $73.6_{+6.1}$ | **$72.9_{+7.7}$** |
| High (25–50) | Dirichlet | $17.3_{-49.3}$ | $37.3_{-25.2}$ | $21.0_{-3.7}$ | $37.9_{-10.4}$ | $65.4_{-0.2}$ | $46.3_{-39.6}$ | $81.3_{-7.8}$ | $46.3_{-24.4}$ | $80.5_{-12.7}$ | $21.1_{-22.4}$ | $43.6_{-23.9}$ | $45.3_{-20.0}$ |
| | ZLaP | $23.8_{-42.8}$ | $32.2_{-30.3}$ | $22.2_{-2.5}$ | $49.3_{+1.0}$ | $45.4_{-20.2}$ | $74.9_{-11.0}$ | $86.5_{-2.6}$ | $56.2_{-14.5}$ | $79.7_{-13.5}$ | $43.6_{+0.1}$ | $60.8_{-6.7}$ | $52.2_{-13.0}$ |
| | GDA-CLIP | $37.8_{-28.8}$ | $41.5_{-21.0}$ | $23.8_{-0.9}$ | $62.4_{+14.1}$ | $49.2_{-16.4}$ | $76.1_{-9.8}$ | $90.9_{+1.8}$ | $64.7_{-6.0}$ | $77.9_{-15.3}$ | $47.4_{+3.9}$ | $61.7_{-5.8}$ | $57.6_{-7.6}$ |
| | TransCLIP | $43.9_{-22.7}$ | $49.6_{-12.9}$ | $24.8_{+0.1}$ | $64.0_{+15.7}$ | $57.3_{-8.3}$ | $83.0_{-2.9}$ | $91.4_{+2.3}$ | $69.1_{-1.6}$ | $85.5_{-7.7}$ | $47.5_{+4.0}$ | $65.4_{-2.1}$ | $62.0_{-3.3}$ |
| | ADAPT | $60.4_{-6.2}$ | $57.9_{-4.6}$ | $24.9_{+0.2}$ | $64.0_{+15.7}$ | $60.7_{-4.9}$ | $82.1_{-3.8}$ | $91.8_{+2.7}$ | $70.2_{-0.5}$ | $90.2_{-3.0}$ | $46.9_{+3.4}$ | $67.0_{-0.5}$ | $65.1_{-0.1}$ |
| | StatA | $71.9_{+5.3}$ | $66.4_{+3.9}$ | $25.9_{+1.2}$ | $60.7_{+12.4}$ | $73.6_{+8.0}$ | $88.0_{+2.1}$ | $91.4_{+2.3}$ | $76.7_{+6.0}$ | $93.2_{\pm0.0}$ | $47.9_{+4.4}$ | $71.5_{+4.0}$ | $69.8_{+4.5}$ |
| | MOON | $81.8_{+15.2}$ | $74.9_{+12.4}$ | $25.5_{+0.8}$ | $59.3_{+11.0}$ | $74.7_{+9.1}$ | $91.0_{+5.1}$ | $89.7_{+0.6}$ | $75.2_{+4.5}$ | $94.8_{+1.6}$ | $45.1_{+1.6}$ | $71.8_{+4.3}$ | **$71.3_{+6.0}$** |
| Very High (50–100) | Dirichlet | $10.8_{-55.8}$ | $15.7_{-46.8}$ | $17.5_{-7.2}$ | $37.8_{-10.5}$ | $51.2_{-14.4}$ | $29.1_{-56.8}$ | $79.3_{-9.8}$ | $24.3_{-46.4}$ | $59.1_{-34.1}$ | $19.0_{-24.5}$ | $26.1_{-41.4}$ | $33.6_{-31.6}$ |
| | ZLaP | $32.7_{-33.9}$ | $44.0_{-18.5}$ | $25.4_{+0.7}$ | $49.3_{+1.0}$ | $55.2_{-10.4}$ | $83.3_{-2.6}$ | $87.3_{-1.8}$ | $64.8_{-5.9}$ | $87.9_{-5.3}$ | $45.2_{+1.7}$ | $67.8_{+0.3}$ | $58.4_{-6.8}$ |
| | GDA-CLIP | $41.7_{-24.9}$ | $48.5_{-14.0}$ | $26.3_{+1.6}$ | $62.4_{+14.1}$ | $56.5_{-9.1}$ | $82.8_{-3.1}$ | $92.3_{+3.2}$ | $73.1_{+2.4}$ | $86.5_{-6.7}$ | $49.2_{+5.7}$ | $70.5_{+3.0}$ | $62.7_{-2.5}$ |
| | TransCLIP | $44.5_{-22.1}$ | $53.0_{-9.5}$ | $25.6_{+0.9}$ | $64.1_{+15.8}$ | $60.9_{-4.7}$ | $85.2_{-0.7}$ | $91.9_{+2.8}$ | $74.3_{+3.6}$ | $90.5_{-2.7}$ | $48.1_{+4.6}$ | $70.7_{+3.2}$ | $64.4_{-0.8}$ |
| | ADAPT | $64.7_{-1.9}$ | $61.9_{-0.6}$ | $25.8_{+1.1}$ | $64.0_{+15.7}$ | $64.9_{-0.7}$ | $85.5_{-0.4}$ | $92.6_{+3.5}$ | $73.2_{+2.5}$ | $92.5_{-0.7}$ | $48.0_{+4.5}$ | $70.3_{-2.8}$ | $67.6_{+2.3}$ |
| | StatA | $71.8_{+5.2}$ | $67.1_{+4.6}$ | $23.9_{-0.8}$ | $60.7_{+12.4}$ | $70.2_{+4.6}$ | $87.1_{+1.2}$ | $91.1_{+2.0}$ | $74.3_{+3.6}$ | $93.7_{+0.5}$ | $48.0_{+4.5}$ | $70.7_{+3.2}$ | $69.0_{+3.7}$ |
| | MOON | $80.4_{+13.8}$ | $71.9_{+9.4}$ | $23.4_{-1.3}$ | $59.3_{+11.0}$ | $70.7_{+5.1}$ | $87.8_{+1.9}$ | $89.3_{+0.2}$ | $72.3_{+1.6}$ | $93.3_{+0.1}$ | $44.3_{+0.8}$ | $68.9_{+1.4}$ | **$69.2_{+4.0}$** |

grows. This empirically validates a significant advantage of vMF mixtures that has been largely overlooked in prior works. We further provide a complexity analysis in App. A.

**Convergence.** Fig. 3 illustrates the convergence curves of four transductive methods on ImageNet and DTD. We find that our MOON converges rapidly within just a few iterations, maintaining stable and consistent improvements. In contrast, methods without anchoring mechanism (e.g., TransCLIP) tend to deviate from global, reliable estimates as the iteration proceeds, leading to collapse. Notably, even a single iteration could yield competitive performance.

## 5.3. Ablation Studies

**Components.** We study the effect of key components in Tab. 4. Obviously, iteratively updating parameters $(\boldsymbol{\mu}, \boldsymbol{\kappa})$

improves performance, as it allows the class-conditional distributions to progressively adapt to the underlying latent structure of data. By dynamically adjusting shrinkage strength, our $\boldsymbol{\alpha}$ robustly mitigates negative transfer from outlier classes, and its benefit scales up as the scenario becomes more imbalanced. Surprisingly, the effect of $\gamma$ is marginal when batch size is small. A possible explanation is that the estimation error is dominated by high statistical variance and class sparsity in small batches; whereas in large batches, sufficient dense samples provide enough stable statistics for $\gamma$ to filter out noise. Overall, simply incorporating zero-shot priors for adjustments enables more fine-grained shrinkage in realistic scenarios, with negligible additional cost.

**Design choices of $\alpha$ and $\lambda$.** We first analyze the design of the anchor weight $\alpha_k$ in Eq. (10). As shown in Tab. 5, directly setting $\alpha_k = \lambda_k$ performs poorly, since it assigns

*Table 2.* **Main results for online adaptation**, averaged over 100 runs. The best and second-best results are marked in **bold** and underlined, respectively. We report four scenarios for batch size 128, with different Dirichlet parameter $\xi$. Separate denotes sequential classes. Subscript green indicates improvement, red indicates decline, and gray indicates no change compared with zero-shot performance.

| Scenario | Method | ImageNet | SUN397 | Aircraft | EuroSAT | StanfordCars | Food101 | Pets | Flowers102 | Caltech101 | DTD | UCF101 | Avg. |
|---|---|---|---|---|---|---|---|---|---|---|---|---|
| | CLIP | 66.6 | 62.5 | 24.7 | 48.3 | 65.6 | 85.9 | 89.1 | 70.7 | 93.2 | 43.5 | 67.5 | 65.2 |
| | MTA | $69.3_{+2.7}$ | $64.8_{+2.3}$ | $27.4_{+2.7}$ | $46.9_{-1.4}$ | $68.0_{+2.4}$ | $87.2_{+1.3}$ | $89.4_{+0.3}$ | $71.7_{+1.0}$ | $94.0_{+0.8}$ | $44.4_{+0.9}$ | $69.0_{+1.5}$ | $66.6_{+1.3}$ |
| Low ($\xi = 0.1$) | TENT | $66.6_{\pm0.0}$ | $64.5_{+2.0}$ | $24.6_{-0.1}$ | $51.8_{+3.5}$ | $65.7_{+0.1}$ | $85.9$ | $89.3_{+0.2}$ | $70.6_{-0.1}$ | $93.4_{+0.2}$ | $44.0_{+0.5}$ | $67.8_{+0.3}$ | $65.8_{+0.6}$ |
| | TDA | $68.3_{+1.7}$ | $66.0_{+3.5}$ | $25.4_{+0.7}$ | $60.6_{+12.3}$ | $66.9_{+1.3}$ | $86.1_{+0.2}$ | $89.6_{+0.5}$ | $72.5_{+1.8}$ | $93.4_{+0.2}$ | $45.5_{+2.0}$ | $71.0_{+3.5}$ | $\underline{67.7}_{+2.5}$ |
| | DMN | $68.0_{+1.4}$ | $64.8_{+2.3}$ | $24.9_{+0.2}$ | $59.8_{+11.5}$ | $67.0_{+1.4}$ | $84.2_{-1.7}$ | $89.9_{+0.8}$ | $73.3_{+2.6}$ | $92.2_{-1.0}$ | $44.8_{+1.3}$ | $70.3_{+2.8}$ | $67.2_{+2.0}$ |
| | OGA | $68.8_{+2.2}$ | $66.1_{+3.6}$ | $24.9_{+0.2}$ | $61.8_{+13.5}$ | $66.2_{+0.6}$ | $86.2_{+0.3}$ | $88.5_{-0.6}$ | $72.0_{+1.3}$ | $93.4_{+0.2}$ | $44.9_{+1.4}$ | $69.4_{+1.9}$ | $\underline{67.6}_{+2.4}$ |
| | ADAPT | $69.2_{+2.6}$ | $65.4_{+2.9}$ | $24.4_{-0.3}$ | $51.9_{+3.6}$ | $67.6_{+2.0}$ | $77.4_{-8.5}$ | $88.5_{-0.6}$ | $72.9_{+2.2}$ | $92.4_{-0.8}$ | $45.2_{+1.7}$ | $69.9_{+2.4}$ | $65.9_{+0.7}$ |
| | StatA | $66.2_{-0.4}$ | $63.6_{+1.1}$ | $24.3_{-0.4}$ | $52.3_{+4.0}$ | $67.4_{+1.8}$ | $88.0_{+2.1}$ | $92.5_{+3.4}$ | $72.7_{+2.0}$ | $94.2_{+1.0}$ | $46.8_{+3.3}$ | $68.8_{+1.3}$ | $67.0_{+1.7}$ |
| | MOON | $67.0_{+0.4}$ | $63.5_{+1.0}$ | $23.2_{-1.5}$ | $50.1_{+1.8}$ | $66.5_{+0.9}$ | $91.2_{+5.3}$ | $91.6_{+2.5}$ | $71.4_{+0.7}$ | $94.0_{+0.8}$ | $45.3_{+1.8}$ | $68.2_{+0.7}$ | $66.5_{+1.3}$ |
| Medium ($\xi = 0.01$) | TENT | $66.7_{+0.1}$ | $64.3_{+1.8}$ | $24.6_{-0.1}$ | $47.9_{-0.4}$ | $65.6_{+0.0}$ | $85.6_{+0.0}$ | $89.4_{+0.3}$ | $70.6_{-0.1}$ | $93.3_{+0.1}$ | $44.0_{+0.5}$ | $67.8_{+0.3}$ | $65.5_{+0.2}$ |
| | TDA | $68.2_{+1.6}$ | $65.6_{+3.1}$ | $25.2_{+0.5}$ | $56.5_{+8.2}$ | $66.5_{+0.9}$ | $85.8_{-0.1}$ | $89.3_{+0.2}$ | $72.6_{+1.9}$ | $93.5_{+0.3}$ | $45.2_{+1.7}$ | $70.1_{+2.6}$ | $67.1_{+1.9}$ |
| | DMN | $68.0_{+1.4}$ | $64.8_{+2.3}$ | $24.9_{+0.2}$ | $56.2_{+7.9}$ | $66.8_{+1.2}$ | $81.9_{-4.0}$ | $89.0_{-0.1}$ | $73.0_{+2.3}$ | $92.1_{-1.1}$ | $44.9_{+1.4}$ | $69.6_{+2.1}$ | $66.5_{+1.2}$ |
| | OGA | $68.6_{+2.0}$ | $65.4_{+2.9}$ | $24.9_{+0.2}$ | $58.6_{+10.3}$ | $66.2_{+0.6}$ | $85.9_{\pm0.0}$ | $89.8_{+0.7}$ | $72.2_{+1.5}$ | $93.4_{+0.2}$ | $44.8_{+1.3}$ | $69.4_{+1.6}$ | $67.2_{+1.9}$ |
| | ADAPT | $69.3_{+2.7}$ | $67.2_{+4.7}$ | $24.6_{-0.1}$ | $58.5_{+10.2}$ | $68.6_{+3.0}$ | $83.9_{-2.0}$ | $90.2_{+1.1}$ | $73.2_{+2.5}$ | $92.5_{-0.7}$ | $45.3_{+1.8}$ | $71.3_{+3.8}$ | $67.7_{+2.5}$ |
| | StatA | $69.6_{+3.0}$ | $65.9_{+3.4}$ | $27.3_{+2.6}$ | $52.3_{+4.0}$ | $73.2_{+7.6}$ | $89.1_{+3.2}$ | $94.6_{+5.5}$ | $75.6_{+4.9}$ | $94.3_{+1.1}$ | $46.8_{+3.3}$ | $69.7_{+2.2}$ | $\underline{68.9}_{+3.7}$ |
| | MOON | $76.5_{+9.9}$ | $73.9_{+11.4}$ | $28.3_{+3.6}$ | $51.5_{+3.2}$ | $75.1_{+9.5}$ | $95.3_{+9.4}$ | $94.8_{+5.7}$ | $75.7_{+5.0}$ | $95.3_{+2.1}$ | $50.0_{+6.5}$ | $73.4_{+5.9}$ | $\mathbf{71.8}_{+6.6}$ |
| High ($\xi = 0.001$) | TENT | $66.8_{+0.2}$ | $64.3_{+1.8}$ | $24.8_{+0.1}$ | $45.6_{-2.7}$ | $65.6_{+0.0}$ | $86.1_{+0.2}$ | $89.4_{+0.3}$ | $70.5_{-0.2}$ | $93.4_{+0.2}$ | $44.0_{+0.5}$ | $67.9_{+0.4}$ | $65.3_{+0.1}$ |
| | TDA | $67.9_{+1.3}$ | $65.1_{+2.6}$ | $25.1_{+0.4}$ | $55.3_{+7.0}$ | $65.6_{+0.0}$ | $86.3_{+0.7}$ | $89.0_{-0.1}$ | $72.5_{+1.8}$ | $93.6_{+0.4}$ | $45.1_{+1.6}$ | $69.7_{+2.2}$ | $66.6_{+1.4}$ |
| | DMN | $67.9_{+1.3}$ | $64.8_{+2.3}$ | $24.9_{+0.2}$ | $56.3_{+8.0}$ | $66.8_{+1.2}$ | $79.9_{-6.0}$ | $88.9_{-0.2}$ | $72.9_{+2.2}$ | $92.1_{-1.1}$ | $44.8_{+1.3}$ | $69.4_{+1.9}$ | $66.3_{+1.0}$ |
| | OGA | $67.9_{+1.3}$ | $64.6_{+2.1}$ | $24.9_{+0.2}$ | $59.0_{+10.7}$ | $66.2_{+0.6}$ | $85.6_{-0.3}$ | $89.7_{+0.6}$ | $72.2_{+1.5}$ | $93.5_{+0.3}$ | $44.7_{+1.2}$ | $68.9_{+1.4}$ | $67.0_{+1.8}$ |
| | ADAPT | $68.6_{+2.0}$ | $66.5_{+4.0}$ | $24.6_{-0.1}$ | $54.7_{+7.1}$ | $68.0_{+2.4}$ | $81.3_{-4.6}$ | $89.1_{\pm0.0}$ | $73.0_{+2.3}$ | $92.5_{-0.7}$ | $45.3_{+1.8}$ | $70.3_{+2.8}$ | $66.8_{+1.5}$ |
| | StatA | $71.9_{+5.3}$ | $66.0_{+3.5}$ | $27.9_{+3.2}$ | $51.8_{+3.5}$ | $74.7_{+9.1}$ | $89.3_{+3.4}$ | $94.8_{+5.7}$ | $76.4_{+5.7}$ | $94.4_{+1.2}$ | $47.0_{+3.5}$ | $69.8_{+2.3}$ | $\underline{69.5}_{+4.2}$ |
| | MOON | $82.2_{+15.6}$ | $77.2_{+14.7}$ | $30.0_{+5.3}$ | $51.6_{+3.3}$ | $77.4_{+11.8}$ | $95.8_{+9.9}$ | $95.3_{+6.2}$ | $77.0_{+6.3}$ | $95.7_{+2.5}$ | $51.2_{+7.7}$ | $74.6_{+7.1}$ | $\mathbf{73.4}_{+8.2}$ |
| Separate | TENT | $66.7_{+0.1}$ | $64.2_{+1.7}$ | $24.7_{\pm0.0}$ | $37.0_{-11.3}$ | $65.6_{+0.0}$ | $86.1_{+0.2}$ | $89.3_{+0.2}$ | $70.8_{+0.1}$ | $93.4_{+0.2}$ | $43.9_{+0.4}$ | $67.9_{+0.4}$ | $64.5_{-0.7}$ |
| | TDA | $67.4_{+0.8}$ | $64.6_{+2.1}$ | $24.9_{+0.2}$ | $55.3_{+7.0}$ | $65.9_{+0.3}$ | $85.2_{-0.7}$ | $88.9_{-0.2}$ | $72.3_{+1.6}$ | $93.6_{+0.4}$ | $45.0_{+1.5}$ | $69.6_{+2.1}$ | $66.6_{+1.4}$ |
| | DMN | $67.7_{+1.1}$ | $64.7_{+2.2}$ | $24.9_{+0.2}$ | $55.1_{+6.8}$ | $66.7_{+1.1}$ | $78.5_{-7.4}$ | $88.0_{-1.1}$ | $72.8_{+2.1}$ | $91.9_{-1.3}$ | $44.8_{+1.3}$ | $69.0_{+1.5}$ | $65.8_{+0.6}$ |
| | OGA | $67.2_{+0.6}$ | $64.1_{+1.6}$ | $24.9_{+0.2}$ | $58.0_{+9.7}$ | $66.1_{+0.5}$ | $85.4_{-0.5}$ | $89.5_{+0.4}$ | $72.1_{+1.4}$ | $93.4_{+0.2}$ | $44.6_{+1.1}$ | $68.7_{+1.2}$ | $66.7_{+1.5}$ |
| | ADAPT | $68.2_{+1.6}$ | $65.7_{+3.2}$ | $25.0_{+0.3}$ | $53.7_{+5.4}$ | $67.4_{+1.8}$ | $78.2_{-7.7}$ | $88.2_{-0.9}$ | $72.4_{+1.7}$ | $92.5_{-0.7}$ | $44.8_{+1.3}$ | $69.5_{+2.0}$ | $66.0_{+0.7}$ |
| | StatA | $71.7_{+5.1}$ | $64.9_{+2.4}$ | $28.9_{+4.2}$ | $48.2_{-0.1}$ | $75.2_{+9.6}$ | $88.9_{+3.0}$ | $95.2_{+6.1}$ | $77.6_{+6.9}$ | $94.3_{+1.1}$ | $45.8_{+2.3}$ | $69.0_{+1.5}$ | $\underline{69.1}_{+3.8}$ |
| | MOON | $82.4_{+15.8}$ | $76.7_{+14.2}$ | $32.1_{+7.4}$ | $51.2_{+2.9}$ | $77.8_{+12.2}$ | $95.5_{+9.6}$ | $95.1_{+6.0}$ | $77.9_{+7.2}$ | $95.9_{+2.7}$ | $53.6_{+10.1}$ | $74.6_{+7.1}$ | $\mathbf{73.9}_{+8.6}$ |

*Table 3.* **Runtime (seconds) on ImageNet.** The second row indicates the number of iterations. "CLIP forward" denotes the inference latency, while others report the net algorithm time.

| Batch size | CLIP forward | TransCLIP | StatA | ADAPT | MOON |
|---|---|---|---|---|---|
| | | 10 | 10 | 1 | 10 |
| 128 | 8.47 | 0.08 | 0.11 | 0.06 | **0.03** |
| 1,000 | 8.99 | 0.24 | 0.31 | 0.13 | **0.03** |
| 50,000 | 36.90 | 0.47 | 0.59 | 0.21 | **0.05** |

*Table 4.* **Ablation study on components.** Each reported performance is averaged over all datasets and scenarios.

| $\alpha$ | $\gamma$ | Update $\mu$ | Update $\kappa$ | Bs=64 | Bs=1,000 |
|---|---|---|---|---|---|
| ✗ | ✓ | ✓ | ✓ | 69.0 | 68.9 |
| ✓ | ✗ | ✓ | ✓ | 72.3 | 70.4 |
| ✓ | ✓ | ✗ | ✓ | 66.8 | 66.5 |
| ✓ | ✓ | ✓ | ✗ | 70.8 | 68.9 |
| ✓ | ✓ | ✓ | ✓ | **72.4** | **71.1** |

stronger shrinkage to more confident classes. We also compare an alternative form $\alpha_k = \exp(-\lambda_k)$, but find it less effective. A possible reason is that it bounds $\alpha_k$ in $(0, 1]$, and in turn bounds $\beta_k$ within $[n_k/(n_k + 1), 1)$, making the update remain noticeably influenced by empirical estimates. These results further support our inverse design $\alpha_k = 1/\lambda_k$.

We then examine the design of class confidence $\lambda_k$ in Tab. 6, using three representative tasks from both extreme and mild imbalance scenarios. Average-only confidence may suppress rare-but-valid classes, while max-only confidence may

*Table 5.* **Design choices of $\alpha_k$.** Results are reported on batch adaptation, *Medium* scenario, batch size of 1,000. "Average" denotes the mean performance over all datasets.

| $\alpha_k$ | $1/\lambda_k$ (ours) | $\lambda_k$ | $\exp(-\lambda_k)$ |
|---|---|---|---|
| ImageNet | **81.6** | 68.5 | 71.1 |
| Average | **72.9** | 67.6 | 68.8 |

overestimate classes with occasional high-confidence predictions. Various types of means mitigate this issue similarly. We choose geometric mean as it yields a more generalizable solution across broader scenarios, especially those mild, near-uniform cases where MOON performs below average.

### 5.4. Further Analysis

**Comparisons with variants.** Although vMF distribution is topologically native to the hypersphere of normalized VLM embeddings, Tab. 7 shows that replacing the GMM in StatA with a vMF mixture unexpectedly underperforms standard StatA. We explain this with the *linear representation hypothesis* (Park et al., 2023), which suggests that VLM representations reside in low-dimensional linear subspaces and exhibit strong anisotropy. Consequently, constrained by isotropic scalar $\kappa$, vMF struggles to capture such manifold geometry, while Gaussian covariance implicitly approximates it in Euclidean space. However, in MOON, this limitation is largely alleviated with our dynamic shrinkage strength modeling: MOON achieves comparable performance

*Table 6.* **Design choices of** $\lambda_k$, across extreme and mild imbalance scenarios. Each reported performance is averaged over all datasets.

| $\lambda_k$ | Task 1 | Task 2 | Task 3 | Avg. |
|---|---|---|---|---|
| *Extreme imbalance* | | | | |
| Geo. mean (ours) | 73.1 | 72.9 | 73.4 | 73.1 |
| Ari. mean | **73.5** | **74.2** | **74.1** | **73.9** |
| Harm. mean | 73.4 | 73.5 | 73.8 | 73.6 |
| Average only | 72.4 | 73.3 | 73.9 | 73.2 |
| Max only | 71.7 | 70.8 | 71.6 | 71.4 |
| *Mild imbalance* | | | | |
| Geo. mean (ours) | **69.4** | **69.2** | 66.5 | **68.4** |
| Ari. mean | 69.0 | 68.6 | 65.7 | 67.8 |
| Harm. mean | 69.3 | 69.0 | 66.1 | 68.1 |
| Average only | 68.2 | 67.1 | 66.7 | 67.3 |
| Max only | 69.0 | 68.9 | **66.9** | 68.3 |

*Table 7.* **Comparisons with variants.** Each reported performance is averaged over all datasets and scenarios.

| Method | Batch | | Online |
|---|---|---|---|
| | Bs=64 | Bs=1,000 | Bs=128 |
| StatA | 69.01 | 69.46 | 68.61 |
| StatA_vMF | 68.69 | 68.15 | 68.31 |
| MOON_Gaussian | 71.49 | 71.07 | 71.23 |
| MOON | **72.36** | **71.14** | **71.42** |

to its Gaussian variant with much higher efficiency.

**Extend MOON to sample-wise.** While online TTA methods usually assume access to only a single image at each step[4], vanilla MOON does not support sample-wise mode, as transductive learning inherently requires a batch of samples to perform probabilistic soft clustering. However, MOON can be easily extended to a sample-wise online mode by introducing a memory bank to collect historical samples, as what recent work (Zhang et al., 2025) has done.

Formally, assume a memory bank $\mathcal{B} = (\mathbf{f}_j, \bar{\mathbf{z}}_j)$, where $\mathbf{f}_j \in \mathcal{S}^{d-1}$ is cached historical feature, and $\bar{\mathbf{z}}_j \in \Delta^K$ is the corresponding soft label. Consider a newly arriving sample with normalized feature $f_\star$ and zero-shot prediction $\hat{\mathbf{y}}_\star$. Assume that current mixture parameters $M_\mathcal{B} = \{(\boldsymbol{\mu}_k^\mathcal{B}, \kappa_k^\mathcal{B})\}_{k=1}^K$ have already been estimated from available data in $\mathcal{B}$. Therefore, the PLE objective becomes[5]:

$$\min_{\mathbf{z}_\star \in \Delta^K} -\mathbf{z}_\star^\top \log \mathbf{p}_\star^\mathcal{B} - \sum_{j \in \mathcal{B}} \omega_{\star j}, \mathbf{z}_\star^\top \bar{\mathbf{z}}_j + \mathrm{KL}(\mathbf{z}_\star \| \hat{\mathbf{y}}_\star). \quad (16)$$

Solving the problem yields the closed-form predictor:

$$\mathbf{z}_\star = \frac{\hat{\mathbf{y}}_\star \odot \exp(\log \mathbf{p}_\star^\mathcal{B} + \sum_{j \in \mathcal{B}} \omega_{\star j} \bar{\mathbf{z}}_j)}{(\hat{\mathbf{y}}_\star \odot \exp(\log \mathbf{p}_\star^\mathcal{B} + \sum_{j \in \mathcal{B}} \omega_{\star j} \bar{\mathbf{z}}_j))^\top \mathbb{1}_K}. \quad (17)$$

---

[4]For consistency, we assume a batch-wise access of test samples for all methods in this paper.

[5]Note that the KL anchor term $\mathcal{R}(\mathbf{M})$ now becomes constant with respect to new assignment $\mathbf{z}_\star$.

*Table 8.* **Extend MOON to sample-wise.** Results are reported on online adaptation, *Medium* scenario, batch size of 128.

| Method | ImageNet | SUN397 | Food101 | DTD |
|---|---|---|---|---|
| CLIP | 66.6 | 62.5 | 85.9 | 43.5 |
| StatA | 69.6 | 65.9 | 89.1 | 46.8 |
| MOON | 76.5 | 73.9 | 95.3 | 50.0 |
| MOON[†] | 73.0 | 69.5 | 92.4 | 44.1 |

*Table 9.* **Scaling to other CLIP backbones and VLMs.**

| | #Params | ImageNet | | Average | |
|---|---|---|---|---|---|
| | | Zero-shot | Ours | Zero-shot | Ours |
| CLIP RN50 | 102M | 58.2 | 76.8 $_{+18.6}$ | 58.7 | 65.7 $_{+7.0}$ |
| CLIP RN101 | 120M | 61.3 | 77.1 $_{+15.8}$ | 59.5 | 64.8 $_{+5.3}$ |
| CLIP ViT-B/32 | 151M | 62.0 | 78.5 $_{+16.5}$ | 61.9 | 68.0 $_{+6.1}$ |
| CLIP ViT-L/14 | 428M | 73.5 | 84.9 $_{+11.4}$ | 72.6 | 77.4 $_{+4.8}$ |
| OpenCLIP | 150M | 73.0 | 83.4 $_{+10.4}$ | 72.5 | 76.7 $_{+4.2}$ |
| SigLIP | 878M | 82.3 | 91.3 $_{+9.0}$ | 81.8 | 85.1 $_{+3.2}$ |
| EVA-CLIP | 1.14B | 78.0 | 81.2 $_{+3.2}$ | 76.5 | 78.4 $_{+1.9}$ |

Hence, MOON can make an immediate prediction for a newly arriving sample without rerunning full-batch transduction.

At the same time, using bank statistics, the mixture parameters continue to be stabilized by the KL-anchored dynamic shrinkage mechanism. Specifically, the parameter update keeps the same anchored form as Eq. (14):

$$\boldsymbol{\mu}_k^\mathcal{B} = \frac{\beta_k^\mathcal{B} \boldsymbol{v}_k^\mathcal{B} + (1 - \beta_k^\mathcal{B}) \boldsymbol{\mu}_k'}{\|\beta_k^\mathcal{B} \boldsymbol{v}_k^\mathcal{B} + (1 - \beta_k^\mathcal{B}) \boldsymbol{\mu}_k'\|},$$
$$\mathcal{A}_d(\kappa_k^\mathcal{B}) = \|\beta_k^\mathcal{B} \boldsymbol{v}_k^\mathcal{B} + (1 - \beta_k^\mathcal{B}) \boldsymbol{\mu}_k'\|, \quad (18)$$

where $\boldsymbol{v}_k^\mathcal{B} = \frac{\sum_{j \in \mathcal{B}} \gamma_{j,k} \bar{\mathbf{z}}_{j,k} \mathbf{f}_j}{\sum_{j \in \mathcal{B}} \gamma_{j,k} \bar{\mathbf{z}}_{j,k}}$ and $\beta_k^\mathcal{B} = \frac{\sum_{j \in \mathcal{B}} \gamma_{j,k} \bar{\mathbf{z}}_{j,k}}{\sum_{j \in \mathcal{B}} \gamma_{j,k} \bar{\mathbf{z}}_{j,k} + \alpha_k}$. According to this, we establish a sample-wise version MOON[†] in Tab. 8. As shown, MOON[†] could still achieve excellent performance. In practice, cold-starting the memory bank $\mathcal{B}$ with a held-out set may lead to better results.

**More backbones and architectures.** We further extend our evaluation to include 4 additional CLIP backbones and 3 other VLMs in Tab. 9, which demonstrate the universal effectiveness of our MOON across diverse model architectures, scales, and types. App. I.2 and I.3 provide more details.

## 6. Conclusion

In this work, we systematically revisit test-time transduction for VLMs under realistic class imbalance from the perspective of PLE, and reveal the brittleness and underlying limitations of existing transductive methods. Therefore, we propose MOON, which is based on a mixture of vMF distributions and dynamically adjusts shrinkage strength at both the instance and class levels to mitigate negative transfer. Extensive experiments validate that MOON achieves effective, efficient and practical adaptation.

## Acknowledgment

Ziyue Qiao is supported by the National Natural Science Foundation of China (No. 62406056) and the Guangdong Basic and Applied Basic Research Foundation (No. 2024A1515140114). The authors are grateful to the anonymous reviewers for their efforts and insightful suggestions to improve this paper.

## Impact Statement

This work advances test-time adaptation for vision-language models by addressing realistic class imbalance and distribution shifts. Our approach leverages a mixture of von Mises-Fisher distributions with dynamic, data-driven shrinkage to suppress unreliable predictions and mitigate negative transfer. We provide theoretical grounding through penalized likelihood estimation, and demonstrate empirical effectiveness and efficiency across multiple datasets and backbones. By enabling robust and efficient adaptation, our method improves the reliability of deployed vision-language systems in practical applications such as image classification, retrieval, and zero-shot reasoning, without requiring access to model weights or additional training data.

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

# A. Overall Procedure of MOON

The overall procedure of our proposed MOON is presented in Alg. 1. The BSUM-style iterative optimization of MOON can be conceptualized as a generalized Expectation-Maximization (EM) algorithm: fixing parameters $(\boldsymbol{\mu}_k, \kappa_k)$ to update assignments $\mathbf{z}$ corresponds to the E-step, while fixing $\mathbf{z}$ and updating $(\boldsymbol{\mu}_k, \kappa_k)$ corresponds to the M-step.

---

**Algorithm 1** Overall procedure of MOON

---

**Require:** Visual feature embeddings $\{\mathbf{f}_i\}_{i=1}^N, \mathbf{f}_i \in \mathcal{S}^{d-1}$, textual class embeddings $\{\mathbf{t}_k\}_{k=1}^K, \mathbf{t}_k \in \mathcal{S}^{d-1}$; Fixed hyperparameters (no need for tuning): VLM temperature $\tau$, Number of neighbors in Laplacian term $m$, Iterations $T$.

**Ensure:** Latent label assignments as final predictions $\mathbf{z} \in [0, 1]^{N \times K}$.

     *# Initialization*

1: Compute zero-shot logits $\hat{\mathbf{y}} = \{\hat{\mathbf{y}}_{i,k}\}_{k=1}^K, \hat{\mathbf{y}}_{i,k} = \frac{\exp(\mathbf{f}_i^\top \mathbf{t}_k / \tau)}{\sum_j \exp(\mathbf{f}_i^\top \mathbf{t}_j / \tau)}$; initialize assignments $\mathbf{z} \leftarrow \hat{\mathbf{y}}$.      ▷ See Eq. (1)

2: Initialize vMF parameters $\boldsymbol{\mu} = \{\boldsymbol{\mu}_k\}_{k=1}^K$ and $\boldsymbol{\kappa} = \{\kappa_k\}_{k=1}^K$; prior anchors $\boldsymbol{\mu}'$ and $\boldsymbol{\kappa}'$.      ▷ See Eq. (7)

3: Build a $m$-NN affinity graph $\mathbf{W} = [\omega_{i,j}] \in \mathcal{R}^{N \times N}$, where $\omega_{i,j} = \begin{cases} \mathbf{f}_i^\top \mathbf{f}_j, & \text{if } \mathbf{f}_j \text{ is the } m\text{-nearest neighbors of } \mathbf{f}_i \\ 0, & \text{otherwise} \end{cases}$.

4: Compute class-level adjustment weights $\boldsymbol{\alpha} = \{\alpha_k\}_{k=1}^K$ with confidence $\boldsymbol{\lambda} = \{\lambda_k\}_{k=1}^K$.      ▷ See Eq. (10) and (11)

5: Initialize instance-level adjustment weights $\boldsymbol{\gamma} = \{\gamma_i\}_{i=1}^N$.      ▷ See Eq. (9)

     *# BSUM-style iterative optimization*

6: **for** $t = 1$ **to** $T$ **do**

        *# Block update with respect to assignments $\mathbf{z}$*

7:      Compute vMF log-likelihood scores: $\log p_{i,k}^{\text{vMF}} \leftarrow \kappa_k \boldsymbol{\mu}_k^\top \mathbf{f}_i + \log C_d(\kappa_k)$.      ▷ See Eq. (6)

8:      Update $\mathbf{z}$ by single-pass linear approximation: $\mathbf{z}_i^{(t+1)} = \frac{\hat{\mathbf{y}}_i \odot \exp(\log \mathbf{p}_i^{\text{vMF}} + \sum_j \omega_{ij} \mathbf{z}_j^{(t)})}{(\hat{\mathbf{y}}_i \odot \exp(\log \mathbf{p}_i^{\text{vMF}} + \sum_j \omega_{ij} \mathbf{z}_j^{(t)}))^\top \mathbb{1}_K}$.      ▷ See Eq. (12)

9:      Update instance-level adjustment weights: $\gamma_i = 1 - \frac{H(\mathbf{z}_i)}{\log K}$.

        *# Compute shrinkage strengths*

10:     Compute shrinkage strengths $\boldsymbol{\beta} = \{\beta_k\}_{k=1}^K$: $n_k \leftarrow \sum_{i=1}^N \gamma_i z_{i,k}, \quad \beta_k \leftarrow \frac{n_k}{n_k + \alpha_k}$.

        *(In practice, we use hard label counts for stability: $n_k = \sum_i \gamma_i \mathbb{I}[\arg\max_j z_{i,j} = k]$.)*

        *# Block update with respect to distribution parameters $(\boldsymbol{\mu}, \boldsymbol{\kappa})$*

11:     Compute empirical estimates from standard MLE: $\mathbf{v}_k \leftarrow \frac{\sum_i \gamma_i z_{i,k} \mathbf{f}_i}{\sum_i \gamma_i z_{i,k}}$.

12:     Update $\boldsymbol{\mu}$ and $\mathcal{A}_d(\boldsymbol{\kappa})$ by closed-form anchor shrinkage: $\boldsymbol{\mu}_k = \frac{\beta_k \boldsymbol{v}_k + (1-\beta_k)\boldsymbol{\mu}'_k}{\|\beta_k \boldsymbol{v}_k + (1-\beta_k)\boldsymbol{\mu}'_k\|}$, $\mathcal{A}_d(\kappa_k) = \|\beta_k \boldsymbol{v}_k + (1-\beta_k)\boldsymbol{\mu}'_k\|$,  ▷ See Eq. (14)

13:     Inversely estimate $\boldsymbol{\kappa}$ using Banerjee approximation: $\kappa_k = \mathcal{A}_d^{-1}(\kappa_k)$.      ▷ See Eq. (15)

14: **end for**

15: **return** $\mathbf{z}$.

---

In implementation, we modify the shrinkage strength $\beta_k$ by replacing the soft assignment predictions with hard ones on the vertices of the probability simplex $\Delta_K$ following StatA (Zanella et al., 2025), which is:

$$\beta_k \approx \frac{\sum_{i=1}^N \mathbb{1}\left[k = \arg\max_r \gamma_{i,k} z_{i,r}\right]}{\sum_{i=1}^N \mathbb{1}\left[k = \arg\max_r \gamma_{i,k} z_{i,r}\right] + \alpha_k}. \tag{19}$$

This design has been proven to be more robust in practice, with experimental results shown in App. J.

**Complexity analysis.** We provide a theoretical complexity analysis of our vMF-based MOON and Gaussian-based state-of-the-art method StatA, further demonstrating the significant efficiency advantage of vMF mixtures that has been largely overlooked in prior works. Formally, let $N, K, d$ denote the number of samples, classes, and feature dimensions, respectively. Let $m$ be the number of neighbors and $T$ the number of iterations.

*(1) Graph construction:* Both methods share the initial cost of constructing the affinity graph. This requires $O(N^2 d)$ for dense similarity computation, which dominates the pre-processing cost.

*(2) Per-iteration cost:* For MOON, each iteration is dominated by two parts: (i) likelihood computation and parameter aggregation, which scales with $O(NKd)$; and (ii) sparse graph smoothing, which scales with $O(NmK)$. Thus, the

per-iteration complexity is $O(NK(d+m))$. In contrast, StatA incurs higher costs due to two factors: (i) it performs $L_G$ inner loops for assignment updates (typically $L_G = 5$), inflating the smoothing cost to $O(L_G NmK)$; and (ii) its parameter update involves computing per-class covariance matrices, adding an $O(Kd^2)$ term.

*(3) Total cost:* Excluding the shared graph construction, the total asymptotic complexities are:

$$\mathcal{O}_{\text{MOON}} \approx T \cdot NK(d+m) \quad \text{vs.} \quad \mathcal{O}_{\text{StatA}} \approx T \cdot (NKd + L_G NmK + Kd^2). \tag{20}$$

MOON achieves superior efficiency by avoiding the quadratic complexity $O(d^2)$ in parameter updates and eliminating the need for inner assignment loops.

**Convergence guarantee of optimization algorithm.** Our algorithm can be analyzed within the standard BSUM framework. Let $\mathcal{L}(z, \Theta)$ denote the objective and $\Theta = \{\mu_k, \kappa_k\}_{k=1}^K$. Given that: *(1) Affinity matrix $W$ is PSD; (2) Anchor weights $\alpha_k$ and $\gamma_i > 0$; (3) Fixing $z$, the parameter subproblem in $\Theta$ has a unique minimizer.*

Then each outer iteration of our algorithm is a valid BSUM step:

- **z-update**: The only non-convex part is the Laplacian term. Since $W$ is PSD, this term is concave and can be upper-bounded by its first-order Taylor expansion at the current iteration (proofs in App. G.1). Minimizing this tight surrogate yields Eq. (12), i.e., the update is an exact minimizer of the majorized subproblem.

- **$\Theta$ update**: Fixing $z$, our objective is strictly convex, and Eqs. (13)-(14) are the corresponding closed-form exact updates. This step further decreases the objective.

Therefore, each outer iteration satisfies

$$\mathcal{L}(z^{(t+1)}, \Theta^{(t+1)}) \leq \mathcal{L}(z^{(t)}, \Theta^{(t)}). \tag{21}$$

Since $\mathcal{L}_a$ is lower-bounded, the objective sequence is **monotonically non-increasing and thus convergent**. And, every limit point of the iterate sequence is a **coordinate-wise minimum**. And, as BSUM doe not require solving subproblems to full convergence at each outer iteration, **single-pass z-update preserves convergence guarantee**.

# B. Revisiting StatA

StatA (Zanella et al., 2025) first discuss transduction under the realistic class imbalanced scenarios at test-time. Unlike our MOON which operates on the unit hypersphere, StatA assumes that the visual feature representations $\{\mathbf{f}_i\}_{i=1}^N$ follow a Gaussian Mixture Model (GMM) in the Euclidean space. To handle performance degradation, it introduces a KL-divergence-based regularization term acting as a statistical anchor. The optimization objective is to minimize the PLE objective penalized by the deviation from zero-shot prior anchors leveraged from text $\{\mathcal{N}(\boldsymbol{\mu}_k', \boldsymbol{\Sigma}_k')\}_{k=1}^K$:

$$\mathcal{L}_{\text{StatA}}(\mathbf{z}, \theta) = \underbrace{-\sum_{i=1}^N \mathbf{z}_i^\top \log \mathcal{N}(f_i|\boldsymbol{\mu}_k, \boldsymbol{\Sigma}_k)}_{\text{negative log-likelihood (NLL)}} + \mathcal{R}(\mathbf{z}) + \alpha \underbrace{\sum_{k=1}^K \text{KL}\big(\mathcal{N}(\boldsymbol{\mu}_k', \boldsymbol{\Sigma}_k') \| \mathcal{N}(\boldsymbol{\mu}_k, \boldsymbol{\Sigma}_k)\big)}_{\text{statistical anchor}},$$

$$\text{where} \quad \mathcal{R}(\mathbf{z}) = \underbrace{-\sum_{i,j} \omega_{ij} \mathbf{z}_i^\top \mathbf{z}_j}_{\text{Laplacian reg.}} + \underbrace{\sum_{i=1}^N \text{KL}(\mathbf{z}_i \| \hat{\mathbf{y}}_i)}_{\text{text supervision}}. \tag{22}$$

where $\theta = \{\boldsymbol{\mu}_k, \boldsymbol{\Sigma}_k\}_{k=1}^K$ are the mixture parameters, and $\alpha > 0$ is a hyperparameter as anchor term weight. The anchor distribution is initialized as: $\boldsymbol{\mu}_k' = \mathbf{t}_k$ and $\boldsymbol{\Sigma}' = \text{Diag}\left(\frac{\sum_{i,k} \hat{y}_{i,k}(\mathbf{f}_i - \boldsymbol{\mu}_k)(\mathbf{f}_i - \boldsymbol{\mu}_k)^\top}{\sum_{i,k} \hat{y}_{i,k}}\right)$.

The algorithm is performed via BSUM-style iterative optimization, which alternates between assignments $\mathbf{z}$ block and distribution parameters $\theta$ block:

- **Assignment update:** Fixing $\theta$, the assignments $z_{i,k}$ are updated based on the Gaussian likelihoods $p_{i,k} \propto \frac{1}{\sqrt{|\boldsymbol{\Sigma}_k|}} \exp\left(-\frac{1}{2}(\mathbf{f}_i - \boldsymbol{\mu}_k)^\top \boldsymbol{\Sigma}_k^{-1}(\mathbf{f}_i - \boldsymbol{\mu}_k)\right)$, similar rule as Eq. (12).

- **Parameter update:** Fixing $z$, the parameters are updated in closed form. Crucially, the update for the mean $\boldsymbol{\mu}_k$ and covariance $\boldsymbol{\Sigma}_k$ exhibits an *anchor shrinkage* behavior:

$$\boldsymbol{\mu}_k = \beta_k \mathbf{v}_k + (1 - \beta_k)\boldsymbol{\mu}'_k, \quad \boldsymbol{\Sigma}_k = \beta_k T_k + (1 - \beta_k)(\Sigma' + \text{Diag}((\boldsymbol{\mu}'_k - \boldsymbol{\mu}_k)^2)), \tag{23}$$

with empirical estimates as:

$$\boldsymbol{v}_k = \frac{\sum_{i=1}^{N} z_{i,k}\mathbf{f}_i}{\sum_{i=1}^{N} z_{i,k}}; \quad \boldsymbol{T}_k = \frac{\sum_{i=1}^{N} z_{i,k}\text{Diag}((\mathbf{f}_i - \boldsymbol{\mu}_k)^2)}{\sum_{i=1}^{N} z_{i,k}}. \tag{24}$$

Here, $n_k = \sum_i z_{i,k}$ is the soft count, and $\beta_k = \frac{n_k}{n_k + \alpha} \approx \frac{\sum_i \mathbb{1}[k=\arg\max_r z_{i,r}]}{\sum_i \mathbb{1}[k=\arg\max_r z_{i,r}]+\alpha} \in [0,1]$ denotes the shrinkage strength.

**Limitations.** As shown in Eq. (23), the shrinkage strength $\beta_k$, controlled by soft count $n_k$ and anchor weight $\alpha$, balances the trade-off between empirical estimate $\mathbf{v}_k$ and the prior anchor $\boldsymbol{\mu}'_k$. However, anchor weight $\alpha$ is a *fixed scalar hyperparameter* shared across all classes and samples. This implies a *static shrinkage strength* modeling that ignores the varying reliability of different classes (e.g., outliers vs. effective classes) or instances, rendering StatA suboptimal in both accuracy and robustness under realistic class imbalance, as discussed in Sec. 1.

# C. Experimental Details

## C.1. Datasets

We evaluate our proposed MOON and other baselines on 11 widely-used public datasets for fine-grained visual classification. These datasets cover a diverse range of domains, including generic objects, scenes, textures, satellite imagery, and specific fine-grained categories. Specifically, the benchmark includes: ImageNet (Deng et al., 2009), SUN397 (Xiao et al., 2010), Aircraft (Maji et al., 2013), EuroSAT (Helber et al., 2019), StanfordCars (Krause et al., 2013), Food101 (Bossard et al., 2014), Pets (Parkhi et al., 2012), Flowers102 (Nilsback & Zisserman, 2008), Caltech101 (Fei-Fei et al., 2004), DTD (Cimpoi et al., 2014), and UCF101 (Soomro et al., 2012). Detailed statistics for these datasets are provided in Tab. 10.

## C.2. Baselines

We compare our MOON against a comprehensive set of baselines, which are categorized into: (1) **Transductive methods**: EM-Dirichlet (Dirichlet) (Martin et al., 2024), ZLaP (Kalantidis et al., 2024), GDA-CLIP (Wang et al., 2024), TransCLIP (Zanella et al., 2024), ADAPT (Zhang et al., 2025), and StatA (Zanella et al., 2025)[6]; (2) **Online TTA methods**: TENT (Wang et al., 2020), TDA (Karmanov et al., 2024), DMN (Zhang et al., 2024b), and OGA (Fuchs et al., 2025). We also incorporate another TTA method MTA (Zanella & Ben Ayed, 2024) that requires per-image augmentation.

The details and experimental configurations of these baselines are listed as below:

- **ZLaP (CVPR'24)**: introduces a non-parametric framework that leverages the graph structure of unlabeled data via label propagation, utilizing geodesic distances on the data manifold to address the modality gap in VLMs. The number of nearest neighbors $m$ is set to 5, scale parameter of RBF kernel function $\gamma$ is set to 5.0, and the clamping factor $\alpha$ is fixed at 0.3. We don't specifically scale the similarity matrix.

- **EM-Dirichlet (CVPR'24)**: frames transduction on the unit simplex by modeling class-conditional feature distributions with a Dirichlet law, solving the MLE problem via a hyperparameter-free Block Majorization-Minimization algorithm. The temperature $T$ in the probabilities is fixed to 30.

- **GDA-CLIP (ICLR'24)**: applies Gaussian Discriminant Analysis (GDA) by assuming a shared covariance matrix for class features, estimating parameters via closed-form solutions and ensembling them with zero-shot logits. The ensemble weight is set to $\alpha$ by default.

---

[6]Since StatA and ADAPT are also evaluated under online settings, we provide their corresponding results in the main manuscript.

- **TransCLIP (NeurIPS'24)**: formulates adaptation as a regularized MLE with a text-guided KL-divergence penalty, employing an iterative optimization procedure that decouples sample assignments and parameter updates. Here, text-guided KL divergence penalty $\lambda$ is set to 1, and the number of nearest neighbors $m$ is set to 3.

- **ADAPT (NeurIPS'25)**: presents a backpropagation-free method that reframes adaptation as Gaussian probabilistic inference with closed-form updates, utilizing a knowledge bank to efficiently support online and transductive settings. We set the bank size $L$ to 12, and the momentum coefficient for parameter update $\alpha$ to 0.9.

- **StatA (CVPR'25)**: addresses realistic scenarios with variable effective classes by employing a statistical anchor regularization within a Gaussian Mixture Model (GMM) to dynamically constrain features near text-derived priors. The anchor term weight $\alpha$ is set to 1, with the number of nearest neighbors $m$ set to 3. We use hard $\beta_k$ by default.

- **TENT (ICLR'21)**: adapts the model by minimizing the Shannon entropy of predictions on target data, specifically updating the affine parameters of Batch Normalization layers to align internal statistics online. We set the learning rate to 1e-3, and perform 5 adaptation steps for each batch.

- **TDA (CVPR'24)**: utilizes a lightweight key-value cache system with entropy-based filtering and introduces a negative cache mechanism to explicitly penalize unlikely classes using negative pseudo-labeling. All the configuration is kept the same as those set in the original paper on ImageNet.

- **DMN (CVPR'24)**: integrates a static memory for pre-trained knowledge and a dynamic memory for historical test features, employing a cross-attention strategy to refine decision boundaries based on temporal context. All the configuration is kept the same as those set in the original paper on ImageNet.

- **OGA (CVPRW'25)**: reframes online adaptation as a Maximum A Posteriori (MAP) estimation problem using multivariate Gaussian distributions and zero-shot priors to calibrate predictions without gradient backpropagation. THe memory update threshold $\tau$ is set to 0.01, with cache memory capacity set to 8.

- **MTA (CVPR'24)**: proposes a training-free strategy that leverages MeanShift on augmented views to identify distribution modes, jointly optimizing a learnable inlierness score to robustly aggregate visual information. All the configuration is kept the same as those set in the original paper on ImageNet.

## C.3. Prompts

Following (Zhang et al., 2022), we adopt default, fixed prompt templates to initialize text embeddings for all methods, as illustrated in Tab. 10.

*Table 10.* Dataset information and prompt templates.

| Name | Other name | # $K$ | # $N$ | Description | Prompt template |
|---|---|---|---|---|---|
| SUN397 | SUN397 | 397 | 19,850 | Scenes classification | `"a photo of a [ ]."` |
| Aircraft | FGVCAircraft | 100 | 3,333 | Aircraft classification | `"a photo of a [ ], a type of aircraft."` |
| EuroSAT | EuroSAT | 10 | 8,100 | Satellite images classification | `"a centered satellite photo of [ ]."` |
| StanfordCars | Cars | 196 | 8,041 | Cars classification | `"a photo of a [ ]."` |
| Food101 | Food101 | 101 | 30,300 | Food classification | `"a photo of [ ], a type of food."` |
| Pets | OxfordPets | 37 | 3,669 | Pets classification | `"a photo of [ ], a type of pet."` |
| Flowers102 | OxfordFlowers | 102 | 2,463 | Flowers classification | `"a photo of a [ ], a type of flower."` |
| Caltech101 | Caltech101 | 101 | 2,465 | Objects classification | `"a photo of a [ ]."` |
| DTD | DTD | 47 | 1,692 | Textures classification | `"[ ] texture."` |
| UCF101 | UCF101 | 101 | 3,783 | Actions classification | `"a photo of a person doing [ ]."` |
| ImageNet | ImageNet-1K | 1000 | 50,000 | Objects classification | `"a photo of a [ ]."` |

## C.4. Data Sampler

In this section, we describe the sampling strategies for constructing realistic test-time scenarios, following the protocols in StatA (Zanella et al., 2025).

**Batch adaptation.**   To simulate realistic class sparsity where the label distribution within a batch is partial, we construct test batches with a limited number of effective classes. Specifically, given a batch size $B$ and a total of $K$ classes, we first determine the number of effective classes $K_{eff}$, which is either fixed or uniformly sampled from $[K_{eff}^{\min}, K_{eff}^{\max}]$. We then randomly select a subset of classes $\mathcal{C}_{batch}$ with size $K_{eff}$ and aggregate all their corresponding samples. The final test batch is formed by randomly sampling $B$ instances without replacement from this restricted pool, ensuring that the batch contains only a fraction of the total categories.

**Online adaptation.**   We generate non-i.i.d. data streams using a Dirichlet-based framework to evaluate robustness against temporal correlation (Yuan et al., 2023). The data stream is divided into slots, where the allocation of each class across slots follows a Dirichlet distribution $\mathrm{Dir}(\xi \cdot \mathbf{1})$. The scalar $\xi$ controls the correlation intensity: large values approximate an i.i.d. stream, while small values concentrate classes into fewer slots to create high temporal correlation. Additionally, we consider a separate sequential setting (simulating $\xi \to 0$), where classes are randomly permuted and all samples from a class appear contiguously before transitioning to the next, representing the most extreme temporal correlation.

### C.5. Implementation Details

Unless otherwise specified, we employ CLIP ViT-B/16 as the default backbone and evaluate performance using the Top-1 accuracy. Consistent with our black-box assumption, we utilize a fixed set of hyperparameters across all experiments without per-task tuning: we set the nearest neighbors $m = 3$ and the number of iterations to 10, while inheriting the temperature parameter $\tau$ directly from pre-trained VLM. For stability and robustness, we employ hard assignments for the shrinkage strength $\beta_k$, and dynamically update the instance-level weights $\gamma_i$ at each iteration. All experiments are conducted on a single NVIDIA RTX 4090 24GB GPU. To ensure statistical reliability given the stochastic data sampling, all reported results represent the average of 1,000 independent runs for batch adaptation and 100 runs for online adaptation, initialized with a fixed random seed of 1.

## D. Generality of KL-Anchored PLE for Exponential Families

In this section, we provide a formal proof that KL-based distribution anchor in the penalized likelihood estimation (PLE) formulation, i.e., $\mathcal{R}(\mathbf{M})$ in Eq. (2), yields an *adaptive shrinkage* behavior that enables convex combination update in the *mean-parameter space* for any (regular) exponential-family class-conditional mixture model.

### D.1. Problem Setup

Consider a $K$-class latent-variable mixture model with unlabeled samples $\{x_i\}_{i=1}^{N}$ and soft assignments $\mathbf{z}_{i,k} \in [0, 1]$ satisfying $\sum_{k=1}^{K} \mathbf{z}_{i,k} = 1$. For each class $k \in \{1, \ldots, K\}$, assume the class-conditional density function belongs to a *regular minimal exponential family*:

$$p(x \mid \eta) = h(x) \exp\big(\eta^\top T(x) - A(\eta)\big), \tag{25}$$

where $\eta$ is the natural parameter, $T(x)$ is the sufficient statistic, and $A(\eta)$ is the log-partition function. For a regular minimal exponential family, $A(\eta)$ is strictly convex and the mapping $\nabla A$ is one-to-one, relating natural parameters to *mean parameters* $\mu = \mathbb{E}_\eta[T(X)] = \nabla A(\eta)$.

Given soft assignments, define the class-wise soft counts and soft sufficient-statistic sums:

$$n_k = \sum_{i=1}^{N} \mathbf{z}_{i,k}, \qquad S_k = \sum_{i=1}^{N} \mathbf{z}_{i,k}\, T(x_i). \tag{26}$$

Let $q_k(x) = p(x \mid \eta_k')$ denote a fixed *anchor* distribution for class $k$. Conditioned on $\{\mathbf{z}_{i,k}\}$, the M-step minimizes the following KL-anchored PLE objective over $\{\eta_k\}_{k=1}^{K}$:

$$\mathcal{L}(\{\eta_k\}) = -\sum_{i=1}^{N} \sum_{k=1}^{K} \mathbf{z}_{i,k} \log p(x_i \mid \eta_k) + \alpha \sum_{k=1}^{K} \mathrm{KL}\Big(q_k \,\|\, p(\cdot \mid \eta_k)\Big), \qquad \alpha > 0. \tag{27}$$

## D.2. Closed-Form KL for Exponential Families

**Lemma D.1** (KL divergence within an exponential family). *Let $q(\cdot) = p(\cdot \mid \eta')$ and $p(\cdot) = p(\cdot \mid \eta)$ be members of the same exponential family* (25). *Then*

$$\mathrm{KL}\Big(p(\cdot \mid \eta') \,\|\, p(\cdot \mid \eta)\Big) = A(\eta) - A(\eta') - (\eta - \eta')^\top \mu', \qquad \mu' = \mathbb{E}_{\eta'}[T(X)] = \nabla A(\eta'). \tag{28}$$

*Proof.* By definition,

$$\mathrm{KL}\Big(p(\cdot \mid \eta') \,\|\, p(\cdot \mid \eta)\Big) = \mathbb{E}_{\eta'}\big[\log p(X \mid \eta') - \log p(X \mid \eta)\big]. \tag{29}$$

Using (25), we have

$$\log p(X \mid \eta') - \log p(X \mid \eta) = \big((\eta')^\top T(X) - A(\eta')\big) - \big(\eta^\top T(X) - A(\eta)\big), \tag{30}$$

since $\log h(X)$ cancels. Taking expectation under $p(\cdot \mid \eta')$ yields

$$\mathrm{KL}\Big(p(\cdot \mid \eta') \,\|\, p(\cdot \mid \eta)\Big) = (\eta' - \eta)^\top \mathbb{E}_{\eta'}[T(X)] + A(\eta) - A(\eta') \tag{31}$$

$$= A(\eta) - A(\eta') - (\eta - \eta')^\top \mu', \tag{32}$$

where $\mu' = \mathbb{E}_{\eta'}[T(X)] = \nabla A(\eta')$. $\qquad\square$

## D.3. KL Anchoring Implies Convex Combination in Mean-Parameter Space

**Theorem D.2** (KL-anchored M-step yields convex combination in mean space). *Assume each class-conditional model $p(\cdot \mid \eta_k)$ belongs to a regular minimal exponential family* (25). *Fixing soft assignments $\{\mathbf{z}_{i,k}\}$, the M-step objective* (27) *is strictly convex in each $\eta_k$ and admits a unique minimizer $\eta_k^\star$. Moreover, the corresponding mean parameter satisfies*

$$\nabla A(\eta_k^\star) = \frac{S_k + \alpha \mu_k'}{n_k + \alpha}, \qquad \mu_k' = \nabla A(\eta_k'). \tag{33}$$

*Equivalently, for $n_k > 0$ with empirical mean parameter $\hat{\mu}_k = S_k/n_k$,*

$$\nabla A(\eta_k^\star) = \beta_k \hat{\mu}_k + (1 - \beta_k)\mu_k', \qquad \beta_k = \frac{n_k}{n_k + \alpha} \in [0, 1]. \tag{34}$$

*Proof.* The objective (27) decomposes across classes: $\mathcal{L}(\{\eta_k\}) = \sum_{k=1}^K \mathcal{L}_k(\eta_k) + \text{const}$, where, using $\log p(x \mid \eta) = \log h(x) + \eta^\top T(x) - A(\eta)$,

$$-\sum_{i=1}^N \mathbf{z}_{i,k} \log p(x_i \mid \eta_k) = -\sum_{i=1}^N \mathbf{z}_{i,k}\big(\log h(x_i) + \eta_k^\top T(x_i) - A(\eta_k)\big) \tag{35}$$

$$= -\eta_k^\top \sum_{i=1}^N \mathbf{z}_{i,k} T(x_i) + \Big(\sum_{i=1}^N \mathbf{z}_{i,k}\Big) A(\eta_k) + \text{const} \tag{36}$$

$$= -S_k^\top \eta_k + n_k A(\eta_k) + \text{const}. \tag{37}$$

For the KL anchor term, apply Lemma D.1 with $q_k(\cdot) = p(\cdot \mid \eta_k')$ and $p(\cdot \mid \eta_k)$:

$$\mathrm{KL}\Big(q_k \,\|\, p(\cdot \mid \eta_k)\Big) = A(\eta_k) - A(\eta_k') - (\eta_k - \eta_k')^\top \mu_k', \qquad \mu_k' = \nabla A(\eta_k'). \tag{38}$$

Combining (37) and (38) (and dropping constants independent of $\eta_k$), we obtain

$$\mathcal{L}_k(\eta_k) = -S_k^\top \eta_k + n_k A(\eta_k) + \alpha\Big(A(\eta_k) - (\eta_k)^\top \mu_k'\Big) + \text{const} \tag{39}$$

$$= -(S_k + \alpha \mu_k')^\top \eta_k + (n_k + \alpha)A(\eta_k) + \text{const}. \tag{40}$$

Since $A(\eta)$ is strictly convex for a regular minimal exponential family and $n_k + \alpha > 0$, it follows that $\mathcal{L}_k(\eta_k)$ is strictly convex in $\eta_k$ and thus has a unique minimizer $\eta_k^\star$.

Taking the gradient of (40) and setting it to zero yields the first-order optimality condition:

$$-(S_k + \alpha \mu_k') + (n_k + \alpha)\nabla A(\eta_k^\star) = 0, \tag{41}$$

which implies (33). When $n_k > 0$, substituting $S_k = n_k \hat{\mu}_k$ into (33) gives (34) with $\beta_k = n_k/(n_k + \alpha) \in [0,1]$. $\qquad\square$

### D.4. Corollaries for Realistic Test-Time Scenarios

Theorem D.2 immediately yields two properties that are particularly relevant to realistic test-time settings with class imbalance and sparse effective label coverage.

**Corollary D.3** (No deviation for outlier classes). *If a class is entirely outlier (absent classes) in the current batch in the sense that $n_k = 0$, then the optimal mean parameter satisfies*

$$\nabla A(\eta_k^\star) = \mu_k' = \nabla A(\eta_k'). \tag{42}$$

*Moreover, since $\nabla A$ is injective for a regular minimal exponential family, it follows that*

$$\eta_k^\star = \eta_k'. \tag{43}$$

*That is, the optimal parameter for an outlier (absent) class is* exactly *its anchor parameter, and will not deviate due to the absence of evidence.*

*Proof.* Setting $n_k = 0$ and $S_k = \mathbf{0}$ in (33) gives $\nabla A(\eta_k^\star) = \mu_k'$. Injectivity of $\nabla A$ in a regular minimal exponential family implies $\eta_k^\star = \eta_k'$. $\qquad\square$

**Corollary D.4** (Bounded deviation for rare classes). *For any class with $n_k > 0$, the deviation from the anchor in mean-parameter space is bounded by*

$$\left\|\nabla A(\eta_k^\star) - \mu_k'\right\| = \beta_k \left\|\hat{\mu}_k - \mu_k'\right\| \leq \frac{n_k}{n_k + \alpha}\left\|\hat{\mu}_k - \mu_k'\right\|. \tag{44}$$

*Thus, when a class appears rarely (small $n_k$), its update magnitude away from the anchor is linearly shrunk by $\beta_k$.*

*Proof.* Equation (34) implies $\nabla A(\eta_k^\star) - \mu_k' = \beta_k(\hat{\mu}_k - \mu_k')$. Taking norms on both sides yields (44). $\qquad\square$

**Remarks.** Theorem D.2 formalizes a key mechanism behind KL-anchored PLE: regardless of the specific exponential-family choice, the anchor regularization transforms the M-step into a strictly convex problem whose first-order condition yields an adaptive shrinkage that enables convex combination in mean-parameter space. This property is particularly beneficial in realistic test-time adaptation, where many classes may be missing or under-represented within a batch (e.g., $K_{\text{eff}} \ll K$ and $\xi \to 0$), and naive maximum-likelihood estimation tends to overfit to locally biased statistics.

## E. Details of von Mises-Fisher Distributions

### E.1. Introduction to vMF distributions

Let $x \in \mathbb{R}^d$ be a random vector on the unit hypersphere $\mathbb{S}^{d-1}$, i.e., $\|x\|_2 = 1$. The probability density function of von Mises-Fisher (vMF) distribution on $\mathbb{S}^{d-1}$ is defined by

$$p(x; \boldsymbol{\mu}, \kappa) = \mathcal{C}_d(\kappa)\exp\left(\kappa\,\boldsymbol{\mu}^\top x\right), \qquad \boldsymbol{\mu} \in \mathbb{S}^{d-1}, \ \kappa \geq 0, \tag{45}$$

where $\boldsymbol{\mu}$ is the mean direction vector and $\kappa$ is the concentration scalar. The normalization constant is given by

$$\mathcal{C}_d(\kappa) = \frac{\kappa^\nu}{(2\pi)^{d/2}I_\nu(\kappa)}, \qquad \nu = \frac{d}{2} - 1, \tag{46}$$

where $I_\nu(\cdot)$ denotes the modified Bessel function of the first kind: $I_\nu(\kappa) = \left(\frac{1}{2}\kappa\right)^\nu \sum_{k=0}^\infty \frac{\left(\frac{1}{4}\kappa^2\right)^k}{k!\,\Gamma(\nu+k+1)}$, $\Gamma(\cdot)$ denotes Gamma function.

In our model, given a normalized visual feature embedding $\mathbf{f}_i \in \mathbb{S}^{d-1}$, we parameterize each class $k$ by vMF parameters $\mathcal{V}_k = (\boldsymbol{\mu}_k, \kappa_k)$. The log-likelihood is

$$\log p_{i,k}^{\mathrm{vMF}} = \log \mathcal{C}_d(\kappa_k) + \kappa_k \boldsymbol{\mu}_k^\top \mathbf{f}_i. \tag{47}$$

**Numerical approximation.** Taking logarithm of Eq. (46) yields

$$\log \mathcal{C}_d(\kappa) = \nu \log \kappa - \frac{d}{2}\log(2\pi) - \log I_\nu(\kappa). \tag{48}$$

Since $I_\nu(\kappa)$ is transcendental, we adopt the large-$\kappa$ asymptotic[7]

$$\log I_\nu(\kappa) \approx \kappa - \frac{1}{2}\log(2\pi\kappa), \tag{49}$$

which leads to

$$\log \mathcal{C}_d(\kappa) \approx \frac{d-1}{2}\log\kappa - \kappa - \frac{d-1}{2}\log(2\pi). \tag{50}$$

In implementation, since our assignment update Eq. (12) involves a softmax over classes; terms independent of $\kappa$ (e.g., $-\frac{d-1}{2}\log(2\pi)$ for fixed $d$) cancel out[8]. Therefore, we use the simplified approximation form

$$\log \mathcal{C}_d(\kappa) = \frac{d-1}{2}\log\kappa - \kappa. \tag{51}$$

### E.2. Derivation of KL divergence between two multivariate vMF distributions

Consider two vMF multivariate distributions $p(x) = \mathcal{V}_p(x; \boldsymbol{\mu}_p, \kappa_p)$ and $q(x) = \mathcal{V}_q(x; \boldsymbol{\mu}_q, \kappa_q)$ on $\mathbb{S}^{d-1}$. The Kullback-Leibler (KL) divergence is

$$\mathrm{KL}(p\|q) = \mathbb{E}_{x\sim p}[\log p(x) - \log q(x)]. \tag{52}$$

Using the vMF log-density from (45), we have

$$\log p(x) - \log q(x) = \log \frac{\mathcal{C}_d(\kappa_p)}{\mathcal{C}_d(\kappa_q)} + \kappa_p \boldsymbol{\mu}_p^\top x - \kappa_q \boldsymbol{\mu}_q^\top x. \tag{53}$$

Taking expectation w.r.t. $x \sim p$ yields

$$\mathrm{KL}(p\|q) = E_{x\sim p}\left[\log \frac{\mathcal{C}_d(\kappa_p)}{\mathcal{C}_d(\kappa_q)} + \kappa_p \boldsymbol{\mu}_p^\top x - \kappa_q \boldsymbol{\mu}_q^\top x\right] \tag{54}$$

$$= \log \frac{\mathcal{C}_d(\kappa_p)}{\mathcal{C}_d(\kappa_q)} + E_{x\sim p}[\kappa_p \boldsymbol{\mu}_p^\top x] - E_{x\sim p}[\kappa_q \boldsymbol{\mu}_q^\top x] \tag{55}$$

$$= \log \frac{\mathcal{C}_d(\kappa_p)}{\mathcal{C}_d(\kappa_q)} + \kappa_p \boldsymbol{\mu}_p^\top \mathbb{E}_p[x] - \kappa_q \boldsymbol{\mu}_q^\top \mathbb{E}_p[x]. \tag{56}$$

A standard vMF identity states that the mean of $x \sim \mathrm{vMF}(\boldsymbol{\mu}_p, \kappa_p)$ is aligned with $\boldsymbol{\mu}_p$:

$$\mathbb{E}_p[x] = \mathcal{A}_d(\kappa_p)\,\boldsymbol{\mu}_p, \qquad \mathcal{A}_d(\kappa) \triangleq \frac{I_{\frac{d}{2}}(\kappa)}{I_{\frac{d}{2}-1}(\kappa)}, \tag{57}$$

---

[7]Since there are also other tighter approximations of $I_\nu(\cdot)$, such as the uniform expansion in (Govindarajan et al., 2024) and DLMF (Lozier, 2003) Eq. 10.41.3: $I_\nu(\nu r) \sim \frac{e^{\nu\eta}}{\sqrt{2\pi\nu}\left(1+r^2\right)^{1/4}} \sum_{s=0}^\infty \frac{U_s(t)}{\nu^s}$, $\eta = \sqrt{1+r^2} + \log\frac{r}{1+\sqrt{1+r^2}}$, using our form is usually sufficient.

[8]This is because softmax function is shift invariant.

Substituting (57) into (56), and using $\|\boldsymbol{\mu}_p\|_2 = 1$, we obtain

$$\text{KL}(p\|q) = \log \frac{\mathcal{C}_d(\kappa_p)}{\mathcal{C}_d(\kappa_q)} + \kappa_p \boldsymbol{\mu}_p^\top (\mathcal{A}_d(\kappa_p)\boldsymbol{\mu}_p) - \kappa_q \boldsymbol{\mu}_q^\top (\mathcal{A}_d(\kappa_p)\boldsymbol{\mu}_p) \tag{58}$$

$$= \log \frac{\mathcal{C}_d(\kappa_p)}{\mathcal{C}_d(\kappa_q)} + \kappa_p \mathcal{A}_d(\kappa_p) (\boldsymbol{\mu}_p^\top \boldsymbol{\mu}_p) - \kappa_q \mathcal{A}_d(\kappa_p) (\boldsymbol{\mu}_q^\top \boldsymbol{\mu}_p) \tag{59}$$

$$= \log \frac{\mathcal{C}_d(\kappa_p)}{\mathcal{C}_d(\kappa_q)} + \kappa_p \mathcal{A}_d(\kappa_p) - \kappa_q \mathcal{A}_d(\kappa_p) \boldsymbol{\mu}_q^\top \boldsymbol{\mu}_p. \tag{60}$$

In our objective in Eq. (8), the anchor term uses $\text{KL}(\mathcal{V}_k'\|\mathcal{V}_k)$ with $\mathcal{V}_k' = (\boldsymbol{\mu}_k', \kappa_k')$ (anchor) and $\mathcal{V}_k = (\boldsymbol{\mu}_k, \kappa_k)$ (empirical estimate). Applying (60) to $p = \mathcal{V}_k'(\boldsymbol{\mu}_k', \kappa_k')$ and $q = \mathcal{V}_k(\boldsymbol{\mu}_k, \kappa_k)$ gives

$$\text{KL}(\mathcal{V}_k'\|\mathcal{V}_k) = \log \frac{\mathcal{C}_d(\kappa_k')}{\mathcal{C}_d(\kappa_k)} + \kappa_k' \mathcal{A}_d(\kappa_k') - \kappa_k \mathcal{A}_d(\kappa_k') \boldsymbol{\mu}_k^\top \boldsymbol{\mu}_k'. \tag{61}$$

Therefore, the anchor term $\mathcal{R}(\mathbf{M})$ is given by

$$\mathcal{R}(\mathbf{M}) = \sum_{k=1}^{K} \Big( \log \mathcal{C}_d(\kappa_k') - \log \mathcal{C}_d(\kappa_k) + \kappa_k' \mathcal{A}_d(\kappa_k') - \kappa_k \mathcal{A}_d(\kappa_k') \boldsymbol{\mu}_k^\top \boldsymbol{\mu}_k' \Big). \tag{62}$$

When optimizing w.r.t. $(\boldsymbol{\mu}_k, \kappa_k)$, the terms depending on $(\boldsymbol{\mu}_k, \kappa_k)$ reduce to

$$- \log \mathcal{C}_d(\kappa_k) \;-\; \kappa_k \mathcal{A}_d(\kappa_k') \boldsymbol{\mu}_k^\top \boldsymbol{\mu}_k', \tag{63}$$

up to constants independent of $(\boldsymbol{\mu}_k, \kappa_k)$.

## F. Derivation of the variable initialization of $\kappa$

In Eq. (7), we initialize $\boldsymbol{\mu}_k'$ from the zero-shot text prototype $\mathbf{t}_k$, and estimate $\mathcal{A}_d(\kappa_k')$ via a mean-squared-distance approximation on the unit hypersphere. We provide the detailed derivation of the latter in this section.

Given $\boldsymbol{\mu}_k'$ fixed, we compute the soft, class-weighted mean squared Euclidean distance $\text{MSE}_k$ as

$$\text{MSE}_k \triangleq \frac{\sum_{i=1}^{N} z_{i,k} \|\mathbf{f}_i - \boldsymbol{\mu}_k'\|_2^2}{\sum_{i=1}^{N} z_{i,k}} = \frac{\sum_{i=1}^{N} z_{i,k} \|\mathbf{f}_i - \boldsymbol{\mu}_k'\|_2^2}{N_k}, \qquad N_k \triangleq \sum_{i=1}^{N} z_{i,k}. \tag{64}$$

On the unit hypersphere, both $\mathbf{f}_i$ and $\boldsymbol{\mu}_k'$ are $\ell_2$-normalized, i.e., $\|\mathbf{f}_i\|_2 = \|\boldsymbol{\mu}_k'\|_2 = 1$. Hence,

$$\|\mathbf{f}_i - \boldsymbol{\mu}_k'\|_2^2 = \|\mathbf{f}_i\|_2^2 + \|\boldsymbol{\mu}_k'\|_2^2 - 2(\mathbf{f}_i^\top \boldsymbol{\mu}_k') = 2\big(1 - \cos \theta_i\big), \tag{65}$$

where $\cos \theta_i \triangleq \mathbf{f}_i^\top \boldsymbol{\mu}_k'$. Taking the weighted average over $i$ for class $k$ gives

$$\text{MSE}_k = 2\Big(1 - \mathbb{E}_k[f^\top \boldsymbol{\mu}_k']\Big), \qquad \mathbb{E}_k[f^\top \boldsymbol{\mu}_k'] \triangleq \frac{\sum_i z_{i,k} \mathbf{f}_i^\top \boldsymbol{\mu}_k'}{\sum_i z_{i,k}}. \tag{66}$$

Therefore,

$$\mathbb{E}_k[f^\top \boldsymbol{\mu}_k'] = 1 - \frac{\text{MSE}_k}{2}. \tag{67}$$

For a vMF distribution on $\mathbb{S}^{d-1}$ with density $p(f) = \mathcal{C}_d(\kappa) \exp(\kappa \mu^\top f)$, it is well-known that[9]

$$\mathbb{E}_{f \sim \text{vMF}(\mu, \kappa)}[\mu^\top f] = \mathcal{A}_d(\kappa) \triangleq \frac{I_{d/2}(\kappa)}{I_{d/2-1}(\kappa)}, \tag{68}$$

---

[9]For completeness, this follows from the partition function $Z(\kappa) = \int_{\mathbb{S}^{d-1}} \exp(\kappa \mu^\top f)\, df = 1/\mathcal{C}_d(\kappa)$ and the identity $\frac{\partial}{\partial \kappa} \log Z(\kappa) = \mathbb{E}[\mu^\top f]$, together with the standard Bessel recursion $I_\nu'(\kappa) = I_{\nu+1}(\kappa) + \frac{\nu}{\kappa} I_\nu(\kappa)$.

where $I_\nu(\cdot)$ is the modified Bessel function of the first kind.

Combining (67) and (68), we obtain the approximation

$$\mathcal{A}_d(\kappa'_k) \approx \mathbb{E}_k[f^\top \boldsymbol{\mu}'_k] = 1 - \frac{\mathrm{MSE}_k}{2} = 1 - \frac{\sum_i z_{i,k} \|\mathbf{f}_i - \boldsymbol{\mu}'_k\|_2^2}{2\sum_i z_{i,k}}. \tag{69}$$

Finally, $\kappa'_k$ is approximated as the inversion of $\mathcal{A}(\kappa_k)$ using Eq. (15).

## G. Derivations of the variable update

### G.1. With respect to assignments z

Following the derivations from TransCLIP (Zanella et al., 2024), we derive the update for assignments $\mathbf{z} = \{\mathbf{z}_i\}_{i=1}^N$ under the simplex constraint $\mathbf{z}_i \in \Delta^K$ with our instance-level, entropy-based weight $\gamma_i$. Note that this derivation is based on the setting in Eq. (9) that $\gamma_i$ is computed solely from the fixed pseudo label $\hat{\mathbf{y}}_i$. Although in actual implementation, we dynamically update $\gamma_i$ with current assignment $\mathbf{z}_i$, we treat it more as an engineering trick.

Fixing the distribution parameters $(\boldsymbol{\mu}_k, \kappa_k)$, the $\mathbf{z}$-dependent part from the objective in Eq. (8) can be written as

$$\min_{\mathbf{z}\in(\Delta^K)^N} \sum_{i=1}^N \gamma_i\Big(-\mathbf{z}_i^\top \log \mathbf{p}_i^{\mathrm{vMF}} + \mathrm{KL}(\mathbf{z}_i\|\hat{\mathbf{y}}_i)\Big) - \sum_{i,j}\gamma_i\omega_{ij}\,\mathbf{z}_i^\top \mathbf{z}_j. \tag{70}$$

Since $\omega_{ij} = \mathbf{f}_i^\top \mathbf{f}_j \geq 0$, the affinity matrix $\mathbf{W} = [\omega_{ij}] \in \mathbb{R}^{N\times N}$ is positive semi-definite (PSD). This makes the Laplacian term concave with respect to $\mathbf{z}$. Therefore, we adopt a BSUM-style approximation by taking the tight linear upper bound of $\mathbf{z}_i$ at iteration $t$.

**Constructing linear upper bound.** To construct such a linear upper bound, we first rewrite the Laplacian term in a matrix form. Let $\mathbf{z} \in \mathbb{R}^{NK}$ denote the concatenation of $\{\mathbf{z}_i\}_{i=1}^N$ (stacked by samples), and $\mathbf{G} = \mathrm{diag}(\boldsymbol{\gamma}) \in \mathbb{R}^{N\times N}$ be the diagonal matrix of instance-level weights $\boldsymbol{\gamma}$. Then, the weighted Laplacian term can be written as

$$-\sum_{i,j}\gamma_i\omega_{ij}\,\mathbf{z}_i^\top \mathbf{z}_j = -\sum_{i,j}(\mathbf{GW})_{ij}\,\mathbf{z}_i^\top \mathbf{z}_j = \mathbf{z}^\top \boldsymbol{\Psi}\,\mathbf{z}, \tag{71}$$

where

$$\boldsymbol{\Psi} \triangleq -(\mathbf{GW})\otimes \mathbf{I}_K, \tag{72}$$

$\otimes$ denotes the Kronecker product, and $\mathbf{I}_K$ is the $K \times K$ identity matrix. For notational simplicity, we use $\mathbf{I}_K$ since each $\mathbf{z}_i \in \mathbb{R}^K$. This is also the standard lifting used in TransCLIP and StatA.

When $\mathbf{W}$ is PSD and $\boldsymbol{\gamma} \succeq \mathbf{0}$, the symmetrized weight matrix $\frac{1}{2}\big(\mathbf{GW} + (\mathbf{GW})^\top\big)$ is also PSD[10], which implies that $\boldsymbol{\Psi}$ is negative semi-definite (NSD). As a results, $\mathbf{z}^\top \boldsymbol{\Psi}\mathbf{z}$ is concave with respect to $\mathbf{z}$.

For a concave quadratic function $q(\mathbf{z}) = \mathbf{z}^\top \boldsymbol{\Psi}\mathbf{z}$ with $\boldsymbol{\Psi} \preceq \mathbf{0}$, its first-order Taylor expansion at the current iterate $\mathbf{z}^{(t)}$ provides a *tight* global upper bound:

$$\mathbf{z}^\top \boldsymbol{\Psi}\mathbf{z} \leq (\mathbf{z}^{(t)})^\top \boldsymbol{\Psi}\mathbf{z}^{(t)} + \Big(\nabla q(\mathbf{z}^{(t)})\Big)^\top (\mathbf{z} - \mathbf{z}^{(t)}), \qquad \nabla q(\mathbf{z}) = (\boldsymbol{\Psi} + \boldsymbol{\Psi}^\top)\mathbf{z}. \tag{73}$$

In particular, if $\boldsymbol{\Psi}$ is symmetric (or using its symmetric part), the gradient simplifies to $\nabla q(\mathbf{z}) = 2\boldsymbol{\Psi}\mathbf{z}$, and the bound becomes

$$\mathbf{z}^\top \boldsymbol{\Psi}\mathbf{z} \leq (\mathbf{z}^{(t)})^\top \boldsymbol{\Psi}\mathbf{z}^{(t)} + 2(\boldsymbol{\Psi}\mathbf{z}^{(t)})^\top (\mathbf{z} - \mathbf{z}^{(t)}). \tag{74}$$

This upper bound is *tight* in the BSUM sense, i.e., it equals the original quadratic term at $\mathbf{z} = \mathbf{z}^{(t)}$. Moreover, replacing the quadratic coupling by (73) or (74) yields a linear surrogate that decouples across $\{\mathbf{z}_i\}$ under simplex constraints, enabling an efficient BSUM update.

---

[10]Equivalently, one may replace $\mathbf{GW}$ by its symmetric part without changing the scalar form of the quadratic term, since $\mathbf{z}^\top \mathbf{A}\mathbf{z} = \mathbf{z}^\top \frac{\mathbf{A}+\mathbf{A}^\top}{2}\mathbf{z}$ for any square matrix $\mathbf{A}$.

Therefore, by fixing the neighbors $\{\mathbf{z}_j^{(t)}\}_j$ and upper-bounding the bilinear term, we rewrite the Laplacian term as

$$-\sum_{i,j}\gamma_i\omega_{ij}\,\mathbf{z}_i^\top\mathbf{z}_j \;\approx\; -\sum_{i,j}\gamma_i\omega_{ij}\,\mathbf{z}_i^\top\mathbf{z}_j^{(t)} \;+\; \mathrm{const} = -\sum_i\gamma_i\mathbf{z}_i^\top\sum_j\omega_{ij}\mathbf{z}_j^{(t)} \;+\; \mathrm{const} \tag{75}$$

Substituting (75) into (70), the problem becomes separable over $i$.

**Per-sample subproblem and cancellation of $\gamma_i$.** For each sample $i$, we obtain the subproblem

$$\mathbf{z}_i^{(t+1)} \in \arg\min_{\mathbf{z}_i\in\Delta^K}\; \gamma_i\Big( -\mathbf{z}_i^\top\log\mathbf{p}_i^{\mathrm{vMF}} + \mathrm{KL}(\mathbf{z}_i\|\hat{\mathbf{y}}_i) - \mathbf{z}_i^\top\sum_j\omega_{ij}\,\mathbf{z}_j^{(t)}\Big). \tag{76}$$

Since $\gamma_i > 0$ is a constant multiplier in (76), it does not affect the minimizer:

$$\arg\min_{\mathbf{z}_i\in\Delta^K}\gamma_i\,g_i(\mathbf{z}_i) \;=\; \arg\min_{\mathbf{z}_i\in\Delta^K}g_i(\mathbf{z}_i), \qquad (\gamma_i > 0), \tag{77}$$

which explains why the assignment update in Eq. (12) does not explicitly depend on $\gamma_i$.

**Closed-form update.** Expanding $\mathrm{KL}(\mathbf{z}_i\|\hat{\mathbf{y}}_i) = \sum_k z_{i,k}\log z_{i,k} - z_{i,k}\log\hat{y}_{i,k}$ and omitting $\mathbf{z}_i$-independent constants, the effective subproblem is

$$\min_{\mathbf{z}_i\in\Delta^K}\;\sum_{k=1}^{K}z_{i,k}\log z_{i,k} - \sum_{k=1}^{K}z_{i,k}\,s_{i,k}^{(t)}, \quad s_{i,k}^{(t)} \triangleq \log\hat{y}_{i,k} + \log p_{i,k}^{\mathrm{vMF}} + \sum_j\omega_{ij}\,z_{j,k}^{(t)}. \tag{78}$$

Introducing a Lagrange multiplier $\lambda_i$ (different from the class confidence in Eq. (11)) for constraint $\sum_k z_{i,k} = 1$, solving the Karush-Kuhn-Tucker (KKT) conditions yields

$$\log z_{i,k} + 1 - s_{i,k}^{(t)} + \lambda_i = 0 \quad\Longrightarrow\quad z_{i,k} \propto \exp\big(s_{i,k}^{(t)}\big). \tag{79}$$

Since $\mathbf{z}_i\in\Delta_k$, we obtain the final form after normalization

$$z_{i,k}^{(t+1)} = \frac{\exp\big(s_{i,k}^{(t)}\big)}{\sum_{r=1}^{K}\exp\big(s_{i,r}^{(t)}\big)} = \frac{\hat{y}_{i,k}\exp\Big(\log p_{i,k}^{\mathrm{vMF}} + \sum_j\omega_{ij}z_{j,k}^{(t)}\Big)}{\sum_{r=1}^{K}\hat{y}_{i,r}\exp\Big(\log p_{i,r} + \sum_j\omega_{ij}z_{j,r}^{(t)}\Big)}. \tag{80}$$

Expressed in vector form, Eq. (80) is given by

$$\mathbf{z}_i^{(t+1)} = \frac{\hat{\mathbf{y}}_i\odot\exp(\log\mathbf{p}_i^{\mathrm{vMF}} + \sum_j\omega_{ij}\mathbf{z}_j^{(t)})}{(\hat{\mathbf{y}}_i\odot\exp(\log\mathbf{p}_i^{\mathrm{vMF}} + \sum_j\omega_{ij}\mathbf{z}_j^{(t)}))^\top\mathbb{1}_K}. \tag{81}$$

This has the same form as StatA, with $\log p_{i,k}^{\mathrm{vMF}} = \log\mathcal{C}_d(\kappa_k) + \kappa_k\boldsymbol{\mu}_k^\top\mathbf{f}_i$ instantiated by our vMF likelihood in Eq. (6).

### G.2. With respect to parameters $\boldsymbol{\mu}$ and $\kappa$

In this subsection, we derive the closed-form updates for the vMF parameters $\{\boldsymbol{\mu}_k, \kappa_k\}_{k=1}^{K}$ by fixing the soft assignments $\mathbf{z}$. Recall that all feature vectors are $\ell_2$-normalized, i.e., $\|\mathbf{f}_i\|_2 = 1$.

**Update of $\boldsymbol{\mu}_k$.** We first consider the mean direction vector $\boldsymbol{\mu}_k$. Collecting all terms in the PLE objective Eq. (8) that depend on $\boldsymbol{\mu}_k$, we obtain

$$J(\boldsymbol{\mu}_k) = -\sum_i\gamma_i\mathbf{z}_{i,k}(\kappa_k\boldsymbol{\mu}_k^\top\mathbf{f}_i) + \alpha(\kappa_k\mathcal{A}_d(\kappa_k')\boldsymbol{\mu}_k^\top\boldsymbol{\mu}_k'), \quad \text{s.t. } \|\boldsymbol{\mu}_k\|_2 = 1. \tag{82}$$

Minimizing this objective is equivalent to maximizing

$$\max_{\boldsymbol{\mu}_k}\;\kappa_k\boldsymbol{\mu}_k^\top\left(\sum_i\gamma_i\mathbf{z}_{i,k}\mathbf{f}_i + \alpha\mathcal{A}_d(\kappa_k')\boldsymbol{\mu}_k'\right), \quad \text{s.t. } \|\boldsymbol{\mu}_k\|_2 = 1, \tag{83}$$

Let

$$R_k \triangleq \sum_i \gamma_i \mathbf{z}_{i,k} \mathbf{f}_i, \qquad T_k \triangleq \alpha \mathcal{A}_d(\kappa_k') \boldsymbol{\mu}_k', \tag{84}$$

and define the combined resultant vector

$$R_k^{\mathrm{tot}} \triangleq R_k + T_k. \tag{85}$$

Then, the objective in (83) reduces to maximizing $\boldsymbol{\mu}_k^\top R_k^{\mathrm{tot}}$ under a unit-norm constraint. The optimum is achieved when $\boldsymbol{\mu}_k$ aligns with $R_k^{\mathrm{tot}}$, yielding

$$\boldsymbol{\mu}_k = \frac{R_k^{\mathrm{tot}}}{\|R_k^{\mathrm{tot}}\|_2} = \frac{\sum_i \gamma_i \mathbf{z}_{i,k} \mathbf{f}_i + \alpha \mathcal{A}_d(\kappa_k') \boldsymbol{\mu}_k'}{\|\sum_i \gamma_i \mathbf{z}_{i,k} \mathbf{f}_i + \alpha \mathcal{A}_d(\kappa_k') \boldsymbol{\mu}_k'\|_2}. \tag{86}$$

**Update of $\kappa_k$.**  Considering concentration parameter $\kappa_k$. Similarly, when fixing $\boldsymbol{\mu}_k$, the terms involving $\kappa_k$ in Eq. (8) can be written as

$$\mathcal{L}(\kappa_k) = -(N_k + \alpha) \log \mathcal{C}_d(\kappa_k) - \kappa_k \boldsymbol{\mu}_k^\top \left( \sum_i \gamma_i \mathbf{z}_{i,k} \mathbf{f}_i + \alpha \mathcal{A}_d(\kappa_k') \boldsymbol{\mu}_k' \right), \tag{87}$$

where $N_k = \sum_i \gamma_i \mathbf{z}_{i,k}$ denotes the soft assignment count (with weights $\gamma_i$).

From the update of $\boldsymbol{\mu}_k$ in (86), the direction of $\boldsymbol{\mu}_k$ coincides with that of $R_k^{\mathrm{tot}}$, and thus

$$\boldsymbol{\mu}_k^\top R_k^{\mathrm{tot}} = \|R_k^{\mathrm{tot}}\|_2. \tag{88}$$

Denoting $\bar{R}_k^{\mathrm{tot}} \triangleq \|R_k^{\mathrm{tot}}\|_2$, the objective (87) becomes

$$\mathcal{L}(\kappa_k) = -(N_k + \alpha) \log \mathcal{C}_d(\kappa_k) - \kappa_k \bar{R}_k^{\mathrm{tot}}. \tag{89}$$

Taking the derivative with respect to $\kappa_k$ and setting it to zero yields

$$\frac{\partial L(\kappa_k)}{\partial \kappa} = -(N_k + \alpha) \frac{\partial}{\partial \kappa} \log \mathcal{C}_d(\kappa_k) - \frac{\partial}{\partial \kappa} \kappa_k \bar{R}_k^{\mathrm{tot}} \tag{90}$$

$$= -(N_k + \alpha) \frac{\partial}{\partial \kappa} \log \mathcal{C}_d(\kappa_k) - \bar{R}_k^{\mathrm{tot}} \tag{91}$$

$$= (N_k + \alpha) \mathcal{A}_d(\kappa) - \bar{R}_k^{\mathrm{tot}} \tag{92}$$

$$= 0, \tag{93}$$

where we used the standard vMF identity $\frac{\partial}{\partial \kappa} \log \mathcal{C}_d(\kappa) = -\mathcal{A}_d(\kappa)$. Therefore, $\kappa_k$ satisfies

$$\mathcal{A}_d(\kappa_k) = \frac{\bar{R}_k^{\mathrm{tot}}}{N_k + \alpha} = \frac{\|\sum_i \gamma_i \mathbf{z}_{i,k} \mathbf{f}_i + \alpha \mathcal{A}_d(\kappa_k') \boldsymbol{\mu}_k'\|_2}{\sum_i \gamma_i \mathbf{z}_{i,k} + \alpha}. \tag{94}$$

Then, $\kappa_k$ is approximated as the inverse of $\mathcal{A}_d(\kappa_k)$ using Eq. (15).

## H. Equivalence to the Adaptive Shrinkage Form for Parameter Updates

In this section, we show that the parameter updates in Eq. (13) can be written in an equivalent *adaptive shrinkage form*, i.e., Eq. (14). For class $k$, define

$$N_k \triangleq \sum_i \gamma_i \mathbf{z}_{i,k}, \qquad R_k \triangleq \sum_i \gamma_i \mathbf{z}_{i,k} \mathbf{f}_i, \qquad \boldsymbol{v}_k \triangleq \frac{R_k}{N_k}, \qquad \beta_k \triangleq \frac{N_k}{N_k + \alpha}. \tag{95}$$

Here, we set $\alpha$ as a fixed scalar for simplicity.

**Remark.**  To make the equivalence more transparent, we first present the proof under the mild simplification $\mathcal{A}_d(\kappa_k') = 1$, i.e., distribution anchor provides a deterministic direction (which is intuitive and natural). The same derivation holds for general $\mathcal{A}_d(\kappa_k') \neq 1$ by replacing $\alpha$ with $\alpha \mathcal{A}_d(\kappa_k')$.

**With respect to $\boldsymbol{\mu}_k$.**    Let

$$V_k^{\text{ours}} \triangleq R_k + \alpha \boldsymbol{\mu}_k' \;=\; \sum_i \gamma_i \mathbf{z}_{i,k} \mathbf{f}_i + \alpha \boldsymbol{\mu}_k', \qquad V_k^{\text{StatA}} \triangleq \beta_k \boldsymbol{v}_k + (1-\beta_k)\boldsymbol{\mu}_k'. \tag{96}$$

Eq. (13) updates $\boldsymbol{\mu}_k$ by normalizing $V_k^{\text{ours}}$, i.e.,

$$\boldsymbol{\mu}_k = \frac{V_k^{\text{ours}}}{\|V_k^{\text{ours}}\|_2}. \tag{97}$$

We now show that this is equivalent to normalizing $V_k^{\text{StatA}}$. Indeed, substituting the definitions in (95) yields

$$V_k^{\text{StatA}} = \left(\frac{N_k}{N_k + \alpha}\right)\frac{R_k}{N_k} + \left(\frac{\alpha}{N_k + \alpha}\right)\boldsymbol{\mu}_k' \tag{98}$$

$$= \frac{R_k}{N_k + \alpha} + \frac{\alpha\boldsymbol{\mu}_k'}{N_k + \alpha} \tag{99}$$

$$= \frac{1}{N_k + \alpha}\left(R_k + \alpha\boldsymbol{\mu}_k'\right) \tag{100}$$

$$= \frac{1}{N_k + \alpha}V_k^{\text{ours}}. \tag{101}$$

Therefore, $V_k^{\text{StatA}}$ and $V_k^{\text{ours}}$ are colinear and share the same direction. After normalization, we obtain the exact equivalence

$$\frac{V_k^{\text{ours}}}{\|V_k^{\text{ours}}\|_2} \;\equiv\; \frac{V_k^{\text{StatA}}}{\|V_k^{\text{StatA}}\|_2} = \frac{\beta_k \boldsymbol{v}_k + (1-\beta_k)\boldsymbol{\mu}_k'}{\|\beta_k \boldsymbol{v}_k + (1-\beta_k)\boldsymbol{\mu}_k'\|_2}, \tag{102}$$

which proves the equivalence between Eq. (13) and (14).

**With respect to $\kappa_k$.**    Eq. (13) computes

$$\mathcal{A}_d(\kappa_k) = \frac{\|R_k + \alpha\boldsymbol{\mu}_k'\|_2}{N_k + \alpha} = \frac{\|V_k^{\text{ours}}\|_2}{N_k + \alpha}. \tag{103}$$

According to Eq. (101), we have

$$\|\beta_k \boldsymbol{v}_k + (1-\beta_k)\boldsymbol{\mu}_k'\|_2 = \left\|V_k^{\text{StatA}}\right\|_2 = \left\|\frac{1}{N_k + \alpha}V_k^{\text{ours}}\right\|_2 \tag{104}$$

$$= \frac{1}{N_k + \alpha}\left\|V_k^{\text{ours}}\right\|_2 \tag{105}$$

$$= \frac{\|R_k + \alpha\boldsymbol{\mu}_k'\|_2}{N_k + \alpha} \tag{106}$$

$$= \mathcal{A}_d(\kappa_k). \tag{107}$$

Thus, the concentration update in Eq. (13) is also exactly equivalent to the form in Eq. (14).

## I. Additional Experimental Results

### I.1. Results with different batch sizes

To verify the robustness of MOON concerning data availability, we conduct experiments with varying batch sizes as reported in Tab. 11. Across three different batch sizes and two realistic scenarios, our method consistently outperforms all baselines and significantly improves upon the zero-shot CLIP initialization. This indicates the effectiveness of MOON, regardless of whether the incoming data batch is sparse or abundant, ensuring reliable adaptation performance across different data scales.

*Table 11.* **Results with different batch sizes.** The best and second-best results are marked in **bold** and underlined, respectively. Subscript green indicates improvement, red indicates decline, and gray indicates no change compared with zero-shot performance.

(a) Batch Size: 128.

| $K_{\text{eff}}$ | Method | ImageNet | SUN397 | Aircraft | EuroSAT | StanfordCars | Food101 | Pets | Flowers102 | Caltech101 | DTD | UCF101 | Avg. |
|---|---|---|---|---|---|---|---|---|---|---|---|---|---|
| | CLIP | 66.6 | 62.5 | 24.7 | 48.3 | 65.6 | 85.9 | 89.1 | 70.7 | 93.2 | 43.5 | 67.5 | 65.2 |
| Medium | StatA | $72.0_{+5.4}$ | $66.5_{+4.0}$ | $26.6_{+1.9}$ | $48.2_{-0.1}$ | $72.7_{+7.1}$ | $88.7_{+2.8}$ | $92.1_{+3.0}$ | $75.4_{+4.7}$ | $93.7_{+0.5}$ | $47.1_{+4.2}$ | $70.0_{+2.5}$ | $68.5_{+3.3}$ |
| | MOON | $82.3_{+15.7}$ | $76.2_{+13.7}$ | $27.1_{+2.4}$ | $46.2_{-2.1}$ | $75.7_{+10.1}$ | $93.8_{+7.9}$ | $92.1_{+3.0}$ | $75.9_{+5.2}$ | $94.8_{+1.6}$ | $47.1_{+3.6}$ | $72.9_{+5.4}$ | $\mathbf{71.3}_{+6.1}$ |
| High | StatA | $69.4_{+2.8}$ | $64.9_{+2.4}$ | $23.6_{-1.1}$ | $47.2_{-1.1}$ | $68.0_{+2.4}$ | $87.0_{+1.1}$ | $88.2_{-0.9}$ | $72.0_{+1.3}$ | $94.0_{+0.8}$ | $46.9_{+3.4}$ | $68.2_{+0.7}$ | $66.3_{+1.1}$ |
| | MOON | $76.4_{+9.8}$ | $69.6_{+7.1}$ | $21.7_{-3.0}$ | $46.2_{-2.1}$ | $68.6_{+3.0}$ | $88.9_{+3.0}$ | $86.4_{-2.7}$ | $71.3_{+0.6}$ | $93.7_{+0.5}$ | $40.2_{-3.3}$ | $67.1_{-0.4}$ | $\mathbf{66.4}_{+1.1}$ |

(b) Batch Size: 256.

| $K_{\text{eff}}$ | Method | ImageNet | SUN397 | Aircraft | EuroSAT | StanfordCars | Food101 | Pets | Flowers102 | Caltech101 | DTD | UCF101 | Avg. |
|---|---|---|---|---|---|---|---|---|---|---|---|---|---|
| | CLIP | 66.6 | 62.5 | 24.7 | 48.3 | 65.6 | 85.9 | 89.1 | 70.7 | 93.2 | 43.5 | 67.5 | 65.2 |
| Medium | StatA | $72.0_{+5.4}$ | $66.7_{+4.2}$ | $27.1_{+2.4}$ | $56.0_{+7.7}$ | $74.1_{+8.5}$ | $88.9_{+3.0}$ | $92.9_{+3.8}$ | $76.0_{+5.3}$ | $93.6_{+0.4}$ | $47.0_{+3.5}$ | $70.5_{+3.0}$ | $69.5_{+4.3}$ |
| | MOON | $82.9_{+16.3}$ | $77.8_{+15.3}$ | $27.9_{+3.2}$ | $49.4_{+1.1}$ | $76.6_{+11.0}$ | $94.1_{+8.2}$ | $92.9_{+3.8}$ | $76.1_{+5.4}$ | $95.2_{+2.0}$ | $48.4_{+4.9}$ | $73.7_{+6.2}$ | $\mathbf{72.3}_{+7.0}$ |
| High | StatA | $71.1_{+4.5}$ | $66.3_{+3.8}$ | $24.2_{-0.5}$ | $55.5_{+7.2}$ | $70.6_{+5.0}$ | $87.6_{+1.7}$ | $88.9_{-0.2}$ | $73.7_{+3.0}$ | $94.1_{+0.9}$ | $47.0_{+3.5}$ | $69.9_{+2.4}$ | $68.1_{+2.9}$ |
| | MOON | $80.5_{+13.9}$ | $73.2_{+10.7}$ | $23.2_{-1.5}$ | $49.1_{+0.8}$ | $71.8_{+6.2}$ | $90.4_{+4.5}$ | $88.2_{-0.9}$ | $73.1_{+2.4}$ | $94.1_{+0.9}$ | $42.7_{-0.8}$ | $69.6_{+2.1}$ | $\mathbf{68.7}_{+3.5}$ |

(c) Batch Size: 500.

| $K_{\text{eff}}$ | Method | ImageNet | SUN397 | Aircraft | EuroSAT | StanfordCars | Food101 | Pets | Flowers102 | Caltech101 | DTD | UCF101 | Avg. |
|---|---|---|---|---|---|---|---|---|---|---|---|---|---|
| | CLIP | 66.6 | 62.5 | 24.7 | 48.3 | 65.6 | 85.9 | 89.1 | 70.7 | 93.2 | 43.5 | 67.5 | 65.2 |
| Medium | StatA | $71.5_{+4.9}$ | $65.5_{+3.0}$ | $27.8_{+3.1}$ | $59.3_{+11.0}$ | $74.9_{+9.3}$ | $88.3_{+2.4}$ | $93.1_{+4.0}$ | $76.8_{+6.1}$ | $93.1_{-0.1}$ | $47.1_{+3.6}$ | $69.9_{+2.4}$ | $69.8_{+4.5}$ |
| | MOON | $82.4_{+15.8}$ | $76.7_{+14.2}$ | $28.6_{+3.9}$ | $55.5_{+7.2}$ | $77.0_{+11.4}$ | $93.7_{+7.8}$ | $92.7_{+3.6}$ | $76.5_{+5.8}$ | $95.0_{+1.8}$ | $49.0_{+5.5}$ | $73.4_{+5.9}$ | $\mathbf{72.8}_{+7.5}$ |
| High | StatA | $72.1_{+5.5}$ | $67.3_{+4.8}$ | $25.1_{+0.4}$ | $60.0_{+11.7}$ | $72.3_{+6.7}$ | $88.2_{+2.3}$ | $90.3_{+1.2}$ | $75.5_{+4.8}$ | $93.8_{+0.6}$ | $47.2_{+3.7}$ | $70.7_{+3.2}$ | $69.3_{+4.1}$ |
| | MOON | $82.3_{+15.7}$ | $75.5_{+13.0}$ | $24.3_{-0.4}$ | $55.8_{+7.5}$ | $73.8_{+8.2}$ | $91.2_{+5.3}$ | $89.3_{+0.2}$ | $74.5_{+3.8}$ | $94.7_{+1.5}$ | $44.1_{+0.6}$ | $70.8_{+3.3}$ | $\mathbf{70.6}_{+5.3}$ |

*Table 12.* **Overview of the VLMs employed in our experiments.**

| Model name | Full name | #Param | Detailed #Param | FLOPS (B) | Resolution | Pretrained |
|---|---|---|---|---|---|---|
| CLIP | clip-resnet50 / RN50 | 102M | 102,007,137 | 18.18 | 224x224 | WIT |
| CLIP | clip-resnet101 / RN101 | 120M | 119,688,033 | 25.50 | 224x224 | WIT |
| CLIP | clip-vit-base-patch16 / ViT-B/16 | 150M | 149,620,737 | 41.09 | 224x224 | WIT |
| CLIP | clip-vit-base-patch32 / ViT-B/32 | 151M | 151,277,313 | 14.78 | 224x224 | WIT |
| CLIP | clip-vit-large-patch14 / ViT-L/14 | 428M | 427,616,513 | 175.33 | 224x224 | WIT |
| OpenCLIP | ViT-B-16 | 150M | 149,620,737 | 41.09 | 224x224 | DataComp |
| SigLIP | ViT-SO400M-14-SigLIP-384 | 878M | 877,960,498 | 723.48 | 384x384 | WebLI |
| EVA-CLIP | EVA01-g-14 | 1.14B | 1,136,435,841 | 547.36 | 224x224 | LAION |

### I.2. Results with other backbones

Following the discussion in Sec. 5.3, we extend our evaluation to four additional CLIP visual backbones, including ResNet (He et al., 2016) architectures (RN50, RN101) and Vision Transformers (ViT) (Dosovitskiy et al., 2021) of varying scales (ViT-B/32, ViT-L/14), as detailed in Tab. 13, 14 and 15. These experiments cover the full range of realistic batch and online adaptation settings. MOON demonstrates remarkable universality, achieving the highest accuracy in nearly all evaluated scenarios (winning in 33 out of 36 cases) with only negligible margins in the few exceptions. This consistent superiority across diverse architectures and model capacities confirms that our vMF-based dynamic shrinkage mechanism is a generalized solution, capable of enhancing VLM performance regardless of the specific underlying visual encoder.

### I.3. Results with other VLM architectures

To validate the generalizability of our MOON beyond standard CLIP models, we evaluate three additional VLMs with diverse architectures, parameter scales, and training procedures: OpenCLIP (151M) (Cherti et al., 2023), SigLIP (878M) (Zhai et al., 2023), and EVA-CLIP (1.1B) (Sun et al., 2023), all implemented using the OpenCLIP codebase (Ilharco et al., 2021). Details regarding the VLMs in our experiment is presented in Tab. 12. Experiments conducted with a batch size of 1,000 across three realistic scenarios (Tables 16, 17, and 18) reveal that MOON consistently enhances zero-shot performance and achieves state-of-the-art results in nearly all nine settings. Notably, the relative improvements are particularly distinct for smaller models (e.g., OpenCLIP), suggesting our method effectively compensates for weaker initial representations, while maintaining substantial gains on challenging large-scale datasets like ImageNet and SUN397. Collectively, these findings demonstrate the universal effectiveness of MOON across diverse model architectures, scales, and pre-training paradigms.

### I.4. Results on full dataset with all classes

We further evaluate the extreme scenario where the model adapts to the full dataset containing all classes simultaneously, representing a dense label distribution that deviates from our sparsity assumption. As shown in Tab. 19, MOON trails the strongest baseline StatA by a marginal gap ($\sim$2%). This behavior is consistent with our analysis in the main text: our dynamic shrinkage mechanism is inherently designed with an inductive bias to favor sparse effective class sets, which is less optimal when the ground-truth distribution is uniform. However, unlike many specialized methods that collapse when assumptions are violated, MOON maintains competitive high-accuracy performance without severe degradation. Considering its significant computational efficiency, our method offers a robust trade-off, serving as a reliable solution even in scenarios with maximal class presence.

### I.5. Results on random class scenarios

To simulate a highly unpredictable deployment environment, we introduce a challenging "Random" scenario where the number of effective classes $K_{eff}$ varies stochastically between 1 and $\min\{N, K\}$ for each adaptation step. This setting effectively aggregates the characteristics of varying sparsity levels into a single dynamic evaluation, with results demonstrated in Tab. 20. Under these volatile conditions, MOON exhibits remarkable stability, achieving performance comparable to the state-of-the-art StatA across varying batch sizes. The fact that our method matches the strongest baseline in accuracy while operating with significantly lower latency and a simpler optimization procedure (as detailed in Appendix A) further underscores its practicality for handling real-world data streams with unknown and fluctuating statistics.

*Table 13.* **Results on four additional CLIP backbones, batch adaptation with batch size of 64.** Subscript green indicates improvement, red indicates decline, and gray indicates no change compared with zero-shot performance.

### (a) ResNet-50.

| $K_{\text{eff}}$ | Method | ImageNet | SUN397 | Aircraft | EuroSAT | StanfordCars | Food101 | Pets | Flowers102 | Caltech101 | DTD | UCF101 | Avg. |
|---|---|---|---|---|---|---|---|---|---|---|---|---|---|
| | CLIP | 58.2 | 58.9 | 17.0 | 36.2 | 55.8 | 77.4 | 85.7 | 66.1 | 85.7 | 42.8 | 61.8 | 58.7 |
| Very Low | StatA | 68.2$_{+10.0}$ | 63.7$_{+4.8}$ | 21.1$_{+4.1}$ | 43.3$_{+7.1}$ | 71.3$_{+15.5}$ | 87.1$_{+9.7}$ | 93.1$_{+7.4}$ | 74.1$_{+8.0}$ | 90.1$_{+4.4}$ | 45.3$_{+2.5}$ | 66.2$_{+4.4}$ | 65.8$_{+7.1}$ |
| | MOON | 79.7$_{+21.5}$ | 77.5$_{+18.6}$ | 25.4$_{+8.4}$ | 41.8$_{+5.6}$ | 74.8$_{+19.0}$ | 94.9$_{+17.5}$ | 95.5$_{+9.8}$ | 75.2$_{+9.1}$ | 92.2$_{+6.5}$ | 54.4$_{+11.6}$ | 74.1$_{+12.3}$ | **71.4**$_{+12.7}$ |
| Low | StatA | 65.0$_{+6.8}$ | 62.8$_{+3.9}$ | 17.8$_{+0.8}$ | 31.7$_{-4.5}$ | 67.1$_{+11.3}$ | 83.6$_{+6.2}$ | 88.2$_{+2.5}$ | 71.7$_{+5.6}$ | 89.0$_{+3.3}$ | 44.9$_{+2.1}$ | 64.0$_{+2.2}$ | 62.3$_{+3.7}$ |
| | MOON | 79.5$_{+21.3}$ | 77.9$_{+19.0}$ | 21.9$_{+4.9}$ | 35.2$_{-1.0}$ | 72.8$_{+17.0}$ | 92.9$_{+15.5}$ | 94.0$_{+8.3}$ | 75.6$_{+9.5}$ | 91.7$_{+6.0}$ | 49.2$_{+6.4}$ | 71.6$_{+9.8}$ | **69.3**$_{+10.6}$ |
| Medium | StatA | 61.1$_{+2.9}$ | 59.5$_{+0.6}$ | 16.4$_{-0.6}$ | 27.3$_{-8.9}$ | 62.1$_{+6.3}$ | 78.7$_{+1.3}$ | 81.4$_{-4.3}$ | 64.1$_{-2.0}$ | 87.9$_{+2.2}$ | 43.8$_{+1.0}$ | 62.0$_{+0.2}$ | 58.6$_{-0.1}$ |
| | MOON | 73.6$_{+15.4}$ | 71.3$_{+12.4}$ | 18.6$_{+1.6}$ | 31.9$_{-4.3}$ | 67.2$_{+11.4}$ | 87.6$_{+10.2}$ | 88.5$_{+2.8}$ | 70.5$_{+4.4}$ | 89.8$_{+4.1}$ | 42.2$_{-0.6}$ | 67.5$_{+5.7}$ | **64.4**$_{+5.7}$ |

### (b) ResNet-101.

| $K_{\text{eff}}$ | Method | ImageNet | SUN397 | Aircraft | EuroSAT | StanfordCars | Food101 | Pets | Flowers102 | Caltech101 | DTD | UCF101 | Avg. |
|---|---|---|---|---|---|---|---|---|---|---|---|---|---|
| | CLIP | 61.3 | 59.0 | 17.9 | 32.9 | 63.2 | 80.7 | 86.9 | 64.3 | 89.9 | 37.3 | 61.1 | 59.5 |
| Very Low | StatA | 73.0$_{+11.7}$ | 66.5$_{+7.5}$ | 22.5$_{+4.6}$ | 30.4$_{-2.5}$ | 76.2$_{+13.0}$ | 89.5$_{+8.8}$ | 95.2$_{+8.3}$ | 74.6$_{+10.3}$ | 91.9$_{+2.0}$ | 42.9$_{+5.6}$ | 65.1$_{+4.0}$ | 66.2$_{+6.7}$ |
| | MOON | 80.0$_{+18.7}$ | 74.5$_{+15.5}$ | 24.8$_{+6.9}$ | 30.8$_{-2.1}$ | 77.9$_{+14.7}$ | 94.1$_{+13.4}$ | 93.9$_{+7.0}$ | 72.4$_{+8.1}$ | 93.2$_{+3.3}$ | 46.1$_{+8.8}$ | 70.3$_{+9.2}$ | **68.9**$_{+9.4}$ |
| Low | StatA | 71.2$_{+9.9}$ | 65.9$_{+6.9}$ | 20.0$_{+2.1}$ | 29.6$_{-3.3}$ | 73.1$_{+9.9}$ | 88.1$_{+7.4}$ | 92.9$_{+6.0}$ | 74.9$_{+10.6}$ | 92.8$_{+2.9}$ | 42.9$_{+5.6}$ | 64.4$_{+3.3}$ | 65.1$_{+5.6}$ |
| | MOON | 79.8$_{+18.5}$ | 74.5$_{+15.5}$ | 21.8$_{+3.9}$ | 30.0$_{-2.9}$ | 76.4$_{+13.2}$ | 93.1$_{+12.4}$ | 92.5$_{+5.6}$ | 72.6$_{+8.3}$ | 94.0$_{+4.1}$ | 42.6$_{+5.3}$ | 68.4$_{+7.3}$ | **67.8**$_{+8.3}$ |
| Medium | StatA | 67.0$_{+5.7}$ | 62.7$_{+3.7}$ | 18.6$_{+0.7}$ | 28.7$_{-4.2}$ | 69.6$_{+6.4}$ | 84.9$_{+4.2}$ | 88.8$_{+1.9}$ | 70.1$_{+5.8}$ | 92.1$_{+2.2}$ | 40.9$_{+3.6}$ | 63.6$_{+2.5}$ | 62.5$_{+3.0}$ |
| | MOON | 74.5$_{+13.2}$ | 69.1$_{+10.1}$ | 19.7$_{+1.8}$ | 29.9$_{-3.0}$ | 72.7$_{+9.5}$ | 89.0$_{+8.3}$ | 89.0$_{+2.1}$ | 68.2$_{+3.9}$ | 92.6$_{+2.7}$ | 37.5$_{+0.2}$ | 65.3$_{+4.2}$ | **64.3**$_{+4.8}$ |

### (c) ViT-B/32.

| $K_{\text{eff}}$ | Method | ImageNet | SUN397 | Aircraft | EuroSAT | StanfordCars | Food101 | Pets | Flowers102 | Caltech101 | DTD | UCF101 | Avg. |
|---|---|---|---|---|---|---|---|---|---|---|---|---|---|
| | CLIP | 62.0 | 62.1 | 19.1 | 45.4 | 60.2 | 80.4 | 87.3 | 66.6 | 91.4 | 42.7 | 63.5 | 61.9 |
| Very Low | StatA | 68.1$_{+6.1}$ | 65.6$_{+3.5}$ | 23.0$_{+3.9}$ | 53.2$_{+7.8}$ | 71.9$_{+11.7}$ | 85.8$_{+5.4}$ | 94.3$_{+7.0}$ | 74.9$_{+8.3}$ | 93.4$_{+2.0}$ | 45.2$_{+2.5}$ | 64.5$_{+1.0}$ | 67.3$_{+5.4}$ |
| | MOON | 80.9$_{+18.9}$ | 78.4$_{+16.3}$ | 25.7$_{+6.6}$ | 52.2$_{+6.8}$ | 77.8$_{+17.6}$ | 94.8$_{+14.4}$ | 94.7$_{+7.4}$ | 75.4$_{+8.8}$ | 95.1$_{+3.7}$ | 53.6$_{+10.9}$ | 70.4$_{+6.9}$ | **72.6**$_{+10.7}$ |
| Low | StatA | 67.2$_{+5.2}$ | 65.7$_{+3.6}$ | 21.9$_{+2.8}$ | 50.1$_{+4.7}$ | 69.3$_{+9.1}$ | 84.5$_{+4.1}$ | 92.5$_{+5.2}$ | 75.3$_{+8.7}$ | 93.2$_{+1.8}$ | 46.1$_{+3.4}$ | 64.6$_{+1.1}$ | 66.4$_{+4.5}$ |
| | MOON | 80.3$_{+18.3}$ | 78.8$_{+16.7}$ | 23.8$_{+4.7}$ | 45.6$_{+0.2}$ | 77.0$_{+16.8}$ | 93.7$_{+13.3}$ | 93.1$_{+5.8}$ | 76.4$_{+9.8}$ | 94.8$_{+3.4}$ | 47.9$_{+5.2}$ | 69.6$_{+6.1}$ | **71.0**$_{+9.1}$ |
| Medium | StatA | 65.5$_{+3.5}$ | 64.8$_{+2.7}$ | 20.0$_{+0.9}$ | 45.4$_{\pm0.0}$ | 65.0$_{+4.8}$ | 82.6$_{+2.2}$ | 89.4$_{+2.1}$ | 70.6$_{+4.0}$ | 92.9$_{+1.5}$ | 46.7$_{+4.0}$ | 64.4$_{+0.9}$ | 64.3$_{+2.4}$ |
| | MOON | 75.6$_{+13.6}$ | 73.6$_{+11.5}$ | 20.3$_{+1.2}$ | 40.8$_{-4.6}$ | 70.8$_{+10.6}$ | 89.1$_{+8.7}$ | 89.6$_{+2.3}$ | 71.1$_{+4.5}$ | 93.6$_{+2.2}$ | 42.0$_{-0.7}$ | 67.1$_{+3.6}$ | **66.7**$_{+4.8}$ |

### (d) ViT-L/14.

| $K_{\text{eff}}$ | Method | ImageNet | SUN397 | Aircraft | EuroSAT | StanfordCars | Food101 | Pets | Flowers102 | Caltech101 | DTD | UCF101 | Avg. |
|---|---|---|---|---|---|---|---|---|---|---|---|---|---|
| | CLIP | 73.5 | 67.7 | 32.5 | 60.3 | 76.9 | 90.9 | 93.5 | 79.5 | 95.2 | 53.5 | 74.9 | 72.6 |
| Very Low | StatA | 78.9$_{+5.4}$ | 71.3$_{+3.6}$ | 40.4$_{+7.9}$ | 71.4$_{+11.1}$ | 84.4$_{+7.5}$ | 94.2$_{+3.3}$ | 97.1$_{+3.6}$ | 82.9$_{+3.4}$ | 97.0$_{+1.8}$ | 55.3$_{+1.8}$ | 77.1$_{+2.2}$ | 77.3$_{+4.7}$ |
| | MOON | 85.8$_{+12.3}$ | 80.1$_{+12.4}$ | 41.3$_{+8.8}$ | 72.4$_{+12.1}$ | 86.2$_{+9.3}$ | 97.9$_{+7.0}$ | 97.5$_{+4.0}$ | 83.5$_{+4.0}$ | 98.1$_{+2.9}$ | 64.7$_{+11.2}$ | 80.8$_{+5.9}$ | **80.8**$_{+8.2}$ |
| Low | StatA | 78.2$_{+4.7}$ | 71.6$_{+3.9}$ | 38.4$_{+5.9}$ | 65.6$_{+5.3}$ | 82.4$_{+5.5}$ | 93.1$_{+2.2}$ | 96.3$_{+2.8}$ | 82.8$_{+3.3}$ | 96.1$_{+0.9}$ | 55.4$_{+1.9}$ | 76.8$_{+1.9}$ | 76.1$_{+3.5}$ |
| | MOON | 86.4$_{+12.9}$ | 80.8$_{+13.1}$ | 39.7$_{+7.2}$ | 65.2$_{+4.9}$ | 85.2$_{+8.3}$ | 97.2$_{+6.3}$ | 97.5$_{+4.0}$ | 83.7$_{+4.2}$ | 97.7$_{+2.5}$ | 60.7$_{+7.2}$ | 79.7$_{+4.8}$ | **79.4**$_{+6.8}$ |
| Medium | StatA | 76.6$_{+3.1}$ | 70.0$_{+2.3}$ | 36.4$_{+3.9}$ | 62.6$_{+2.3}$ | 80.6$_{+3.7}$ | 92.1$_{+1.2}$ | 93.9$_{+0.4}$ | 80.8$_{+1.3}$ | 95.6$_{+0.4}$ | 54.6$_{+1.1}$ | 77.1$_{+2.2}$ | 74.5$_{+2.0}$ |
| | MOON | 83.3$_{+9.8}$ | 76.3$_{+8.6}$ | 36.3$_{+3.8}$ | 61.4$_{+1.1}$ | 83.2$_{+6.3}$ | 95.4$_{+4.5}$ | 94.9$_{+1.4}$ | 81.5$_{+2.0}$ | 96.7$_{+1.5}$ | 54.5$_{+1.0}$ | 78.2$_{+3.3}$ | **76.5**$_{+3.9}$ |

*Table 14.* **Results on four additional CLIP backbones, batch adaptation with batch size of 1,000.** Subscript green indicates improvement, red indicates decline, and gray indicates no change compared with zero-shot performance.

(a) ResNet-50.

| $K_{\text{eff}}$ | Method | ImageNet | SUN397 | Aircraft | EuroSAT | StanfordCars | Food101 | Pets | Flowers102 | Caltech101 | DTD | UCF101 | Avg. |
|---|---|---|---|---|---|---|---|---|---|---|---|---|---|
| | CLIP | 58.2 | 58.9 | 17.0 | 36.2 | 55.8 | 77.4 | 85.7 | 66.1 | 85.7 | 42.8 | 61.8 | 58.7 |
| Medium | StatA | $65.2_{+7.0}$ | $61.5_{+2.6}$ | $18.6_{+1.6}$ | $51.2_{+15.0}$ | $67.2_{+11.4}$ | $80.9_{+3.5}$ | $89.1_{+3.4}$ | $70.7_{+4.6}$ | $88.5_{+2.8}$ | $46.8_{+4.0}$ | $65.3_{+3.5}$ | $64.1_{+5.4}$ |
| | MOON | $78.2_{+20.0}$ | $75.1_{+16.2}$ | $21.3_{+4.3}$ | $43.3_{+7.1}$ | $70.6_{+14.8}$ | $87.5_{+10.1}$ | $89.7_{+4.0}$ | $73.4_{+7.3}$ | $91.1_{+5.4}$ | $47.8_{+5.0}$ | $72.2_{+10.4}$ | $68.2_{+9.5}$ |
| High | StatA | $65.4_{+7.2}$ | $63.1_{+4.2}$ | $16.5_{-0.5}$ | $51.7_{+15.5}$ | $65.4_{+9.6}$ | $81.0_{+3.6}$ | $84.4_{-1.3}$ | $70.0_{+3.9}$ | $88.3_{+2.6}$ | $47.2_{+4.4}$ | $66.0_{+4.2}$ | $63.5_{+4.8}$ |
| | MOON | $77.5_{+19.3}$ | $74.1_{+15.2}$ | $18.1_{+1.1}$ | $43.2_{+7.0}$ | $67.5_{+11.7}$ | $84.3_{+6.9}$ | $86.5_{+0.8}$ | $71.3_{+5.2}$ | $89.3_{+3.6}$ | $43.6_{+0.8}$ | $69.2_{+7.4}$ | $65.9_{+7.2}$ |
| Very High | StatA | $63.5_{+5.3}$ | $62.4_{+3.5}$ | $14.8_{-2.2}$ | $51.7_{+15.5}$ | $60.8_{+5.0}$ | $77.8_{+0.4}$ | $83.5_{-2.2}$ | $66.2_{+0.1}$ | $87.9_{+2.2}$ | $46.6_{+3.8}$ | $64.5_{+2.7}$ | $61.8_{+3.1}$ |
| | MOON | $74.7_{+16.5}$ | $70.5_{+11.6}$ | $16.7_{-0.3}$ | $43.2_{+7.0}$ | $62.3_{+6.5}$ | $79.6_{+2.2}$ | $85.9_{+0.2}$ | $68.3_{+2.2}$ | $86.1_{+0.4}$ | $42.5_{-0.3}$ | $64.8_{+3.0}$ | $63.1_{+4.4}$ |

(b) ResNet-101.

| $K_{\text{eff}}$ | Method | ImageNet | SUN397 | Aircraft | EuroSAT | StanfordCars | Food101 | Pets | Flowers102 | Caltech101 | DTD | UCF101 | Avg. |
|---|---|---|---|---|---|---|---|---|---|---|---|---|---|
| | CLIP | 61.3 | 59.0 | 17.9 | 32.9 | 63.2 | 80.7 | 86.9 | 64.3 | 89.9 | 37.3 | 61.1 | 59.5 |
| Medium | StatA | $70.5_{+9.2}$ | $65.3_{+6.3}$ | $20.5_{+2.6}$ | $33.6_{+0.7}$ | $73.9_{+10.7}$ | $85.4_{+4.7}$ | $91.1_{+4.2}$ | $73.1_{+8.8}$ | $92.2_{+2.3}$ | $43.2_{+5.9}$ | $66.5_{+5.4}$ | $65.0_{+5.5}$ |
| | MOON | $77.6_{+16.3}$ | $72.6_{+13.6}$ | $21.5_{+3.6}$ | $33.3_{+0.4}$ | $75.3_{+12.1}$ | $88.9_{+8.2}$ | $89.6_{+2.7}$ | $71.2_{+6.9}$ | $94.2_{+4.3}$ | $41.6_{+4.3}$ | $68.7_{+7.6}$ | $66.8_{+7.3}$ |
| High | StatA | $71.4_{+10.1}$ | $66.2_{+7.2}$ | $18.6_{+0.7}$ | $32.8_{-0.1}$ | $72.2_{+9.0}$ | $85.1_{+4.4}$ | $87.9_{+1.0}$ | $71.9_{+7.6}$ | $92.2_{+2.3}$ | $42.5_{+5.2}$ | $66.5_{+5.4}$ | $64.3_{+4.8}$ |
| | MOON | $77.7_{+16.4}$ | $71.2_{+12.2}$ | $19.2_{+1.3}$ | $33.1_{+0.2}$ | $72.6_{+9.4}$ | $86.3_{+5.6}$ | $87.6_{+0.7}$ | $69.1_{+4.8}$ | $93.1_{+3.2}$ | $38.7_{+1.4}$ | $65.9_{+4.8}$ | $65.0_{+5.5}$ |
| Very High | StatA | $70.1_{+8.8}$ | $65.4_{+6.4}$ | $16.9_{-1.0}$ | $32.9_{\pm0.0}$ | $68.2_{+5.0}$ | $82.4_{+1.7}$ | $87.2_{+0.3}$ | $68.7_{+4.4}$ | $91.3_{+1.4}$ | $41.9_{+4.6}$ | $63.8_{+2.7}$ | $62.6_{+3.1}$ |
| | MOON | $75.9_{+14.6}$ | $67.9_{+8.9}$ | $17.9_{\pm0.0}$ | $33.1_{+0.2}$ | $68.7_{+5.5}$ | $82.7_{+2.0}$ | $87.1_{+0.2}$ | $65.8_{+1.5}$ | $90.2_{+0.3}$ | $37.7_{+0.4}$ | $62.2_{+1.1}$ | $62.7_{+3.2}$ |

(c) ViT-B/32.

| $K_{\text{eff}}$ | Method | ImageNet | SUN397 | Aircraft | EuroSAT | StanfordCars | Food101 | Pets | Flowers102 | Caltech101 | DTD | UCF101 | Avg. |
|---|---|---|---|---|---|---|---|---|---|---|---|---|---|
| | CLIP | 62.0 | 62.1 | 19.1 | 45.4 | 60.2 | 80.4 | 87.3 | 66.6 | 91.4 | 42.7 | 63.5 | 61.9 |
| Medium | StatA | $65.9_{+3.9}$ | $63.3_{+1.2}$ | $21.9_{+2.8}$ | $51.3_{+5.9}$ | $69.3_{+9.1}$ | $82.2_{+1.8}$ | $90.3_{+3.0}$ | $74.1_{+7.5}$ | $92.6_{+1.2}$ | $47.4_{+4.7}$ | $66.1_{+2.6}$ | $65.9_{+4.0}$ |
| | MOON | $79.1_{+17.1}$ | $76.0_{+13.9}$ | $22.7_{+3.6}$ | $52.1_{+6.7}$ | $74.4_{+14.2}$ | $89.1_{+8.7}$ | $90.3_{+3.0}$ | $74.0_{+7.4}$ | $94.2_{+2.8}$ | $47.0_{+4.3}$ | $69.2_{+5.7}$ | $69.8_{+7.9}$ |
| High | StatA | $67.0_{+5.0}$ | $65.0_{+2.9}$ | $20.2_{+1.1}$ | $51.1_{+5.7}$ | $68.5_{+8.3}$ | $82.7_{+2.3}$ | $88.5_{+1.2}$ | $73.7_{+7.1}$ | $92.5_{+1.1}$ | $49.5_{+6.8}$ | $66.9_{+3.4}$ | $66.0_{+4.1}$ |
| | MOON | $79.3_{+17.3}$ | $75.1_{+13.0}$ | $20.2_{+1.1}$ | $52.1_{+6.7}$ | $71.6_{+11.4}$ | $86.5_{+6.1}$ | $87.9_{+0.6}$ | $71.8_{+5.2}$ | $93.7_{+2.3}$ | $43.8_{+1.1}$ | $67.9_{+4.4}$ | $68.2_{+6.3}$ |
| Very High | StatA | $66.6_{+4.6}$ | $66.0_{+3.9}$ | $18.8_{-0.3}$ | $51.0_{+5.6}$ | $65.1_{+4.9}$ | $81.5_{+1.1}$ | $88.0_{+0.7}$ | $70.6_{+4.0}$ | $91.9_{+0.5}$ | $49.5_{+6.8}$ | $66.5_{+3.0}$ | $65.1_{+3.2}$ |
| | MOON | $76.9_{+14.9}$ | $72.3_{+10.2}$ | $18.4_{-0.7}$ | $52.1_{+6.7}$ | $66.9_{+6.7}$ | $82.6_{+2.2}$ | $87.5_{+0.2}$ | $68.6_{+2.0}$ | $91.6_{+0.2}$ | $43.0_{+0.3}$ | $65.3_{+1.8}$ | $65.9_{+4.0}$ |

(d) ViT-L/14.

| $K_{\text{eff}}$ | Method | ImageNet | SUN397 | Aircraft | EuroSAT | StanfordCars | Food101 | Pets | Flowers102 | Caltech101 | DTD | UCF101 | Avg. |
|---|---|---|---|---|---|---|---|---|---|---|---|---|---|
| | CLIP | 73.5 | 67.7 | 32.5 | 60.3 | 76.9 | 90.9 | 93.5 | 79.5 | 95.2 | 53.5 | 74.9 | 72.6 |
| Medium | StatA | $76.2_{+2.7}$ | $69.4_{+1.7}$ | $39.1_{+6.6}$ | $71.0_{+10.7}$ | $81.9_{+5.0}$ | $91.7_{+0.8}$ | $94.8_{+1.3}$ | $81.9_{+2.4}$ | $95.6_{+0.4}$ | $56.9_{+3.4}$ | $77.6_{+2.7}$ | $76.0_{+3.4}$ |
| | MOON | $85.0_{+11.5}$ | $79.2_{+11.5}$ | $38.9_{+6.4}$ | $67.4_{+7.1}$ | $84.0_{+7.1}$ | $95.9_{+5.0}$ | $95.4_{+1.9}$ | $83.3_{+3.8}$ | $97.7_{+2.5}$ | $59.4_{+5.9}$ | $80.2_{+5.3}$ | $78.8_{+6.2}$ |
| High | StatA | $77.2_{+3.7}$ | $70.9_{+3.2}$ | $36.8_{+4.3}$ | $71.2_{+10.9}$ | $82.0_{+5.1}$ | $92.3_{+1.4}$ | $94.3_{+0.8}$ | $81.9_{+2.4}$ | $95.3_{+0.1}$ | $58.7_{+5.2}$ | $78.8_{+3.9}$ | $76.3_{+3.7}$ |
| | MOON | $85.3_{+11.8}$ | $78.3_{+10.6}$ | $35.3_{+2.8}$ | $67.2_{+6.9}$ | $83.6_{+6.7}$ | $94.6_{+3.7}$ | $94.2_{+0.7}$ | $82.5_{+3.0}$ | $96.9_{+1.7}$ | $55.9_{+2.4}$ | $79.0_{+4.1}$ | $77.5_{+5.0}$ |
| Very High | StatA | $77.3_{+3.8}$ | $71.6_{+3.9}$ | $33.7_{+1.2}$ | $71.2_{+10.9}$ | $79.5_{+2.6}$ | $91.7_{+0.8}$ | $94.1_{+0.6}$ | $80.7_{+1.2}$ | $94.9_{-0.3}$ | $59.0_{+5.5}$ | $78.7_{+3.8}$ | $75.7_{+3.1}$ |
| | MOON | $84.5_{+11.0}$ | $76.0_{+8.3}$ | $32.6_{+0.1}$ | $67.2_{+6.9}$ | $80.9_{+4.0}$ | $92.4_{+1.5}$ | $94.0_{+0.5}$ | $80.3_{+0.8}$ | $95.1_{-0.1}$ | $54.8_{+1.3}$ | $76.6_{+1.7}$ | $75.9_{+3.3}$ |

*Table 15.* **Results on four additional CLIP backbones, online adaptation with batch size of 128.** Subscript green indicates improvement, red indicates decline, and gray indicates no change compared with zero-shot performance.

### (a) ResNet-50.

| Scenario | Method | ImageNet | SUN397 | Aircraft | EuroSAT | StanfordCars | Food101 | Pets | Flowers102 | Caltech101 | DTD | UCF101 | Avg. |
|---|---|---|---|---|---|---|---|---|---|---|---|---|---|
| | CLIP | 58.2 | 58.9 | 17.0 | 36.2 | 55.8 | 77.4 | 85.7 | 66.1 | 85.7 | 42.8 | 61.8 | 58.7 |
| Low | StatA | $54.6_{-3.6}$ | $56.6_{-2.3}$ | $15.1_{-1.9}$ | $39.7_{+3.5}$ | $57.6_{+1.8}$ | $79.4_{+2.0}$ | $85.1_{-0.6}$ | $60.7_{-5.4}$ | $87.8_{+2.1}$ | $44.4_{+1.6}$ | $61.7_{-0.1}$ | $58.4_{-0.3}$ |
| | MOON | $58.2_{\pm0.0}$ | $60.4_{+1.5}$ | $16.2_{-0.8}$ | $39.5_{+3.3}$ | $57.7_{+1.9}$ | $85.2_{+7.8}$ | $89.5_{+3.8}$ | $66.7_{+0.6}$ | $87.8_{+2.1}$ | $43.0_{+0.2}$ | $62.9_{+1.1}$ | $\mathbf{60.6}_{+1.9}$ |
| Medium | StatA | $59.6_{+1.4}$ | $60.8_{+1.9}$ | $17.7_{+0.7}$ | $43.5_{+7.3}$ | $65.9_{+10.1}$ | $84.5_{+7.1}$ | $90.6_{+4.9}$ | $68.1_{+2.0}$ | $89.3_{+3.6}$ | $45.5_{+2.7}$ | $64.5_{+2.7}$ | $62.8_{+4.1}$ |
| | MOON | $70.3_{+12.1}$ | $72.9_{+14.0}$ | $20.9_{+3.9}$ | $41.0_{+4.8}$ | $69.7_{+13.9}$ | $92.5_{+15.1}$ | $93.7_{+8.0}$ | $72.0_{+5.9}$ | $90.5_{+4.8}$ | $48.3_{+5.5}$ | $70.6_{+8.8}$ | $\mathbf{67.5}_{+8.8}$ |
| High | StatA | $64.7_{+6.5}$ | $62.6_{+3.7}$ | $18.5_{+1.5}$ | $43.6_{+7.4}$ | $68.5_{+12.7}$ | $85.8_{+8.4}$ | $92.2_{+6.5}$ | $70.1_{+4.0}$ | $89.8_{+4.1}$ | $45.9_{+3.1}$ | $65.2_{+3.4}$ | $64.3_{+5.6}$ |
| | MOON | $78.3_{+20.1}$ | $77.1_{+18.2}$ | $22.2_{+5.2}$ | $41.3_{+5.1}$ | $72.9_{+17.1}$ | $93.4_{+16.0}$ | $94.2_{+8.5}$ | $73.6_{+7.5}$ | $91.4_{+5.7}$ | $49.7_{+6.9}$ | $72.8_{+11.0}$ | $\mathbf{69.7}_{+11.0}$ |
| Separate | StatA | $66.6_{+8.4}$ | $62.6_{+3.7}$ | $19.8_{+2.8}$ | $44.3_{+8.1}$ | $69.5_{+13.7}$ | $85.6_{+8.2}$ | $93.8_{+8.1}$ | $71.9_{+5.8}$ | $90.2_{+4.5}$ | $46.0_{+3.2}$ | $65.3_{+3.5}$ | $65.1_{+6.4}$ |
| | MOON | $79.4_{+21.2}$ | $76.7_{+17.8}$ | $24.7_{+7.7}$ | $40.3_{+4.1}$ | $73.3_{+17.5}$ | $92.8_{+15.4}$ | $94.1_{+8.4}$ | $74.9_{+8.8}$ | $92.3_{+6.6}$ | $53.0_{+10.2}$ | $73.8_{+12.0}$ | $\mathbf{70.5}_{+11.8}$ |

### (b) ResNet-101.

| Scenario | Method | ImageNet | SUN397 | Aircraft | EuroSAT | StanfordCars | Food101 | Pets | Flowers102 | Caltech101 | DTD | UCF101 | Avg. |
|---|---|---|---|---|---|---|---|---|---|---|---|---|---|
| | CLIP | 61.3 | 59.0 | 17.9 | 32.9 | 63.2 | 80.7 | 86.9 | 64.3 | 89.9 | 37.3 | 61.1 | 59.5 |
| Low | StatA | $60.5_{-0.8}$ | $59.3_{+0.3}$ | $16.9_{-1.0}$ | $32.7_{-0.2}$ | $65.5_{+2.3}$ | $84.9_{+4.2}$ | $91.0_{+4.1}$ | $67.8_{+3.5}$ | $92.2_{+2.3}$ | $41.1_{+3.8}$ | $62.8_{+1.7}$ | $\mathbf{61.3}_{+1.8}$ |
| | MOON | $61.8_{+0.5}$ | $60.3_{+1.3}$ | $17.6_{-0.3}$ | $32.4_{-0.5}$ | $65.9_{+2.7}$ | $87.5_{+6.8}$ | $89.9_{+3.0}$ | $65.7_{+1.4}$ | $91.8_{+1.9}$ | $37.7_{+0.4}$ | $61.9_{+0.8}$ | $61.1_{+1.6}$ |
| Medium | StatA | $66.1_{+4.8}$ | $64.2_{+5.2}$ | $19.7_{+1.8}$ | $33.3_{+0.4}$ | $72.2_{+9.0}$ | $88.1_{+7.4}$ | $94.1_{+7.2}$ | $72.1_{+7.8}$ | $93.2_{+3.3}$ | $42.9_{+5.6}$ | $65.2_{+4.1}$ | $64.6_{+5.1}$ |
| | MOON | $71.9_{+10.6}$ | $70.5_{+11.5}$ | $21.0_{+3.1}$ | $33.0_{+0.1}$ | $74.1_{+10.9}$ | $92.6_{+11.9}$ | $92.4_{+5.5}$ | $69.6_{+5.3}$ | $93.9_{+4.0}$ | $41.4_{+4.1}$ | $67.8_{+6.7}$ | $\mathbf{66.2}_{+6.7}$ |
| High | StatA | $70.5_{+9.2}$ | $65.9_{+6.9}$ | $20.6_{+2.7}$ | $33.5_{+0.6}$ | $74.1_{+10.9}$ | $88.7_{+8.0}$ | $94.4_{+7.5}$ | $73.1_{+8.8}$ | $93.4_{+3.5}$ | $43.0_{+5.7}$ | $65.7_{+4.6}$ | $65.7_{+6.2}$ |
| | MOON | $78.2_{+16.9}$ | $74.1_{+15.1}$ | $21.8_{+3.9}$ | $33.4_{+0.5}$ | $76.3_{+13.1}$ | $93.2_{+12.5}$ | $92.8_{+5.9}$ | $70.8_{+6.5}$ | $94.6_{+4.7}$ | $42.5_{+5.2}$ | $69.5_{+8.4}$ | $\mathbf{67.9}_{+8.4}$ |
| Separate | StatA | $71.4_{+10.1}$ | $65.7_{+6.7}$ | $22.1_{+4.2}$ | $32.2_{-0.7}$ | $74.9_{+11.7}$ | $88.5_{+7.8}$ | $94.2_{+7.3}$ | $73.9_{+9.6}$ | $93.4_{+3.5}$ | $41.9_{+4.6}$ | $65.7_{+4.6}$ | $65.8_{+6.3}$ |
| | MOON | $78.8_{+17.5}$ | $74.0_{+15.0}$ | $23.8_{+5.9}$ | $33.8_{+0.9}$ | $76.8_{+13.6}$ | $92.7_{+12.0}$ | $92.7_{+5.8}$ | $72.1_{+7.8}$ | $95.2_{+5.3}$ | $44.1_{+6.8}$ | $70.3_{+9.2}$ | $\mathbf{68.6}_{+9.1}$ |

### (c) ViT-B/32.

| Scenario | Method | ImageNet | SUN397 | Aircraft | EuroSAT | StanfordCars | Food101 | Pets | Flowers102 | Caltech101 | DTD | UCF101 | Avg. |
|---|---|---|---|---|---|---|---|---|---|---|---|---|---|
| | CLIP | 62.0 | 62.1 | 19.1 | 45.4 | 60.2 | 80.4 | 87.3 | 66.6 | 91.4 | 42.7 | 63.5 | 61.9 |
| Low | StatA | $61.4_{-0.6}$ | $62.7_{+0.6}$ | $19.2_{+0.1}$ | $51.0_{+5.6}$ | $61.8_{+1.6}$ | $82.6_{+2.2}$ | $91.0_{+3.7}$ | $69.0_{+2.4}$ | $92.9_{+1.5}$ | $46.4_{+3.7}$ | $64.4_{+0.9}$ | $\mathbf{63.9}_{+2.0}$ |
| | MOON | $62.4_{+0.4}$ | $63.6_{+1.5}$ | $18.2_{-0.9}$ | $49.9_{+4.5}$ | $62.2_{+2.0}$ | $87.4_{+7.0}$ | $90.3_{+3.0}$ | $67.8_{+1.2}$ | $92.6_{+1.2}$ | $42.3_{-0.4}$ | $64.1_{+0.6}$ | $63.7_{+1.8}$ |
| Medium | StatA | $64.6_{+2.6}$ | $64.8_{+2.7}$ | $21.4_{+2.3}$ | $49.9_{+4.5}$ | $68.1_{+7.9}$ | $84.4_{+4.0}$ | $92.8_{+5.5}$ | $72.5_{+5.9}$ | $93.5_{+2.1}$ | $46.4_{+3.7}$ | $65.5_{+2.0}$ | $65.8_{+3.9}$ |
| | MOON | $72.9_{+10.9}$ | $74.3_{+12.2}$ | $22.2_{+3.1}$ | $50.7_{+5.3}$ | $73.7_{+13.5}$ | $93.1_{+12.7}$ | $93.1_{+5.8}$ | $72.2_{+5.6}$ | $94.2_{+2.8}$ | $47.4_{+4.7}$ | $69.2_{+5.7}$ | $\mathbf{69.4}_{+7.5}$ |
| High | StatA | $66.9_{+4.9}$ | $64.9_{+2.8}$ | $22.0_{+2.9}$ | $50.1_{+4.7}$ | $69.9_{+9.7}$ | $84.6_{+4.2}$ | $93.2_{+5.9}$ | $73.5_{+6.9}$ | $93.7_{+2.3}$ | $46.3_{+3.6}$ | $65.6_{+2.1}$ | $66.4_{+4.6}$ |
| | MOON | $79.7_{+17.7}$ | $77.7_{+15.6}$ | $23.4_{+4.3}$ | $51.1_{+5.7}$ | $76.6_{+16.4}$ | $93.8_{+13.4}$ | $93.6_{+6.3}$ | $73.6_{+7.0}$ | $94.7_{+3.3}$ | $48.4_{+5.7}$ | $70.4_{+6.9}$ | $\mathbf{71.2}_{+9.3}$ |
| Separate | StatA | $67.0_{+5.0}$ | $63.8_{+1.7}$ | $22.9_{+3.8}$ | $44.9_{-0.5}$ | $70.4_{+10.2}$ | $84.1_{+3.7}$ | $92.8_{+5.5}$ | $74.6_{+8.0}$ | $94.0_{+2.6}$ | $45.1_{+2.4}$ | $65.0_{+1.5}$ | $65.9_{+4.0}$ |
| | MOON | $80.3_{+18.3}$ | $77.3_{+15.2}$ | $25.4_{+6.3}$ | $49.1_{+3.7}$ | $77.0_{+16.8}$ | $93.1_{+12.7}$ | $93.4_{+6.1}$ | $75.0_{+8.4}$ | $95.1_{+3.7}$ | $50.6_{+7.9}$ | $70.4_{+6.9}$ | $\mathbf{71.5}_{+9.6}$ |

### (d) ViT-L/14.

| Scenario | Method | ImageNet | SUN397 | Aircraft | EuroSAT | StanfordCars | Food101 | Pets | Flowers102 | Caltech101 | DTD | UCF101 | Avg. |
|---|---|---|---|---|---|---|---|---|---|---|---|---|---|
| | CLIP | 73.5 | 67.7 | 32.5 | 60.3 | 76.9 | 90.9 | 93.5 | 79.5 | 95.2 | 53.5 | 74.9 | 72.6 |
| Low | StatA | $73.3_{-0.2}$ | $68.2_{+0.5}$ | $34.1_{+1.6}$ | $68.8_{+8.5}$ | $77.7_{+0.8}$ | $92.0_{+1.1}$ | $95.0_{+1.5}$ | $80.2_{+0.7}$ | $95.6_{+0.4}$ | $55.4_{+1.9}$ | $76.9_{+2.0}$ | $\mathbf{74.3}_{+1.7}$ |
| | MOON | $73.8_{+0.3}$ | $68.3_{+0.6}$ | $32.2_{-0.3}$ | $67.8_{+7.5}$ | $77.7_{+0.8}$ | $94.5_{+3.6}$ | $95.2_{+1.7}$ | $79.6_{+0.1}$ | $96.0_{+0.8}$ | $54.0_{+0.5}$ | $75.9_{+1.0}$ | $74.1_{+1.5}$ |
| Medium | StatA | $75.8_{+2.3}$ | $70.6_{+2.9}$ | $38.3_{+5.8}$ | $69.8_{+9.6}$ | $81.9_{+5.0}$ | $93.2_{+2.3}$ | $96.3_{+2.8}$ | $81.5_{+2.0}$ | $95.8_{+0.6}$ | $55.6_{+2.1}$ | $77.6_{+2.7}$ | $76.0_{+3.4}$ |
| | MOON | $81.3_{+7.8}$ | $77.3_{+9.6}$ | $37.9_{+5.4}$ | $69.0_{+8.7}$ | $83.9_{+7.0}$ | $97.2_{+6.3}$ | $96.8_{+3.3}$ | $82.0_{+2.5}$ | $97.1_{+1.9}$ | $58.8_{+5.3}$ | $79.6_{+4.7}$ | $\mathbf{78.3}_{+5.7}$ |
| High | StatA | $77.6_{+4.1}$ | $71.1_{+3.4}$ | $39.6_{+7.1}$ | $68.9_{+8.6}$ | $82.9_{+6.0}$ | $93.5_{+2.6}$ | $96.5_{+3.0}$ | $81.9_{+2.4}$ | $95.7_{+0.5}$ | $55.5_{+2.0}$ | $77.5_{+2.6}$ | $76.4_{+3.8}$ |
| | MOON | $85.6_{+12.1}$ | $80.3_{+12.6}$ | $39.7_{+7.2}$ | $69.4_{+9.1}$ | $85.2_{+8.3}$ | $97.5_{+6.6}$ | $97.0_{+3.5}$ | $82.6_{+3.1}$ | $97.4_{+2.2}$ | $60.3_{+6.8}$ | $80.4_{+5.5}$ | $\mathbf{79.6}_{+7.0}$ |
| Separate | StatA | $77.6_{+4.1}$ | $70.5_{+2.8}$ | $41.3_{+8.8}$ | $66.3_{+6.0}$ | $83.2_{+6.3}$ | $93.5_{+2.6}$ | $96.3_{+2.8}$ | $82.0_{+2.5}$ | $95.8_{+0.6}$ | $54.5_{+1.0}$ | $76.8_{+1.9}$ | $76.1_{+3.6}$ |
| | MOON | $85.7_{+12.2}$ | $80.4_{+12.7}$ | $41.4_{+8.9}$ | $68.1_{+7.8}$ | $85.2_{+8.3}$ | $97.4_{+6.5}$ | $96.8_{+3.3}$ | $83.4_{+3.9}$ | $97.7_{+2.5}$ | $62.6_{+9.1}$ | $80.5_{+5.6}$ | $\mathbf{79.9}_{+7.3}$ |

*Table 16.* **Results on OpenCLIP (151M), batch adaptation with batch size of 1,000.** Subscript green indicates improvement, red indicates decline, and gray indicates no change compared with zero-shot performance.

| $K_{\text{eff}}$ | Method | ImageNet | SUN397 | Aircraft | EuroSAT | StanfordCars | Food101 | Pets | Flowers102 | Caltech101 | DTD | UCF101 | Avg. |
|---|---|---|---|---|---|---|---|---|---|---|---|---|---|
| | OpenCLIP | 73.0 | 69.9 | 29.7 | 56.4 | 89.9 | 87.5 | 92.8 | 75.4 | 96.7 | 58.3 | 67.5 | 72.5 |
| Medium | StatA | $73.4_{+0.4}$ | $68.8_{-1.1}$ | $32.6_{+2.8}$ | $62.3_{+5.9}$ | $91.2_{+1.3}$ | $87.4_{-0.2}$ | $93.9_{+1.0}$ | $79.7_{+4.2}$ | $96.4_{-0.3}$ | $59.7_{+1.4}$ | $70.2_{+2.7}$ | $74.1_{+1.7}$ |
| | MOON | $83.5_{+10.5}$ | $80.0_{+10.1}$ | $32.6_{+2.9}$ | $61.5_{+5.1}$ | $93.9_{+4.0}$ | $93.3_{+5.8}$ | $95.3_{+2.5}$ | $78.4_{+3.0}$ | $98.1_{+1.4}$ | $65.1_{+6.8}$ | $74.0_{+6.6}$ | $\mathbf{77.8}_{+5.3}$ |
| High | StatA | $74.1_{+1.1}$ | $70.1_{+0.3}$ | $32.1_{+2.4}$ | $62.4_{+6.0}$ | $91.4_{+1.5}$ | $88.2_{+0.7}$ | $93.7_{+0.9}$ | $79.7_{+4.3}$ | $96.7_{+0.1}$ | $62.1_{+3.7}$ | $71.1_{+3.6}$ | $74.7_{+2.2}$ |
| | MOON | $83.7_{+10.7}$ | $80.2_{+10.4}$ | $31.0_{+1.2}$ | $61.5_{+5.1}$ | $93.9_{+4.0}$ | $91.7_{+4.2}$ | $93.7_{+0.8}$ | $77.5_{+2.0}$ | $97.9_{+1.2}$ | $61.3_{+2.9}$ | $72.3_{+4.8}$ | $\mathbf{76.8}_{+4.3}$ |
| Very High | StatA | $74.6_{+1.6}$ | $71.3_{+1.4}$ | $30.9_{+1.1}$ | $62.4_{+6.0}$ | $91.2_{+1.3}$ | $88.2_{+0.7}$ | $93.6_{+0.8}$ | $77.8_{+2.3}$ | $96.7_{\pm0.0}$ | $62.6_{+4.3}$ | $71.4_{+3.9}$ | $74.6_{+2.1}$ |
| | MOON | $83.1_{+10.1}$ | $79.1_{+9.2}$ | $28.8_{-0.9}$ | $61.4_{+5.0}$ | $93.1_{+3.2}$ | $89.3_{+1.7}$ | $93.4_{+0.6}$ | $75.6_{+0.1}$ | $96.6_{-0.1}$ | $60.5_{+2.1}$ | $70.4_{+2.9}$ | $\mathbf{75.6}_{+3.1}$ |

*Table 17.* **Results on SigLIP (878M), batch adaptation with batch size of 1,000.** Subscript green indicates improvement, red indicates decline, and gray indicates no change compared with zero-shot performance.

| $K_{\text{eff}}$ | Method | ImageNet | SUN397 | Aircraft | EuroSAT | StanfordCars | Food101 | Pets | Flowers102 | Caltech101 | DTD | UCF101 | Avg. |
|---|---|---|---|---|---|---|---|---|---|---|---|---|---|
| | SigLIP | 82.3 | 75.4 | 60.2 | 57.1 | 94.7 | 94.7 | 96.5 | 92.7 | 98.2 | 64.8 | 83.7 | 81.8 |
| Medium | StatA | $82.7_{+0.5}$ | $76.1_{+0.7}$ | $64.7_{+4.5}$ | $60.4_{+3.3}$ | $95.4_{+0.7}$ | $94.0_{-0.7}$ | $95.4_{-1.1}$ | $90.4_{-2.3}$ | $97.6_{-0.6}$ | $68.0_{+3.2}$ | $86.0_{+2.2}$ | $82.8_{+1.0}$ |
| | MOON | $91.4_{+9.1}$ | $86.0_{+10.6}$ | $65.8_{+5.6}$ | $59.4_{+2.3}$ | $96.8_{+2.1}$ | $98.1_{+3.4}$ | $98.3_{+1.8}$ | $93.5_{+0.7}$ | $98.8_{+0.6}$ | $71.9_{+7.1}$ | $87.0_{+3.3}$ | $\mathbf{86.1}_{+4.2}$ |
| High | StatA | $82.8_{+0.6}$ | $77.3_{+1.9}$ | $64.1_{+3.9}$ | $60.8_{+3.7}$ | $95.5_{+0.8}$ | $94.8_{+0.1}$ | $95.0_{-1.5}$ | $90.5_{-2.2}$ | $97.8_{-0.4}$ | $68.6_{+3.8}$ | $85.6_{+1.9}$ | $83.0_{+1.1}$ |
| | MOON | $91.5_{+9.2}$ | $86.6_{+11.2}$ | $64.0_{+3.8}$ | $60.1_{+3.0}$ | $96.6_{+2.0}$ | $97.1_{+2.4}$ | $97.2_{+0.7}$ | $93.5_{+0.7}$ | $98.7_{+0.5}$ | $66.6_{+1.7}$ | $85.3_{+1.6}$ | $\mathbf{85.2}_{+3.3}$ |
| Very High | StatA | $82.8_{+0.5}$ | $77.6_{+2.2}$ | $60.4_{+0.2}$ | $60.8_{+3.7}$ | $95.4_{+0.7}$ | $94.8_{+0.1}$ | $94.9_{-1.6}$ | $90.4_{-2.3}$ | $97.7_{-0.5}$ | $69.0_{+4.1}$ | $84.5_{+0.7}$ | $82.6_{+0.7}$ |
| | MOON | $91.0_{+8.8}$ | $85.2_{+9.8}$ | $58.8_{-1.4}$ | $60.1_{+3.0}$ | $96.2_{+1.5}$ | $95.6_{+0.9}$ | $97.0_{+0.5}$ | $93.4_{+0.6}$ | $97.6_{-0.6}$ | $65.1_{+0.3}$ | $82.7_{-1.1}$ | $\mathbf{83.9}_{+2.0}$ |

*Table 18.* **Results on EVA-CLIP (1.1B), batch adaptation with batch size of 1,000.** Subscript green indicates improvement, red indicates decline, and gray indicates no change compared with zero-shot performance.

| $K_{\text{eff}}$ | Method | ImageNet | SUN397 | Aircraft | EuroSAT | StanfordCars | Food101 | Pets | Flowers102 | Caltech101 | DTD | UCF101 | Avg. |
|---|---|---|---|---|---|---|---|---|---|---|---|---|---|
| | EVA-CLIP | 78.0 | 72.9 | 33.6 | 70.1 | 91.2 | 91.0 | 94.2 | 75.0 | 97.3 | 60.6 | 78.0 | 76.5 |
| Medium | StatA | $77.8_{-0.2}$ | $71.5_{-1.5}$ | $36.9_{+3.3}$ | $72.7_{+2.6}$ | $91.8_{+0.6}$ | $90.8_{-0.2}$ | $93.9_{-0.3}$ | $77.8_{+2.7}$ | $97.1_{-0.2}$ | $62.4_{+1.8}$ | $79.7_{+1.7}$ | $77.5_{+1.0}$ |
| | MOON | $81.1_{+3.1}$ | $75.6_{+2.7}$ | $36.5_{+2.8}$ | $72.7_{+2.5}$ | $93.1_{+1.8}$ | $93.2_{+2.2}$ | $94.8_{+0.6}$ | $76.4_{+1.4}$ | $97.7_{+0.4}$ | $64.8_{+4.2}$ | $80.4_{+2.5}$ | $\mathbf{78.7}_{+2.2}$ |
| High | StatA | $78.3_{+0.4}$ | $73.0_{+0.1}$ | $36.2_{+2.6}$ | $73.2_{+3.1}$ | $92.2_{+1.0}$ | $91.6_{+0.6}$ | $94.2_{\pm0.0}$ | $78.4_{+3.4}$ | $97.4_{+0.1}$ | $64.8_{+4.2}$ | $80.3_{+2.4}$ | $78.2_{+1.6}$ |
| | MOON | $81.3_{+3.3}$ | $76.2_{+3.3}$ | $35.0_{+1.4}$ | $72.7_{+2.6}$ | $93.1_{+1.9}$ | $92.7_{+1.7}$ | $94.2_{\pm0.0}$ | $76.7_{+1.7}$ | $97.8_{+0.5}$ | $63.5_{+2.9}$ | $80.1_{+2.1}$ | $\mathbf{78.5}_{+2.0}$ |
| Very High | StatA | $78.7_{+0.8}$ | $73.8_{+0.8}$ | $35.4_{+1.8}$ | $73.2_{+3.1}$ | $92.3_{+1.0}$ | $91.6_{+0.6}$ | $94.2_{\pm0.0}$ | $77.7_{+2.7}$ | $97.4_{+0.1}$ | $65.5_{+5.0}$ | $80.7_{+2.7}$ | $\mathbf{78.2}_{+1.7}$ |
| | MOON | $81.2_{+3.2}$ | $76.1_{+3.2}$ | $33.6_{-0.1}$ | $72.7_{+2.6}$ | $92.9_{+1.6}$ | $91.8_{+0.8}$ | $94.1_{-0.1}$ | $76.0_{+0.9}$ | $97.5_{+0.2}$ | $63.1_{+2.5}$ | $79.5_{+1.5}$ | $78.0_{+1.5}$ |

*Table 19.* **Results on full dataset with all classes.** Subscript green indicates improvement, red indicates decline, and gray indicates no change compared with zero-shot performance.

| $K_{\text{eff}}$ | Method | ImageNet | SUN397 | Aircraft | EuroSAT | StanfordCars | Food101 | Pets | Flowers102 | Caltech101 | DTD | UCF101 | Avg. |
|---|---|---|---|---|---|---|---|---|---|---|---|---|---|
| | CLIP | 66.6 | 62.5 | 24.7 | 48.3 | 65.6 | 85.9 | 89.1 | 70.7 | 93.2 | 43.5 | 67.5 | 65.2 |
| All | StatA | $69.9_{+3.3}$ | $68.7_{+6.2}$ | $24.7_{\pm0.0}$ | $67.3_{+19.0}$ | $68.0_{+2.4}$ | $87.1_{+1.2}$ | $92.4_{+3.3}$ | $75.2_{+4.5}$ | $94.2_{+1.0}$ | $48.4_{+4.9}$ | $73.5_{+6.0}$ | $\mathbf{69.9}_{+4.7}$ |
| | MOON | $68.7_{+2.1}$ | $65.4_{+2.9}$ | $24.4_{-0.3}$ | $59.1_{+10.8}$ | $67.2_{+1.6}$ | $86.7_{+0.8}$ | $90.2_{+1.1}$ | $72.8_{+2.1}$ | $93.4_{+0.2}$ | $45.0_{+1.5}$ | $70.9_{+3.4}$ | $67.6_{+2.4}$ |

*Table 20.* **Results on random class scenarios, where $K_{\text{eff}}$ is randomly sampled within [1, min{$N$, $K$}].** Subscript green indicates improvement, red indicates decline, and gray indicates no change compared with zero-shot performance.

(a) Batch Size: 64.

| Method | ImageNet | SUN397 | Aircraft | EuroSAT | StanfordCars | Food101 | Pets | Flowers102 | Caltech101 | DTD | UCF101 | Avg. |
|---|---|---|---|---|---|---|---|---|---|---|---|---|
| CLIP | 66.6 | 62.5 | 24.7 | 48.3 | 65.6 | 85.9 | 89.1 | 70.7 | 93.2 | 43.5 | 67.5 | 65.2 |
| StatA | $68.7_{+2.1}$ | $64.8_{+2.3}$ | $24.1_{-0.6}$ | $50.6_{+2.3}$ | $68.9_{+3.3}$ | $87.1_{+1.2}$ | $90.8_{+1.7}$ | $72.1_{+1.4}$ | $93.8_{+0.6}$ | $46.6_{+3.1}$ | $68.7_{+1.2}$ | $66.9_{+1.7}$ |
| MOON | $75.0_{+8.4}$ | $69.0_{+6.5}$ | $22.5_{-2.2}$ | $47.8_{-0.5}$ | $69.6_{+4.0}$ | $89.2_{+3.3}$ | $90.0_{+0.9}$ | $71.7_{+1.0}$ | $93.5_{+0.3}$ | $42.5_{-1.0}$ | $68.8_{+1.3}$ | $\mathbf{67.2}_{+2.0}$ |

(b) Batch Size: 128.

| Method | ImageNet | SUN397 | Aircraft | EuroSAT | StanfordCars | Food101 | Pets | Flowers102 | Caltech101 | DTD | UCF101 | Avg. |
|---|---|---|---|---|---|---|---|---|---|---|---|---|
| CLIP | 66.6 | 62.5 | 24.7 | 48.3 | 65.6 | 85.9 | 89.1 | 70.7 | 93.2 | 43.5 | 67.5 | 65.2 |
| StatA | $68.6_{+2.0}$ | $64.3_{+1.8}$ | $23.6_{-1.1}$ | $53.3_{+5.0}$ | $67.5_{+1.9}$ | $86.9_{+1.0}$ | $91.1_{+2.0}$ | $71.5_{+0.8}$ | $93.8_{+0.6}$ | $46.9_{+3.4}$ | $67.9_{+0.4}$ | $\mathbf{66.8}_{+1.6}$ |
| MOON | $74.1_{+7.5}$ | $67.3_{+4.8}$ | $22.0_{-2.7}$ | $50.6_{+2.3}$ | $66.4_{+0.8}$ | $87.9_{+2.0}$ | $90.7_{+1.6}$ | $70.2_{-0.5}$ | $93.2_{\pm0.0}$ | $44.1_{+0.6}$ | $66.4_{-1.1}$ | $66.6_{+1.4}$ |

(c) Batch Size: 256.

| Method | ImageNet | SUN397 | Aircraft | EuroSAT | StanfordCars | Food101 | Pets | Flowers102 | Caltech101 | DTD | UCF101 | Avg. |
|---|---|---|---|---|---|---|---|---|---|---|---|---|
| CLIP | 66.6 | 62.5 | 24.7 | 48.3 | 65.6 | 85.9 | 89.1 | 70.7 | 93.2 | 43.5 | 67.5 | 65.2 |
| StatA | $68.2_{+1.6}$ | $64.1_{+1.6}$ | $24.1_{-0.6}$ | $55.3_{+7.0}$ | $66.3_{+0.7}$ | $87.4_{+1.5}$ | $91.9_{+2.8}$ | $73.2_{+2.5}$ | $93.8_{+0.6}$ | $47.4_{+3.9}$ | $68.7_{+1.2}$ | $\mathbf{67.3}_{+2.1}$ |
| MOON | $72.9_{+6.3}$ | $64.7_{+2.2}$ | $23.3_{-1.4}$ | $51.9_{+3.6}$ | $64.8_{-0.8}$ | $89.0_{+3.1}$ | $91.7_{+2.6}$ | $72.2_{+1.5}$ | $93.5_{+0.3}$ | $46.5_{+3.0}$ | $67.9_{+0.4}$ | $67.1_{+1.9}$ |

(d) Batch Size: 500.

| Method | ImageNet | SUN397 | Aircraft | EuroSAT | StanfordCars | Food101 | Pets | Flowers102 | Caltech101 | DTD | UCF101 | Avg. |
|---|---|---|---|---|---|---|---|---|---|---|---|---|
| CLIP | 66.6 | 62.5 | 24.7 | 48.3 | 65.6 | 85.9 | 89.1 | 70.7 | 93.2 | 43.5 | 67.5 | 65.2 |
| StatA | $68.1_{+1.5}$ | $64.2_{+1.7}$ | $24.9_{+0.2}$ | $54.5_{+6.2}$ | $67.2_{+1.6}$ | $87.5_{+1.6}$ | $92.5_{+3.4}$ | $74.2_{+3.5}$ | $93.5_{+0.3}$ | $47.1_{+3.6}$ | $69.4_{+1.9}$ | $\mathbf{67.6}_{+2.3}$ |
| MOON | $70.9_{+4.3}$ | $62.4_{-0.1}$ | $24.7_{\pm0.0}$ | $52.7_{+4.4}$ | $66.5_{+0.9}$ | $89.8_{+3.9}$ | $91.8_{+2.7}$ | $73.1_{+2.4}$ | $93.7_{+0.5}$ | $47.6_{+4.1}$ | $69.1_{+1.6}$ | $67.5_{+2.3}$ |

(e) Batch Size: 1,000.

| Method | ImageNet | SUN397 | Aircraft | EuroSAT | StanfordCars | Food101 | Pets | Flowers102 | Caltech101 | DTD | UCF101 | Avg. |
|---|---|---|---|---|---|---|---|---|---|---|---|---|
| CLIP | 66.6 | 62.5 | 24.7 | 48.3 | 65.6 | 85.9 | 89.1 | 70.7 | 93.2 | 43.5 | 67.5 | 65.2 |
| StatA | $67.5_{+0.9}$ | $65.2_{+2.7}$ | $25.4_{+0.7}$ | $55.0_{+6.7}$ | $68.0_{+2.4}$ | $87.2_{+1.3}$ | $93.0_{+3.9}$ | $75.3_{+4.6}$ | $93.3_{+0.1}$ | $47.8_{+4.3}$ | $70.7_{+3.2}$ | $68.0_{+2.8}$ |
| MOON | $67.9_{+1.3}$ | $64.0_{+1.5}$ | $25.4_{+0.7}$ | $54.7_{+6.4}$ | $68.1_{+2.5}$ | $89.7_{+3.8}$ | $92.0_{+2.9}$ | $74.0_{+3.3}$ | $94.1_{+0.9}$ | $48.1_{+4.6}$ | $70.7_{+3.2}$ | $\mathbf{68.1}_{+2.8}$ |

(f) Batch Size: 2000.

| Method | ImageNet | SUN397 | Aircraft | EuroSAT | StanfordCars | Food101 | Pets | Flowers102 | Caltech101 | DTD | UCF101 | Avg. |
|---|---|---|---|---|---|---|---|---|---|---|---|---|
| CLIP | 66.6 | 62.5 | 24.7 | 48.3 | 65.6 | 85.9 | 89.1 | 70.7 | 93.2 | 43.5 | 67.5 | 65.2 |
| StatA | $68.1_{+1.5}$ | $66.1_{+3.6}$ | $26.3_{+1.6}$ | $56.7_{+8.4}$ | $69.6_{+4.0}$ | $86.7_{+0.8}$ | $93.0_{+3.9}$ | $77.0_{+6.3}$ | $93.3_{+0.1}$ | $47.4_{+3.9}$ | $71.5_{+4.0}$ | $\mathbf{68.7}_{+3.5}$ |
| MOON | $68.8_{+2.2}$ | $66.2_{+3.7}$ | $26.3_{+1.6}$ | $55.0_{+6.7}$ | $69.9_{+4.3}$ | $89.4_{+3.5}$ | $92.0_{+2.9}$ | $75.3_{+4.6}$ | $94.2_{+1.0}$ | $47.7_{+4.2}$ | $71.4_{+3.9}$ | $\mathbf{68.7}_{+3.5}$ |

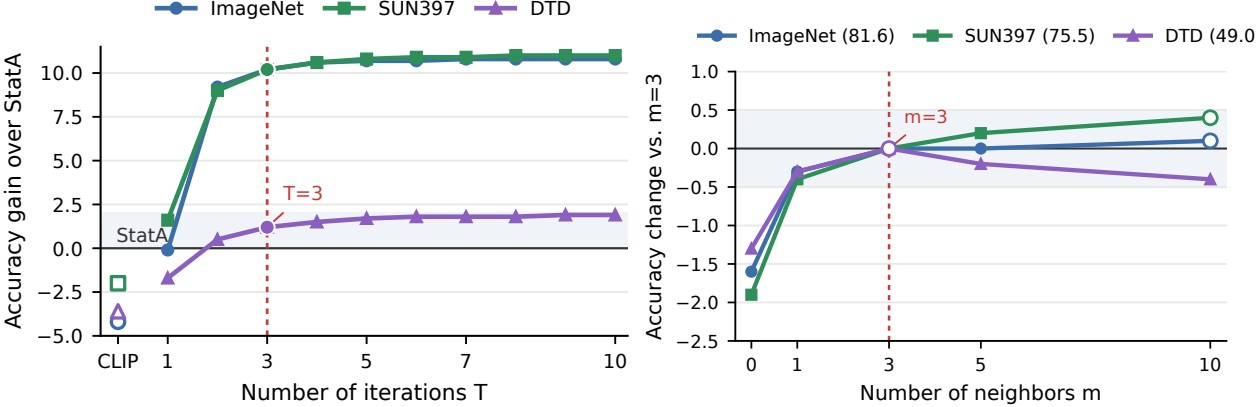

*Figure 4.* **Hyperparameter sensitivity analysis.** Results are reported on batch adaptation, *Medium* scenario, batch size of 1,000.

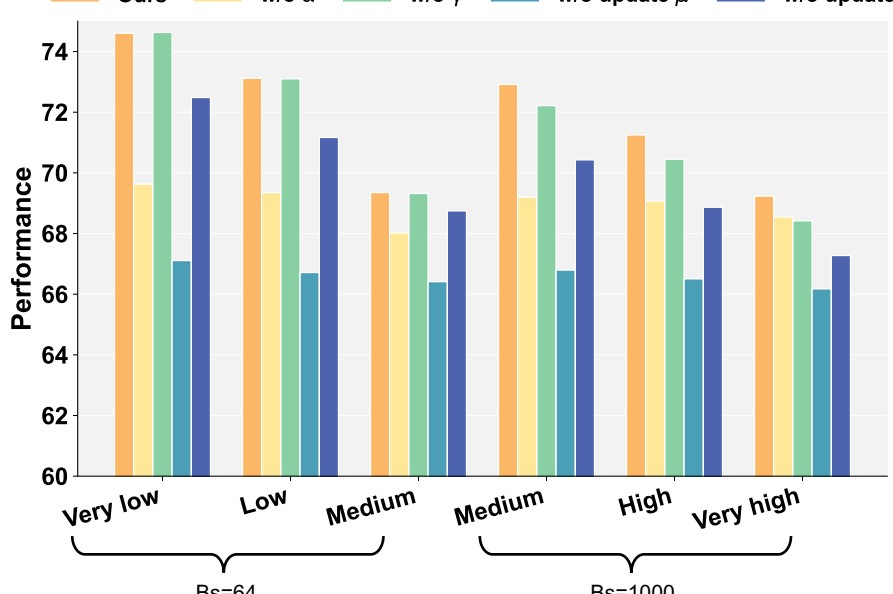

*Figure 5.* **Detailed ablation study on components, over various batch sizes and scenarios.** Each reported performance is averaged over all datasets and runs.

## J. Additional Analyses

**Hyperparameter sensitivity.** We analyze the sensitivity of MOON to the existing hyperparameters, including iteration number $T$ and the number of neighbors in Laplacian term $m$ in Fig. 4. The results show that MOON is robust to both hyperparameters. Moreover, the performance improves rapidly within the first few iterations and outperforms StatA nearly saturates after $T = 3$, suggesting fast practical convergence. Together with the dynamic shrinkage mechanism, MOON thus requires no task-specific hyperparameter tuning. We set $T = 10$ and $m = 3$ by default for all experiments.

**Fine-grained ablation analysis.** We present a fine-grained ablation study across batch sizes and sparsity levels in Fig. 5. Consistent with our design motivation, the impact of the class-level weight $\alpha$ diminishes as the effective class set becomes denser (e.g., in the *Very High* scenario). This confirms that $\alpha$ functions precisely as intended: effectively suppressing outlier classes in sparse settings while relaxing constraints when the distribution approaches uniformity. Conversely, the instance-level weight $\gamma$ shows marginal influence at small batch sizes due to high statistical variance but becomes increasingly significant at larger batch sizes, where it can leverage stable batch statistics to filter unreliable samples effectively. Furthermore, the performance gain from iterative parameter updates tends to saturate in scenarios with large batches and dense classes, suggesting that abundant data naturally provides sufficient empirical evidence for reliable estimation. Overall,

the full `MOON` framework consistently yields optimal performance.

*Table 21.* **Implementation of shrinkage strength $\beta_k$.**

*(a)* Batch adaptation, with batch size of 64.

| Scenario | Very Low | Low | Medium | Avg. |
|---|---|---|---|---|
| `MOON` w/ soft $\beta_k$ | **75.3** | **73.3** | 68.6 | **72.4** |
| `MOON` w/ hard $\beta_k$ | 74.6 | 73.1 | **69.4** | **72.4** |

*(b)* Batch adaptation, with batch size of 1,000.

| Scenario | Medium | High | Very High | Avg. |
|---|---|---|---|---|
| `MOON` w/ soft $\beta_k$ | 72.1 | 70.3 | 68.1 | 70.2 |
| `MOON` w/ hard $\beta_k$ | **72.9** | **71.3** | **69.2** | **71.1** |

*(c)* Online adaptation, with batch size of 128.

| Scenario | Low | Medium | High | Separate | Avg. |
|---|---|---|---|---|---|
| `MOON` w/ soft $\beta_k$ | 64.9 | **71.8** | **73.6** | **74.2** | 71.1 |
| `MOON` w/ hard $\beta_k$ | **66.5** | **71.8** | 73.4 | 73.9 | **71.4** |

**Implementation of $\beta_k$.** We investigate the implementation strategy for the shrinkage strength $\beta_k$ by comparing the standard soft assignment against the hard assignment (i.e., discretization via argmax) on the probability simplex. As shown in Tab. 21, employing hard assignments consistently yields superior robustness and stability. This advantage stems from the inherent property of the softmax operation, which produces non-zero residual probabilities for all classes. In a soft assignment regime, these residuals can accumulate to form misleading counts for absent classes, thereby weakening the necessary shrinkage. By adopting hard assignments, we effectively eliminate this background noise, ensuring that $\beta_k$ accurately reflects the true class presence and enforces strict anchoring for outlier categories.

## K. Limitations and Future Work

While our `MOON` demonstrates robust performance and efficiency, there remain promising avenues for future exploration. First, vMF distributions inherently assume isotropy on the hypersphere. Explicitly modeling the anisotropy of VLM representations, for instance, by exploring Fisher-Bingham distributions or other non-isotropic spherical models, could potentially capture more complex feature geometries. Second, `MOON` can be more deeply integrated with memory banks or caches, enabling more efficient and effective adaptation in sample-wise, online-TTA mode. Finally, the construction of the affinity graph still entails a quadratic complexity with respect to the batch size. Incorporating approximate nearest neighbor search strategies could be beneficial, especially for scaling to large-scale offline adaptation tasks.

