# OpenReview forum: "Von Mises-Fisher Mixture Model with Dynamic Shrinkage for Realistic Test-Time Transduction"
_ICML.cc/2026/Conference — ICML 2026 regular_

### Official Review · Reviewer_VCqZ · 2026-02-22

**Soundness:** 3
**Presentation:** 1
**Significance:** 2
**Originality:** 3
**Overall Recommendation:** 4
**Confidence:** 3

**Summary:**

Existing TTA works often struggle in highly imbalanced class distributions, this paper systematically revisit transduction from the perspective of penalized likelihood estimation (PLE), showing that PLE with a KL-divergence anchor term naturally yields an adaptive shrinkage behavior between prior anchors and empirical estimates. Based on this, this paper propose Moon which utilizes a mixture of von Mises-Fisher distributions to model feature representations on the unit hypersphere. Extensive experiments validate the effectiveness of the proposed method.

**Compliance With Llm Reviewing Policy:**

Affirmed.

**Final Justification:**

My concerns have been fully addressed, therefore, i increase my score from weak reject to weak accept and recommend positive support for this paper.

**Key Questions For Authors:**

Please ref to the weaknesses above.

**Limitations:**

Please ref to the weaknesses above.

**Strengths And Weaknesses:**

## Strengths
1. Extensive experiments have been conducted to demonstrate the effectiveness of the proposed method, including the generality on cross-domain and out domain datasets with various batch size, the efficiency of runtime cost and so on.
2. A detailed theoretical analysis has been conducted to explore the limitations of the previous works.

## Weaknesses
1. The most critical issue is the poor readability of the manuscript. For instance, in lines 186–197, many mathematical symbols are not explained. For example, the definitions of $T(\cdot)$, $A(\cdot)$, and $\eta$ are not specified at all. Without these explanations, it is very difficult for readers to understand the meaning of the equation.
2. The experiment setting should be re-discussed. Although authors claim that the proposed Moon is targeting highly imbalanced scenarios, I haven't seen any special way to construct the highly class imbalanced batches in the main experiment, only controlling the batch size in Table 1 is not convincing to demonstrate the effectiveness of Moon in this setting, making each batch with only a few or even one single class maybe a better way.
3. This paper assume a batch-wise access of test samples while avoiding the setting that users can only access a single image at each step, without discussion on this setting makes the motivation "highly imbalanced class distributions causes existing works performance degradation or even collapse" not convincing, as a lot of works take single image batch as a default setting, which can also be seen as a highly imbalanced class distributions. Therefore, I think discussion and comparison with these works is needed.

---

> ### Author Rebuttal · Authors · 2026-03-30
>
> We are truly grateful for the time you have taken and your insightful review! Here we address your concerns as follows:
>
> > Q1: Poor readability by undefined notations
>
> We sincerely apologize for any confusion. We placed the full definitions in Appendix D.1. Formally, for a density function belongs to a regular minimal exponential family $p(x\mid\eta)=h(x)\exp(\eta^T T(x)-A(\eta))$:
>
> - $\eta$: The natural parameter.
> - $T(x)$: The sufficient statistic.
> - $A(\eta)$: The log-partition function (different from Bessel function ratio $A_d(\cdot)$ in Eq. (6)).
> - $h(x)$: The base measure.
>
> We will explicitly add them in the main text, ensuring all notations are defined.
>
> > Q2: Imbalance settings
>
> Thanks. We completely agree that restricting batches to "few or even one class" is the right way to evaluate imbalanced scenarios. In fact, this is exactly how our experiments were done. As defined in **Section 3 and Appendix C.4**, we explicitly constructed two distinct imbalanced settings rather than merely controlling the batch size:
>
> - **Batch Adaptation (Class Sparsity):** We construct batches by strictly limiting the number of effective classes ($K_{eff}$) present in each independently processed batch. For example, our *Very Low* scenario strictly limits $K_{eff}$ to the range of 1-4 classes, exactly as suggested.
> - **Online Adaptation (Temporal Correlation):** Test samples arrive as a non-i.i.d. data stream, following a Dirichlet distribution $Dir(\xi \cdot \mathbf{1})$. By adopting a small $\xi$, we concentrate classes into few slots to create high temporal correlation. We further include a *Separate* scenario where classes arrive sequentially one by one.
>
> We apologize if this was not clear enough. We will move the detailed sampling procedure to the main text to ensure this core setting is clear to all readers.
>
> > Q3: Comparison with sample-wise TTA methods
>
> **Sample-wise baselines.** Generally, these methods can be divided into two groups: episodic and online TTA. The former updates parameters for each sample with episodic reset, while the latter maintains a memory bank for training-free adaptation. Actually, we've already discussed several online TTA methods in our paper (e.g., TDA, DMN) and shown that MOON outperforms them. For episodic TTA, we supplement a representative method TPT:
>
> (Online Medium)
> |ImageNet|CLIP|TDA|TPT|MOON|
> |-|-|-|-|-|
> |Acc.|66.6|68.2|68.0|76.5|
> |Time|9min|55min|~12h|11min|
>
> As shown, such methods are highly inefficient due to gradient backpropagation. Moreover, test-time parameter fine-tuning (e.g., prompt tuning) does not bring significant performance gains compared with training-free methods, which has also been observed in prior works [1,2].
>
> **Extend MOON to sample-wise.** Vanilla MOON does not support sample-wise, as transductive learning inherently requires a batch of samples to perform clustering. However, MOON can be easily extended to a sample-wise online mode by introducing a memory bank, as what ADAPT [3] has done.
>
> Formally, assume memory bank $\mathcal B={(f_j,\bar{\mathbf{z}}_j)}$, where $f_j$ is historical feature, and $\bar{\mathbf{z}}_j$ is soft label. Assume arriving sample with feature $f\_\*$ and zero-shot prediction $\hat{y}\_\*$, current mixture parameters $M\_{\mathcal{B}}=(\mu\_k^{B},\kappa\_k^{B})\_{k=1}^K$. The PLE objective becomes (Note: KL anchor term becomes constant w.r.t. new assignment $\mathbf z\_\*$)
>
> $$\min_{\mathbf z\_\* \in\Delta^K}-\mathbf z\_\*^T\log\mathbf p\_\*^{B}-\sum_{j\in B}\omega_{\*\_j},\mathbf z\_\*^T\bar{\mathbf z}_j+\mathrm{KL}(\mathbf z\_\* |\hat{y}\_\*).$$
> Solving the problem yields the closed-form predictor
> $$\mathbf z\_\*=\frac{\hat{y}\_\* \odot\exp(\log\mathbf p\_\*^{B}+\sum\_{j\in B}\omega\_{\* j}\bar{\mathbf z}\_j)}{(\hat{y}\_\* \odot\exp(\log\mathbf p\_\*^{B}+\sum\_{j\in B}\omega\_{\* j}\bar{\mathbf z}\_j))^T \mathbf 1}.$$
> Hence, MOON can make immediate predictions for newly arriving samples.
>
> The mixture parameters also continue to be stabilized by KL-anchored shrinkage mechanism using bank statistics. Let
> $$S_k^{B}=\sum_{j\in B}\gamma_j\bar z_{j,k}f_j,\quad n_k^{B}=\sum_{j\in B}\gamma_j \bar z_{j,k}.$$
> Then the bank-based parameter update keeps the same anchored form:
> $$\mu_k^{B}=\frac{\beta_k^{B}v_k^{B}+(1-\beta_k^{B})\mu_k'}
> {|\beta_k^{B}v_k^{B}+(1-\beta_k^{B})\mu_k'|},\quad A_d(\kappa_k^{B})=|\beta_k^{B}v_k^{B}+(1-\beta_k^{B})\mu_k'|,$$
> where $v_k^{B}=\frac{S_k^{B}}{n_k^{B}}$ and $\beta_k^{B}=\frac{n_k^{B}}{n_k^{B}+\alpha_k}$.
>
> We conduct experiments to validate this:
>
> (Online Medium)
> ||CLIP|MOON(online)|MOON|
> |-|-|-|-|
> |ImageNet|66.6|73.0|76.5|
> |SUN397|62.5|69.5|73.9|
> |DTD|43.5|44.1|50.0|
>
> As shown, MOON could still achieve excellent performance.
>
> [1] Efficient test-time adaptation of vision-language models. CVPR'24.
>
> [2] On the test-time zero-shot generalization of vision-language models: Do we really need prompt learning? CVPR'24.
>
> [3] Backpropagation-Free Test-Time Adaptation via Probabilistic Gaussian Alignment. NIPS'25.

---

> > ### Author Rebuttal · Reviewer_VCqZ · 2026-04-01
> >
> > Thanks for the authors thorough explanation, my concerns have been well addressed. I will adjust the score accordingly. I also strongly recommend the authors to supplement the rebuttal content into the revised version to improve the readability and interpretability.

---

> > > ### Author Response · Authors · 2026-04-01
> > >
> > > Thanks again for your feedback and increasing the rating! We are pleased to know that we have solved your concerns and will definitely add these contents in the final version according to your suggestion. We sincerely thank you for your dedication and effort in evaluating our submission!

---

### Official Review · Reviewer_dAzL · 2026-03-09

**Soundness:** 3
**Presentation:** 3
**Significance:** 3
**Originality:** 3
**Overall Recommendation:** 4
**Confidence:** 3

**Summary:**

This paper focuses on the test-time transduction for CLIP. The authors analyze the class imbalance problem from the perspective of penalized likelihood estimation. To address the imbalance issue, the paper proposes MOON. MOON introduces dynamic shrinkage at both the instance and class level to suppress unreliable assignments and outlier classes. Extensive experiments across 11 datasets demonstrate that MOON efficiently improves VLM zero-shot performance in batch and online adaptation settings.

**Compliance With Llm Reviewing Policy:**

Affirmed.

**Final Justification:**

I have read the author's rebuttal carefully and will maintain the positive score.

**Key Questions For Authors:**

1.	Many prior transductive approaches also rely on probabilistic modeling or feature clustering. Could the authors more explicitly discuss how MOON differs from these methods in the method section?

**Limitations:**

yes

**Strengths And Weaknesses:**

Strengths:

1.	The problem formulation is clear and well-motivated. The paper highlights that many TTA methods implicitly assume balanced class distributions, which is unrealistic in real-world deployment.
2.	The theoretical analysis is convincing. The paper provides a strong theoretical foundation by formulating test-time transduction as a penalized likelihood estimation. The authors rigorously prove that KL-anchored estimators naturally yield adaptive shrinkage for exponential family distributions.
3.	The evaluation is comprehensive. MOON is evaluated on 11 standard datasets in 2 imbalance scenarios. Moreover, MOON consistently improves performance across different model architectures and sizes.


Weaknesses:

1.	The discussion may overlook prior work in imbalance TTA. The author states that “most existing methods and benchmarks assume that class marginals are fixed and uniform.” However, the class imbalance problem has already been studied in several test-time adaptation works (e.g., [1–3]). It would be better for the authors to clarify how their class sparsity differs from the imbalance settings considered in prior TTA works.
2.	The advantage of transduction over TTA may be overstated. The paper claims that transductive methods are preferable because they avoid model drift. However, TTA methods can also adopt an episodic protocol， where parameters can be reset after each batch to mitigate the drift.
3.	The practical applicability under a small batch size is unclear. It is unclear how the method behaves when the batch size is extremely small (e.g., batch size = 1), which typically occurs in real-time deployment scenarios.
4.	The comparison between TTA and transduction methods may be unfair. TTA baselines are evaluated under continual adaptation, where parameters are updated across batches. In contrast, the proposed transduction method operates independently on each batch. Such an evaluation may be unfair since continual TTA is known to suffer from drift. A fairer comparison is to include episodic TTA that reset model parameters after each batch.
5.	The analysis on hyperparameter sensitivity is limited. The method introduces several hyperparameters, such as the number of neighbors $𝑚$ and the number of inference iterations $T$. It would be better to include a sensitivity analysis on these parameters.

[1] Towards stable test-time adaptation in dynamic wild world. ICLR’23.

[2] Towards Real-World Test-Time Adaptation: Tri-Net Self-Training with Balanced Normalization. AAAI’24.

[3] DELTA: degradation-free fully test-time adaptation. ICLR’24.

---

> ### Author Rebuttal · Authors · 2026-03-30
>
> Thanks for appreciating our work, your positive feedback is incredibly encouraging for us! In the following response, we would like to address your major concern and provide additional clarification:
>
> > Q1: Difference from prior imbalance-TTA works
>
> Thanks for pointing this out. We acknowledge that "most existing methods" might be ambiguous, and we will revise it. However, we would like to clarify several key differences:
>
> - **Task:** These works focus on vision-only models within closed-set label spaces. MOON focuses on VLMs, which handles complex multimodal data structures across open-vocabulary spaces.
> - **Protocol:** These works require access to model weights or raw images for backpropagation or augmentation. MOON operates under a black-box assumption, requiring only output logits and classnames.
> - **Imbalance Settings:** These works mostly discuss non-i.i.d. online data streams controlled via Dirichlet distributions, corresponding to *Online adaptation* in MOON. We further consider the *Batch adaptation* setting. Here, historical information is unavailable, forcing predictions to rely entirely on data structure within batch.
> - **Methodology :** These methods mitigate imbalance by calibrating normalization layers or reweighting gradients during continuous backpropagation. In contrast, MOON is training-free. It constructs a statistical mixture model instead.
>
> > Q2: Comparison with episodic TTA methods
>
> We would like to clarify that most of our evaluated baselines are actually training-free. To address your concern, we focus on TENT, the only baseline involving parameter updates. We provide the TENT-episodic results below:
>
> (Bs=128, Online Medium)
>
> | |TENT|TENT-episodic|MOON|
> |-|-|-|-|
> | **DTD** | 44.0 | 44.3 | **50.0** |
> | **Caltech101**| 93.3 | 93.4 | **95.3** |
>
> As shown, MOON outperforms TENT even under episodic protocol. Moreover, MOON does not require computationally expensive gradient backpropagation, leading to its significant advantage in efficiency. We also include another representative episodic baseline TPT. Please refer to **VCqZ@Q3** for detailed analyses and experimental results.
>
> > Q3: Applicability under a small batch size (e.g., bs=1)
>
> The vanilla MOON does not support extreme small batches, as transductive learning inherently requires a statistically significant batch of samples to perform probabilistic soft clustering. However, for real-time deployments, MOON can be easily extended to a sample-wise online mode. Specifically, by introducing a lightweight memory bank/cache, we can collect enough historical samples to enable clustering. Due to space limit, please refer to **VCqZ@Q3** for detailed analyses.
>
> > Q4: Hyperparameter sensitivity analysis of m and T
>
> Thanks for suggestion! We have supplemented the sensitivity analyses of $m$ and $T$:
>
> (Bs=1000, Batch Medium)
> | | CLIP | StatA | MOON | | | | | | | | | |
> |-|-|-|-|-|-|-|-|-|-|-|-|-|
> | **T** | 0 | 10 | 1 | 2 | 3 | 4 | 5 | 6| 7 | 8 | 9 | 10 (ours)|
> |ImageNet| 66.6 | 70.8 | 70.7| 80.0| 81.0| 81.4| 81.5| 81.5|81.6|81.6| 81.6 | 81.6  |
> | SUN397|62.5|64.5|66.1|73.5|74.7|75.1|75.3|75.4| 75.4|75.5|75.5|75.5|
> | DTD |43.5|47.1|45.4|47.6|48.3|48.6|48.8|48.9| 48.9|48.9|49.0|49.0|
>
> | m  | 0  | 1|3 (ours)|5|10|
> |-|-|-|-|-|-|
> | ImageNet| 80.0| 81.3| 81.6| 81.6| 81.7|
> | SUN397 | 73.6| 75.1| 75.5| 75.7| 75.9|
> | DTD | 47.7| 48.7| 49.0| 48.8| 48.6|
>
> The results show that: (1) MOON exhibits strong robustness to variations in both hyperparameters; (2) MOON converges remarkably fast within $T=3$ iterations with SoTA performance. We will include these results in the final version.
>
> > Q5: Discussion of other transductive methods
>
> Existing transductive methods can be briefly categorized as follows:
>
> - **Non-parametric models:** ZLaP leverage label propagation via graph structures and geodesic distances on the data manifold to address the modality gap.
> - **Parametric probabilistic models:** EM-Dirichlet feature distributions on the unit simplex, and GDA-CLIP applies Gaussian Discriminant Analysis with closed-form parameter estimation.
> - **PLE-style mixture models:** TransCLIP performs GMM soft clustering with a text-guided KL penalty and a Laplacian term. StatA introduces a KL anchor term for regularization during test-time. ADAPT extends StatA to online mode with a knowledge bank.
>
> Compared with previous SoTA method StatA, our MOON's core advantages are:
>
> 1.  We systematically analyze transductive methods from the perspective of PLE, and theoretically obtain the dynamic shrinkage form. This explains the brittleness of previous methods along with two identified limitations.
> 2. We introduce a faster and better mixture of vMF distributions to model features. After that, based on our findings above, we introduce two dynamic shrinkage weights based on zero-shot priors.
> 3. The above modifications lead to significantly better efficiency and superior performance.
>
> We promise to add this in the method section of our final manuscript.

---

> > ### Author Rebuttal · Reviewer_dAzL · 2026-04-01
> >
> > I have read the author's rebuttal carefully and will maintain the positive score.

---

> > > ### Author Response · Authors · 2026-04-01
> > >
> > > Thank you for your response, we are happy to have addressed your concerns! We really appreciate your efforts on reviewing our paper, your insightful comments and support.

---

### Official Review · Reviewer_a9nn · 2026-03-10

**Soundness:** 3
**Presentation:** 4
**Significance:** 3
**Originality:** 3
**Overall Recommendation:** 4
**Confidence:** 4

**Summary:**

This work studies test-time transduction for VLMs under realistic class imbalance through a penalized likelihood estimation (PLE) framework, proving that KL-anchored PLE induces adaptive shrinkage for the entire exponential family, and uses this insight to identify two concrete limitations of prior work. The authors propose MOON, a mixture of von Mises-Fisher distributions with dynamic shrinkage weights at both instance level (entropy-based) and class level (confidence-based). The method is empirically validated across 11 datasets, multiple backbone architectures, and realistic imbalanced settings.

**Compliance With Llm Reviewing Policy:**

Affirmed.

**Final Justification:**

I have read the author's rebuttal carefully and thus, will maintain the positive score.

**Key Questions For Authors:**

1. Why is the geometric mean specifically chosen in Eq. 10 over simpler alternatives like average or max confidence?
2. Is there a practical threshold of K above which StatA is simply a better choice than MOON?
3. Was incorporating γi into the assignment update considered, and if so, what was the empirical or theoretical reason for excluding it?

**Limitations:**

Yes.

**Strengths And Weaknesses:**

**Strengths**
1. The method is well motivated where theoretical analysis motivates the method instead of using theory as a post-hoc justification.
2. The problem addressed is practically important and underexplored. Most TTA benchmarks assume balanced class distributions, which is not always true in deployment.
3. The paper is clearly written and well-structured. The narrative flows logically from problem motivation through theoretical analysis to method design, with each component clearly justified before being introduced.
4. The self-evaluation adds value to the work, the failures of naive vMF are well explained.

**Weaknesses**
1. The class-level weight (Eq. 10) uses a geometric mean of average and maximum class confidence. This design choice feels under-examined and would be rather insightful to compare to average-only or a harmonic mean for example.
2. The instance-level weight cancels in the assignment update (Eq. 11) and only affects parameter estimation indicating that the uncertain samples are down-weighted in the M-step but still fully participate in graph smoothing during the assignment update. If a sample is deemed uncertain enough to be down-weighted in parameter estimation, it is unclear why its label uncertainty should still propagate freely.

---

> ### Author Rebuttal · Authors · 2026-03-30
>
> Thanks for your time and effort, we really appreciate your recognition of our work! We are willing to take your valuable suggestions and address each of your concerns to improve this paper as below:
>
> > Q1: Design choice of $\lambda = \sqrt{ \text{mean}_i \hat{\mathbf{y}}_{i} \odot \max_{i}\hat{\mathbf{y}}_{i}}$
>
> PDA methods directly use average class confidence as the target label space is fixed and average confidence reliably reflects global frequencies. In test-time settings, however, the effective label set varies across batches. Therefore, relying solely on average confidence might confuse those rare-but-valid classes with true outliers: $\lambda_{eff} \gg \lambda_{rare} \approx \lambda_{outlier}$, which instead suppresses positive transfer. Using only max confidence treats rare and effective (dominant) classes almost equally: $\lambda_{eff} \approx \lambda_{rare} \gg \lambda_{outlier}$. Combining them with means helps distinguish these classes: $\lambda_{eff} > \lambda_{rare} \gg \lambda_{outlier}$.
>
> We empirically validate this across 3 extreme imbalance and 3 mild (near-uniform) scenarios:
>
> (*Extreme 1*: Bs=64, Batch, Low; *Extreme 2*: Bs=1000, Batch, Medium
>
> *Extreme 3*: Bs=128, Online, High; *Mild 1*: Bs=64, Batch, Medium
>
> *Mild 2*: Bs=1000, Batch, Very high; *Mild 3*: Bs=128, Online, Low)
>
> |                       | Extreme 1 | Extreme 2 | Extreme 3 | Mild 1   | Mild 2   | Mild 3   |
> | --------------------- | --------- | --------- | --------- | -------- | -------- | -------- |
> | Geometric mean (ours) | 73.1      | 72.9      | 73.4      | **69.4** | **69.2** | 66.5     |
> | Arithmetic mean       | **73.5**  | **74.2**  | **74.1**  | 69.0     | 68.6     | 65.7     |
> | Harmonic mean         | 73.4      | 73.5      | 73.8      | 69.3     | 69.0     | 66.1     |
> | Average only          | 72.4      | 73.3      | 73.9      | 68.2     | 67.1     | 66.7     |
> | Max only              | 71.7      | 70.8      | 71.6      | 69.0     | 68.9     | **66.9** |
>
> As shown, only using average or max confidence overfits to extreme or mild scenarios. Various types of means could mitigate this issue similarly. We choose geometric mean as it yields a more generalizable solution across broader scenarios, especially those mild ones where MOON performs below average.
>
> > Q2: Is there a practical threshold of K above which StatA is simply a better choice than MOON?
>
> Rather than a strict, quantitative threshold for $K$, the performance boundary is qualitatively determined by the imbalance ratio: $K_{eff}/\min(K,N)$. As it decreases, MOON’s advantage over StatA increases. Thus, fixing $N$ and $K_{eff}$, a larger $K$ inherently favors MOON.
>
> As shown in Appendix I, MOON consistently outperforms or matches StatA across diverse model architectures, scales, and types. A rare exception is the near-uniform "all-class" setting (Tab. 16). As detailed in **qru3@Q4**, this stems from MOON's inherent inductive bias. Despite that, MOON remains a superior choice due to its exceptional efficiency and practicality.
>
> > Q3: Incorporating γi into the assignment update
>
> Thanks for your insightful suggestion. We evaluated two variants that incorporate $\gamma$ into graph smoothing: **(1)** applying $\gamma_i$ only to the Laplacian term in the objective Eq. (7) , where $\gamma_i$ does not affect parameter update; **(2)** extra scaling $\sum_j\omega_{ij}\mathbf{z}_j^{(t)}$ in Eq. (11) with $\gamma_i$ while keeping the original objective unchanged, where $\gamma_i$ affects both parameter and assignment update.
>
> (Bs=1000, Batch Medium)
>
> | Method    | ImageNet | SUN397   | Aircraft | EuroSAT  | StanfordCars | Food101  | Pets     | Flower102 | Caltech101 | DTD      | UCF101   | Average  |
> | --------- | -------- | -------- | -------- | -------- | ------------ | -------- | -------- | --------- | ---------- | -------- | -------- | -------- |
> | MOON      | **81.6** | **75.5** | **29.6** | 58.9     | **76.1**     | **93.1** | **92.4** | **77.3**  | **94.9**   | **49.0** | 73.6     | **72.9** |
> | Variant 1 | 81.4     | 75.3     | 29.5     | 52.1     | 76.0         | 93.0     | 92.1     | 77.1      | 94.8       | 48.8     | **73.7** | 72.2     |
> | Variant 2 | 81.4     | 75.2     | 29.5     | **59.0** | 75.9         | 93.0     | 92.2     | 77.1      | 94.8       | 48.6     | 73.6     | 72.7     |
>
> As shown above, both variants slightly underperform original MOON. We believe $\gamma_i$ is better used as an reliability weight during parameter estimation, rather than a gate on graph propagation. Since uncertain samples are precisely those that benefit most from neighborhood label propagation, suppressing them in the assignment update instead weakens correction and keeps them closer to noisy zero-shot priors.

---

> > ### Author Rebuttal · Reviewer_a9nn · 2026-04-01
> >
> > I have read the author's rebuttal carefully and will maintain the positive score.

---

> > > ### Author Response · Authors · 2026-04-01
> > >
> > > Thanks again for your insightful comments! We are pleased to know that you agree with our response, and your feedback has been instrumental in enhancing the quality of our work. Please do not hesitate to let us know if you need any clarification or have additional suggestions!

---

### Official Review · Reviewer_qru3 · 2026-03-13

**Soundness:** 2
**Presentation:** 3
**Significance:** 2
**Originality:** 3
**Overall Recommendation:** 4
**Confidence:** 3

**Summary:**

This paper studies test-time transduction for vision-language models under more realistic class-imbalanced settings, rather than the commonly assumed balanced setting. The central claim is that existing transductive methods are brittle because they either lack an anchoring mechanism or use a fixed shrinkage strength, which can cause unreliable local statistics to dominate and lead to collapse. To address this, the paper revisits transduction through a penalized likelihood estimation framework with KL-based anchoring, derives an adaptive shrinkage interpretation, and then proposes MOON, a training-free method based on a von Mises-Fisher mixture model with dynamic instance-level and class-level shrinkage driven by zero-shot priors. Empirically, the method is evaluated on 11 datasets under batch and online realistic test-time settings, where it generally improves over zero-shot CLIP and prior transductive baselines while remaining computationally efficient.

**Compliance With Llm Reviewing Policy:**

Affirmed.

**Final Justification:**

Thank you for the rebuttal. My main concerns have been largely addressed, and the response provided enough clarification on the issues raised in my original review. As a result, I am changing my score from Weak Reject to Weak Accept.

**Key Questions For Authors:**

1. How sensitive is MOON to the specific definition of class confidence in Eq. (10)?
2. How well does the method transfer beyond CLIP ViT-B/16 to other backbones or VLM families?
3. Can the paper provide deeper analysis of the scenarios where MOON underperforms its own average trend, such as lower-correlation online settings or certain datasets?
4. What are the concrete convergence guarantees under the implemented algorithm, especially after simplifying the inner-loop optimization to a single pass per iteration?

**Limitations:**

Yes

**Strengths And Weaknesses:**

**Strengths**
1. This paper provides a shrinkage-based reformulation that offers a explanation for why prior transductive methods may fail under realistic imbalance.
2. This paper proposes a method that is matched to normalized CLIP embeddings, and the overall design is training-free.

**Weaknesses**
1. This paper relies on a class-level weighting design based on the inverse of a confidence score defined by the geometric mean of average and maximum confidence, but the reason for this specific choice is not fully justified and still feels somewhat heuristic.
2. This paper mainly evaluates the method on CLIP ViT-B/16 and gives limited analysis of weaker-result settings such as low-correlation online streams or datasets with small/negative gains, so the robustness and scope of the method are not yet fully established.

---

> ### Author Rebuttal · Authors · 2026-03-30
>
> Thanks for your time and effort! We really appreciate your valuable suggestions to improve this paper, and we are willing to address each of your concerns as below:
>
> > Q1: Design choice of $\alpha = 1/\lambda$
>
> Different from PDA, $\alpha_k$ is not the final transfer weight itself; it instead controls the shrinkage strength $\beta_k=\frac{n_k}{n_k+\alpha_k}.$ Therefore, $\alpha_k$ should be **negatively correlated** with $\lambda_k$. We choose the inverse-form $\alpha$, as this form is widely used in statistical learning to impose regularization on less reliable signals [1]. We also compared an alternative form $\exp{(-\lambda_k)}$ but found it less effective. A possible reason is that it bounds $\alpha_k$ in $(0,1]$ and in turn bounds $\beta_k$ within $[\frac{n_k}{n_k+1},1)$, leading the update to remain noticeably influenced by empirical estimate. The results below further support our choice:
>
> (Bs=1000, Batch Medium)
>
> | $\alpha$           | ImageNet | SUN397 | Aircraft | EuroSAT | StanfordCars | Food101 | Pets | Flower102 | Caltech101 | DTD  | UCF101 | Average  |
> |-|-|-|-|-|-|-|-|-|-|-|-|-|
> | $1/\lambda $(ours) | 81.6     | 75.5   | 29.6     | 58.9    | 76.1         | 93.1    | 92.4 | 77.3      | 94.9       | 49.0 | 73.6   | **72.9** |
> | $\lambda$          | 68.5     | 63.9   | 26.1     | 58.7    | 67.9         | 87.3    | 90.3 | 73.7      | 92.9     | 45.7 | 68.7   | 67.6     |
> | $\exp(-\lambda)$   | 71.1     | 66.1   | 27.0    | 58.8    | 69.9         | 88.5    | 90.8 | 74.6      | 93.6     | 46.4 | 70.2   | 68.8     |
>
> [1] The adaptive lasso and its oracle properties[J]. Journal of the American statistical association, 2006, 101(476): 1418-1429.
>
> > Q2: Design choice of confidence $\lambda$
>
> Unlike PDA, relying solely on average confidence in test-time might confuse rare-but-valid classes with true outlier classes. Using a mean of the average and maximum confidence resolves this ambiguity. We specifically choose geometric mean as it yields a more generalizable solution across diverse scenarios. Please see **a9nn@Q1** for details and experiments.
>
> > Q3: transferring to other CLIP backbones and VLMs
>
> Thanks for suggestion. We have extended our evaluation to include 4 additional CLIP backbones and 3 other VLMs:
>
> |               |         | ImageNet  |      | Average   |      |
> |-|-|-|-|-|-|
> |               | #Params | Zero-shot | Ours | Zero-shot | Ours |
> | CLIP RN50     | 102M    | 58.2      | 76.8 | 58.7    | 65.7 |
> | CLIP RN101    | 120M    | 61.3      | 77.1 | 59.5    | 64.8 |
> | CLIP ViT-B/32 | 151M    | 62.0      | 78.5 | 61.9    | 68.0 |
> | CLIP ViT-L/14 | 428M    | 73.5      | 84.9 | 72.6    | 77.4 |
> | OpenCLIP      | 150M    | 73.0      | 83.4 | 72.5    | 76.7 |
> | SigLIP        | 878M    | 82.3      | 91.3 | 81.8    | 85.1 |
> | EVA-CLIP      | 1.14B   | 78.0      | 81.2 | 76.5    | 78.4 |
>
> As shown, our MOON maintains universal effectiveness across diverse model architectures, scales, and types. For more detailed and comprehensive results, please refer to **Appendix I.2 and I.3**.
>
> > Q4: Deeper analysis of the scenarios where MOON underperforms its average trend
>
> This stems from MOON's inherent design. Since our dynamic shrinkage mechanism assumes a source-target label space mismatch, it introduces an inductive bias favoring extreme imbalances (e.g., $K_{eff} \ll K$ or $\xi \to 0$). In scenarios closer to a uniform distribution, this leads to slight over-regularization and thus underperform the average trend. Please refer to **a9nn@Q2** for further analyses.
>
> > Q5: Concrete convergence guarantee of optimization algorithm
>
> Our algorithm can be analyzed within the standard BSUM framework. Let $\mathcal{L}(z,\Theta)$ denotes the objective and $\Theta=(\mu_k,\kappa_k)_{k=1}^K$. Given that:
>
> - Affinity matrix $W$ is PSD;
> - Anchor weights $\alpha_k$ and $\gamma_i > 0$;
> - Fixing $z$, the parameter subproblem in $\Theta$ has a unique minimizer.
>
> Then each outer iteration of our algorithm is a valid BSUM step:
>
> 1. **z-update**: The only non-convex part is the Laplacian term. Since $W$ is PSD, this term is concave and can be upper-bounded by its first-order Taylor expansion at the current iteration (proofs in Appendix G.1). Minimizing this tight surrogate yields Eq. (11), i.e., the update is an exact minimizer of the majorized subproblem.
> 2. **$\Theta$ update**: Fixing $z$, our objective is strictly convex, and Eqs. (12)–(13) are the corresponding closed-form exact updates. This step further decreases the objective.
>
> Therefore, each outer iteration satisfies
> $$
> \mathcal{L} (z^{(t+1)},\Theta^{(t+1)}) \le \mathcal{L} (z^{(t)},\Theta^{(t)}).
> $$
> Since $\mathcal{L}$ is lower-bounded, the objective sequence **is monotonically non-increasing and thus convergent**. Moreover, every limit point of the iterate sequence is a coordinate-wise minimum. And, as BSUM doe not require solving subproblems to full convergence at each outer iteration, **single-pass z-update preserves convergence guarantee**.

---

> > ### Author Rebuttal · Reviewer_qru3 · 2026-04-04
> >
> > Thank you for the rebuttal. My main concerns have been largely addressed, and the response provided enough clarification on the issues raised in my original review. As a result, I am changing my score from Weak Reject to Weak Accept.

---

> > > ### Author Response · Authors · 2026-04-04
> > >
> > > Thanks for your positive feedback and raising the score! We are glad to know that we have addressed your concerns and issues, and we promise to further improve our paper by incorporating the rebuttal discussions into the final version. We sincerely thank you for your expertise, dedication and effort on our submission!

---

### Decision · Program_Chairs · 2026-04-30

**Decision:**

Accept (regular)

**Comment:**

The paper addresses the challenge of test-time transduction for vision-language models (VLMs) like CLIP in realistic scenarios characterized by class imbalance. The authors argue that existing transductive methods often fail under these conditions because they lack an anchoring mechanism or use fixed heuristics, causing unreliable local statistics to dominate. To solve this, the paper introduces MOON, a training-free method based on a mixture of von Mises-Fisher (vMF) distributions. The method is grounded in a penalized likelihood estimation (PLE) framework, which the authors theoretically prove induces more effective solutions at both the instance and class levels.

Reviewers consistently praised the paper's strong theoretical foundation, noting that the analysis of adaptive shrinkage for the exponential family provides a solid justification for the proposed method. The problem of class imbalance in test-time adaptation (TTA) is recognized as practically significant and underexplored, as most benchmarks assume balanced distributions. Empirically, MOON was evaluated across multiple standard datasets in both batch and online settings, demonstrating improved performance over zero-shot CLIP and prior baselines while maintaining computational efficiency.

Several points of concern were raised during the review process, some of them being the following.
- The specific use of the geometric mean for class-level weighting was questioned for its lack of full justification.
- Reviewers sought clarity on why uncertain samples are down-weighted in parameter estimation but still participate in graph smoothing.
- There was a perceived need for better comparisons with episodic TTA and a clearer distinction from prior works addressing imbalanced TTA.
- Initial reviews pointed to poor readability regarding mathematical symbols.

Through the rebuttal process, the authors addressed these concerns by providing deeper clarifications on the experimental settings and technical definitions. While some reviewers noted that the analysis of extreme cases (e.g., very small batch sizes) remains limited, the overall consensus shifted positively. The reviewers mostly agreed that the paper provides a technically sound and theoretically motivated solution to a real-world problem in VLM deployment. Although some heuristic components remain and the evaluation could be even broader, the work offers a significant contribution that the community is likely to build upon.